# Universal Online Learning with Gradient Variations: A Multi-layer Online Ensemble Approach

**Yu-Hu Yan, Peng Zhao, Zhi-Hua Zhou**
National Key Laboratory for Novel Software Technology,
Nanjing University, Nanjing 210023, China
{yanyh, zhaop, zhouzh}@lamda.nju.edu.cn

## Abstract

In this paper, we propose an online convex optimization approach with two different levels of adaptivity. On a higher level, our approach is agnostic to the unknown types and curvatures of the online functions, while at a lower level, it can exploit the unknown niceness of the environments and attain problem-dependent guarantees. Specifically, we obtain $\mathcal{O}(\log V_T)$, $\mathcal{O}(d \log V_T)$ and $\widehat{\mathcal{O}}(\sqrt{V_T})$ regret bounds for strongly convex, exp-concave and convex loss functions, respectively, where $d$ is the dimension, $V_T$ denotes problem-dependent gradient variations and the $\widehat{\mathcal{O}}(\cdot)$-notation omits $\log V_T$ factors. Our result not only safeguards the worst-case guarantees but also directly implies the small-loss bounds in analysis. Moreover, when applied to adversarial/stochastic convex optimization and game theory problems, our result enhances the existing universal guarantees. Our approach is based on a multi-layer online ensemble framework incorporating novel ingredients, including a carefully designed optimism for unifying diverse function types and cascaded corrections for algorithmic stability. Notably, despite its multi-layer structure, our algorithm necessitates only one gradient query per round, making it favorable when the gradient evaluation is time-consuming. This is facilitated by a novel regret decomposition equipped with carefully designed surrogate losses.

## 1  Introduction

Online convex optimization (OCO) is a versatile model that depicts the interaction between a learner and the environments over time [Hazan, 2016, Orabona, 2019]. In each round $t \in [T]$, the learner selects a decision $\mathbf{x}_t$ from a convex compact set $\mathcal{X} \subseteq \mathbb{R}^d$, and simultaneously the environments choose a convex loss function $f_t : \mathcal{X} \mapsto \mathbb{R}$. Subsequently, the learner incurs a loss $f_t(\mathbf{x}_t)$, obtains information about the online function, and updates the decision to $\mathbf{x}_{t+1}$, aiming to optimize the game-theoretical performance measure known as *regret* [Cesa-Bianchi and Lugosi, 2006]:

$$\text{REG}_T \triangleq \sum_{t=1}^{T} f_t(\mathbf{x}_t) - \min_{\mathbf{x} \in \mathcal{X}} \sum_{t=1}^{T} f_t(\mathbf{x}), \tag{1.1}$$

which represents the learner's excess loss compared to the best fixed comparator in hindsight.

In OCO, the type and curvature of online functions significantly impact the minimax regret bounds. Specifically, for convex functions, online gradient descent (OGD) can achieve an $\mathcal{O}(\sqrt{T})$ regret guarantee [Zinkevich, 2003]. For $\alpha$-exp-concave functions, online Newton step (ONS), with prior knowledge of the curvature coefficient $\alpha$, attains an $\mathcal{O}(d \log T)$ regret [Hazan et al., 2007]. For $\lambda$-strongly convex functions, OGD with prior knowledge of the curvature coefficient $\lambda$ and a different parameter configuration enjoys an $\mathcal{O}(\log T)$ regret [Hazan et al., 2007]. Note that the above

---

*Correspondence: Peng Zhao <zhaop@lamda.nju.edu.cn>

37th Conference on Neural Information Processing Systems (NeurIPS 2023).

Table 1: Comparison with existing results. The second column presents the regret bounds for various kinds of functions, where $V_T$ and $F_T$ are problem-dependent quantities that are at most $\mathcal{O}(T)$ and can be much smaller in nice environments. The $\widehat{\mathcal{O}}(\cdot)$-notation omits logarithmic factors on $V_T$ and $F_T$. The third column shows the gradient query complexity. All problem-dependent bounds require the smoothness of online functions.

| Works | Regret Bounds | | | Gradient Query |
|---|---|---|---|---|
| | Strongly Convex | Exp-concave | Convex | |
| van Erven and Koolen [2016] | $\mathcal{O}(d \log T)$ | $\mathcal{O}(d \log T)$ | $\mathcal{O}(\sqrt{T})$ | 1 |
| Wang et al. [2019] | $\mathcal{O}(\log T)$ | $\mathcal{O}(d \log T)$ | $\mathcal{O}(\sqrt{T})$ | 1 |
| Zhang et al. [2022a] | $\mathcal{O}(\min\{\log V_T, \log F_T\})$ | $\mathcal{O}(d \min\{\log V_T, \log F_T\})$ | $\mathcal{O}(\sqrt{F_T})$ | $\mathcal{O}(\log T)$ |
| **Ours** | $\mathcal{O}(\min\{\log V_T, \log F_T\})$ | $\mathcal{O}(d \min\{\log V_T, \log F_T\})$ | $\widehat{\mathcal{O}}(\min\{\sqrt{V_T}, \sqrt{F_T}\})$ | 1 |

algorithms require the function type and curvature beforehand and does not consider the niceness of environments. Recent studies further strengthen the algorithms and results with *two levels of adaptivity*. The higher-level adaptivity requires an algorithm to be agnostic to the unknown types and curvatures of the online functions. And the lower-level adaptivity requires an algorithm to exploit the unknown niceness of the environments within a specific function family. In the following, we delve into an extensive discussion on these two levels of adaptivity.

## 1.1 High Level: Adaptive to Unknown Curvature of Online Functions

Traditionally, the learner needs to know the function type and curvature in advance to select suitable algorithms (and parameter configurations), which can be burdensome in practice. *Universal* online learning [van Erven and Koolen, 2016, Wang et al., 2019, Zhang et al., 2022a] aims to develop a single algorithm agnostic to the specific function type and curvature while achieving the same regret guarantees as if they were known. The pioneering work of van Erven and Koolen [2016] proposed a single algorithm called MetaGrad that achieves an $\mathcal{O}(\sqrt{T})$ regret for convex functions and an $\mathcal{O}(d \log T)$ regret for exp-concave functions. Later, Wang et al. [2019] further obtained the optimal $\mathcal{O}(\log T)$ regret for strongly convex functions. Notably, these approaches are efficient regarding the gradient query complexity, by using only one gradient within each round.

However, the above approaches are not flexible enough since they have to optimize a group of heterogeneous and carefully-designed surrogate loss functions, which can be cumbersome and challenging. To this end, Zhang et al. [2022a] introduced a simple framework that operates on the original online functions with the same optimal results at the expense of $\mathcal{O}(\log T)$ gradient queries.

## 1.2 Low Level: Adaptive to Unknown Niceness of Online Environments

Within a specific function family, the algorithm's performance is also substantially influenced by the niceness of environments. This concept is usually captured through *problem-dependent* quantities in the literature. Therefore, it becomes essential to develope adaptive algorithms with problem-dependent regret guarantees. Specifically, we consider the following problem-dependent quantities:

$$F_T \triangleq \min_{\mathbf{x} \in \mathcal{X}} \sum_{t=1}^{T} f_t(\mathbf{x}), \text{ and } V_T \triangleq \sum_{t=2}^{T} \sup_{\mathbf{x} \in \mathcal{X}} \|\nabla f_t(\mathbf{x}) - \nabla f_{t-1}(\mathbf{x})\|^2,$$

where the small loss $F_T$ represents the cumulative loss of the best comparator [Srebro et al., 2010, Orabona et al., 2012] and the gradient variation $V_T$ characterizes the variation of the function gradients [Chiang et al., 2012]. In particular, the gradient-variation bound demonstrates its fundamental importance in modern online learning from the following three aspects: *(i)* it safeguards the worst-case guarantees in terms of $T$ and implies the small-loss bounds in analysis directly; *(ii)* it draws a profound connection between adversarial and stochastic convex optimization; and *(iii)* it is crucial for fast convergence rates in game theory. We will explain the three aspects in detail in the next part.

## 1.3 Our Contributions and Techniques

In this paper, we consider the two levels of adaptivity simultaneously and propose a novel universal approach that achieves $\mathcal{O}(\log V_T)$, $\mathcal{O}(d \log V_T)$ and $\widehat{\mathcal{O}}(\sqrt{V_T})$ regret bounds for strongly convex,

exp-concave and convex loss functions, respectively, using only one gradient query per round, where $\widehat{\mathcal{O}}(\cdot)$-notation omits factors on $\log V_T$. Table 1 compares our results with existing ones. In summary, relying on the basic idea of online ensemble [Zhao et al., 2021], our approach primarily admits a *multi-layer online ensemble* structure with several important novel ingredients. Specifically, we propose a carefully designed optimism, a hyper-parameter encoding historical information, to handle different kinds of functions universally, particularly exp-concave functions. Nevertheless, it necessitates careful management of the stability of final decisions, which is complicated in the multi-layer structure. To this end, we analyze the negative stability terms in the algorithm and propose cascaded correction terms to realize effective collaboration among layers, thus enhancing the algorithmic stability. Moreover, we facilitate a novel regret decomposition equipped with carefully designed surrogate losses to achieve only one gradient query per round, making our algorithm as efficient as van Erven and Koolen [2016] regarding the gradient complexity. Our result resolves an open problem proposed by Zhang et al. [2022a], who have obtained partial results for exp-concave and strongly convex functions and asked whether it is possible to design designing a single algorithm with universal gradient-variation bounds. Among them, the convex case is particularly important because the improvement from $T$ to $V_T$ is polynomial, whereas logarithmic in the other cases.

Next, we shed light on some applications of our approach. First, it safeguards the worst-case guarantees [van Erven and Koolen, 2016, Wang et al., 2019] and directly implies the small-loss bounds of Zhang et al. [2022a] in analysis. Second, gradient variation is shown to play an essential role in the stochastically extended adversarial (SEA) model [Sachs et al., 2022, Chen et al., 2023b], an interpolation between stochastic and adversarial OCO. Our approach resolves a major open problem left in Chen et al. [2023b] on whether it is possible to develop a single algorithm with universal guarantees for strongly convex, exp-concave, and convex functions in the SEA model. Third, in game theory, gradient variation encodes the changes in other players' actions and can thus lead to fast convergence rates [Rakhlin and Sridharan, 2013b, Syrgkanis et al., 2015, Zhang et al., 2022b]. We demonstrate the universality of our approach by taking two-player zero-sum games as an example.

**Technical Contributions.** Our first contribution is proposing a multi-layer online ensemble approach with effective collaboration among layers, which is achieved by a *carefully-designed optimism* to unify different kinds of functions and *cascaded correction terms* to improve the algorithmic stability within the multi-layer structure. The second contribution arises from efficiency. Although there are multiple layers, our algorithm only requires one gradient query per round, which is achieved by a *novel regret decomposition* equipped with carefully designed surrogate losses. Two interesting byproducts rises in our approach. The first one is the negative stability term in the analysis of MsMwC [Chen et al., 2021], which serves as an important building block of our algorithm. And the second byproduct contains a simple approach and analysis for the optimal worst-case universal guarantees, using one gradient query within each round.

**Organization.** The rest of the paper is structured as follows. Section 2 provides preliminaries. Section 3 proposes our multi-layer online ensemble approach for universal gradient-variation bounds. Section 4 further improves the gradient query complexity. Due to page limits, the applications of our proposed algorithm are deferred to Appendix A. All the proofs can be found in the appendices.

## 2 Preliminaries

In this section, we introduce some preliminary knowledge, including our assumptions, the definitions, the formal problem setup, and a review of the latest progress of Zhang et al. [2022a].

To begin with, we list some notations. Specifically, we use $\|\cdot\|$ for $\|\cdot\|_2$ in default and use $\sum_t, \sum_k, \sum_i$ as abbreviations for $\sum_{t \in [T]}, \sum_{k \in [K]}$ and $\sum_{i \in [N]}$. $a \lesssim b$ represents $a \leq \mathcal{O}(b)$. $\widehat{\mathcal{O}}(\cdot)$-notation omits logarithmic factors on leading terms. For example, $\widehat{\mathcal{O}}(\sqrt{V})$ omits the dependence of $\log V$.

**Assumption 1** (Boundedness). For any $\mathbf{x}, \mathbf{y} \in \mathcal{X}$ and $t \in [T]$, the domain diameter satisfies $\|\mathbf{x} - \mathbf{y}\| \leq D$, and the gradient norm of the online functions is bounded by $\|\nabla f_t(\mathbf{x})\| \leq G$.

**Assumption 2** (Smoothness). All online functions are $L$-smooth: $\|\nabla f_t(\mathbf{x}) - \nabla f_t(\mathbf{y})\| \leq L\|\mathbf{x} - \mathbf{y}\|$ for any $\mathbf{x}, \mathbf{y} \in \mathcal{X}$ and $t \in [T]$.

Both assumptions are common in the literature. Specifically, the boundedness assumption is common in OCO [Hazan, 2016]. The smoothness assumption is essential for first-order algorithms to

achieve gradient-variation bounds [Chiang et al., 2012]. Strong convexity and exp-concavity are defined as follows. For any $\mathbf{x}, \mathbf{y} \in \mathcal{X}$, a function $f$ is $\lambda$-strongly convex if $f(\mathbf{x}) - f(\mathbf{y}) \leq \langle \nabla f(\mathbf{x}), \mathbf{x} - \mathbf{y} \rangle - \frac{\lambda}{2}\|\mathbf{x} - \mathbf{y}\|^2$, and is $\alpha$-exp-concave if $f(\mathbf{x}) - f(\mathbf{y}) \leq \langle \nabla f(\mathbf{x}), \mathbf{x} - \mathbf{y} \rangle - \frac{\alpha}{2}\langle \nabla f(\mathbf{x}), \mathbf{x} - \mathbf{y} \rangle^2$. Note that the formal definition of $\beta$-exp-concavity states that $\exp(-\beta f(\cdot))$ is concave. Under Assumption 1, $\beta$-exp-concavity leads to our definition with $\alpha = \frac{1}{2}\min\{\frac{1}{4GD}, \beta\}$ [Hazan, 2016, Lemma 4.3]. For simplicity, we use it as an alternative definition for exp-concavity.

In the following, we formally describe the problem setup and briefly review the key insight of Zhang et al. [2022a]. Concretely, we consider the problem where the learner has no prior knowledge about the function type (strongly convex, exp-concave, or convex) or the curvature coefficient ($\alpha$ or $\lambda$). Without loss of generality, we study the case where the curvature coefficients $\alpha, \lambda \in [1/T, 1]$. This requirement is natural because if $\alpha$ (or $\lambda$) $< 1/T$, even the optimal regret is $\Omega(T)$ [Hazan et al., 2007], which is vacuous. Conversely, functions with $\alpha$ (or $\lambda$) $> 1$ are also 1-exp-concave (or 1-strongly convex). Thus using $\alpha$ (or $\lambda$) $= 1$ will only worsen the regret by a constant factor, which can be omitted. This condition is also used in previous works [Zhang et al., 2022a].

**A Brief Review of Zhang et al. [2022a].** A general solution to handle the uncertainty is to leverage a two-layer framework, which consists of a group of base learners exploring the environments and a meta learner tracking the best base learner on the fly. To handle the unknown curvature coefficients $\alpha$ and $\lambda$, the authors discretize them into the following candidate pool:

$$\mathcal{H} \triangleq \{1/T, 2/T, 4/T, \ldots, 1\}, \tag{2.1}$$

where $|\mathcal{H}| \approx \log T$. Consequently, they design three groups of base learners:

- *(i)* about $\log T$ base learners, each of which runs the algorithm for strongly convex functions with a guess $\lambda_i \in \mathcal{H}$ of the strong convexity coefficient $\lambda$;
- *(ii)* about $\log T$ base learners, each of which runs the algorithm for exp-concave functions with a guess $\alpha_i \in \mathcal{H}$ of the exp-concavity coefficient $\alpha$;
- *(iii)* 1 base learner that runs the algorithm for convex functions.

Overall, they maintain $N \approx \log T + \log T + 1$ base learners. Denoting by $\boldsymbol{p}_t \triangleq (p_{t,1}, \ldots, p_{t,N})$ the meta learner's weights and $\mathbf{x}_{t,i}$ the $i$-th base learner's decision, the learner submits $\mathbf{x}_t = \sum_i p_{t,i}\mathbf{x}_{t,i}$.

In the two-layer framework, the regret (1.1) can be decomposed into two terms:

$$\text{REG}_T = \left[\sum_{t=1}^{T} f_t(\mathbf{x}_t) - \sum_{t=1}^{T} f_t(\mathbf{x}_{t,i^\star})\right] + \left[\sum_{t=1}^{T} f_t(\mathbf{x}_{t,i^\star}) - \min_{\mathbf{x} \in \mathcal{X}}\sum_{t=1}^{T} f_t(\mathbf{x})\right], \tag{2.2}$$

where the *meta regret* (first term) assesses how well the algorithm tracks the best base learner, and the *base regret* (second term) measures the performance of it. The best base learner is the one which runs the algorithm matching the ground-truth function type with the most accurate guess of the curvature — taking $\alpha$-exp-concave functions as an example, there must exist a base learner indexed by $i^\star$, whose coefficient $\alpha_{i^\star} \in \mathcal{H}$ satisfies $\alpha_{i^\star} \leq \alpha \leq 2\alpha_{i^\star}$.

A direct benefit of the above decomposition is that the meta regret can be bounded by a *constant* $\mathcal{O}(1)$ for exp-concave and strongly convex functions, allowing the algorithm to perfectly inherit the gradient-variation bound from the base learner. Taking $\alpha$-exp-concave functions as an example, by definition, the meta regret can be bounded by $\sum_t r_{t,i^\star} - \frac{\alpha}{2}\sum_t r_{t,i^\star}^2$, where the first term $r_{t,i} \triangleq \langle \nabla f_t(\mathbf{x}_t), \mathbf{x}_t - \mathbf{x}_{t,i}\rangle$ denotes the linearized regret, and the second one is a negative term from exp-concavity. Choosing ADAPT-ML-PROD [Gaillard et al., 2014] as the meta algorithm bounds the first term $\sum_t r_{t,i^\star}$ by $\mathcal{O}(\sqrt{\sum_t (r_{t,i^\star})^2})$, which can be canceled by the negative term, leading to an $\mathcal{O}(1)$ meta regret. Due to the meta algorithm's benefits, their approach can inherit the gradient-variation guarantees from the base learner. Similar derivation also applies to strongly convex functions. However, their approach is not favorable enough in the convex case and is not efficient enough in terms of the gradient query complexity. We will give more discussions about the above issues in Section 3.1 and Section 4.

# 3 Our Approach

This section presents our multi-layer online ensemble approach with universal gradient-variation bounds. Specifically, in Section 3.1, we provide a novel optimism to unify different kinds of functions. In Section 3.2, we exploit two types of negative terms to cancel the positive term caused by

the optimism design. We summarize the overall algorithm in Section 3.3. Finally, in Section 3.4, we present the main results and list several applications of our approach.

## 3.1 Universal Optimism Design

In this part, we propose a novel optimism that simultaneously unifies various kinds of functions. We start by observing that Zhang et al. [2022a] does not enjoy gradient-variation bounds for convex functions, where the main challenge lies in obtaining an $\mathcal{O}(\sqrt{V_T})$ meta regret for convex functions while simultaneously maintaining an $\mathcal{O}(1)$ regret for exp-concave and strongly convex functions. In the following, we focus on the meta regret because the base regret optimization is straightforward by employing the optimistic online learning technique [Rakhlin and Sridharan, 2013a,b] in a black-box fashion. Optimistic online learning is essential in our problem since it can utilize the historical information, e.g., $\nabla f_{t-1}(\cdot)$ for our purpose due to the definition of the gradient variation.

Shifting our focus to the meta regret, we consider upper-bounding it by a second-order bound of $\mathcal{O}(\sqrt{\sum_t (r_{t,i^\star} - m_{t,i^\star})^2})$, where $r_{t,i} = \langle \nabla f_t(\mathbf{x}_t), \mathbf{x}_t - \mathbf{x}_{t,i} \rangle$ and the optimism $m_{t,i^\star}$ can encode historical information. Such a second-order bound can be easily obtained using existing prediction with expert advice algorithms, e.g., ADAPT-ML-PROD [Wei et al., 2016]. Nevertheless, as we will demonstrate in the following, designing an optimism $m_{t,i}$ that effectively unifies various function types is not straightforward, thereby requiring novel ideas in the optimism design.

To begin with, a natural impulse is to choose the optimism as $m_{t,i} = \langle \nabla f_{t-1}(\mathbf{x}_{t-1}), \mathbf{x}_{t,i} - \mathbf{x}_t \rangle$,[2] which yields the following second-order bound:

$$\sum_{t=1}^{T}(r_{t,i^\star} - m_{t,i^\star})^2 \lesssim \begin{cases} \sum_t \|\mathbf{x}_t - \mathbf{x}_{t,i^\star}\|^2, & \textit{(strongly convex)} \\ \sum_t \|\nabla f_t(\mathbf{x}_t) - \nabla f_{t-1}(\mathbf{x}_{t-1})\|^2, & \textit{(convex)} \end{cases}$$

where the inequality is due to the boundedness assumption (Assumption 1). This optimism design handles the strongly convex functions well since the bound can be canceled by the negative term imported by strong convexity (i.e., $-\|\mathbf{x}_t - \mathbf{x}_{t,i^\star}\|^2$). Moreover, it is quite promising for convex functions because the bound essentially consists of the desired gradient variation and a positive term of $\|\mathbf{x}_t - \mathbf{x}_{t-1}\|^2$ (we will deal with it later). However, it fails for exp-concave functions because the negative term imported by exp-concavity (i.e., $-\langle \nabla f_t(\mathbf{x}_t), \mathbf{x}_t - \mathbf{x}_{t,i^\star} \rangle^2$) cannot be used for cancellation due to the mismatch of the formulation.

To unify various kinds of functions, we propose a *novel optimism design* defined by $m_{t,i} = r_{t-1,i}$. This design aims to secure a second-order bound of $\mathcal{O}(\sqrt{\sum_t (r_{t,i^\star} - r_{t-1,i^\star})^2})$, which is sufficient to achieve an $\mathcal{O}(1)$ meta regret for exp-concave functions (with strong convexity being a subcategory thereof) while maintaining an $\mathcal{O}(\sqrt{V_T})$ meta regret for convex functions. The high-level intuition behind this approach is as follows: although the bound cannot be canceled exactly by the negative term imported by exp-concavity (i.e., $-r_{t,i^\star}^2$) within each round, it becomes manageable when aggregated *across the whole time horizon* as $\sum_t (r_{t,i^\star} - r_{t-1,i^\star})^2 \lesssim 4 \sum_t r_{t,i^\star}^2$, because $r_{t,i^\star}$ and $r_{t-1,i^\star}$ differs by merely a single time step. In the following, we propose the key lemma of the universal optimism design and defer the proof to Appendix B.1.

**Lemma 1** (Key Lemma). *Under Assumptions 1 and 2, if the optimism is chosen as $m_{t,i} = r_{t-1,i} = \langle \nabla f_{t-1}(\mathbf{x}_{t-1}), \mathbf{x}_{t-1} - \mathbf{x}_{t-1,i} \rangle$, it holds that*

$$\sum_{t=1}^{T}(r_{t,i^\star} - m_{t,i^\star})^2 \lesssim \begin{cases} \sum_{t=1}^{T} \langle \nabla f_t(\mathbf{x}_t), \mathbf{x}_t - \mathbf{x}_{t,i^\star} \rangle^2, & \textit{(exp-concave)} \\ V_T + \sum_{t=2}^{T} \|\mathbf{x}_{t,i^\star} - \mathbf{x}_{t-1,i^\star}\|^2 + \sum_{t=2}^{T} \|\mathbf{x}_t - \mathbf{x}_{t-1}\|^2. & \textit{(convex)} \end{cases}$$

Moreover, the second part of Lemma 1 shows that the bound for convex functions is also controllable by being further decomposed into three terms. The first term is the desired gradient variation. The

---

[2]Although $\mathbf{x}_t$ is unknown when using $m_{t,i}$, we only need the scalar value of $\langle \nabla f_{t-1}(\mathbf{x}_{t-1}), \mathbf{x}_t \rangle$, which can be efficiently solved via a fixed-point problem of $\langle \nabla f_{t-1}(\mathbf{x}_{t-1}), \mathbf{x}_t(z) \rangle = z$. $\mathbf{x}_t$ relies on $z$ since $m_{t,i}$ relies on $z$ and $\mathbf{x}_t$ relies on $m_{t,i}$. Interested readers can refer to Section 3.3 of Wei et al. [2016] for details.

second one measures the base learner stability, which can be canceled by optimistic algorithms [Rakhlin and Sridharan, 2013a]. We provide a self-contained analysis of the stability of optimistic online mirror descent (OMD) in Appendix E.2. At last, if the stability term of the final decisions (i.e., $\|\mathbf{x}_t - \mathbf{x}_{t-1}\|^2$) can be canceled, an $\mathcal{O}(\sqrt{V_T})$ meta regret for convex functions is achievable.

To this end, we give the positive term $\|\mathbf{x}_t - \mathbf{x}_{t-1}\|^2$ a more detailed decomposition, due to the fact that the final decision is the weighted combination of base learners' decisions (i.e., $\mathbf{x}_t = \sum_i p_{t,i}\mathbf{x}_{t,i}$). A detailed proof is deferred to Lemma 6. Specifically, it holds that

$$\|\mathbf{x}_t - \mathbf{x}_{t-1}\|^2 \lesssim \|\boldsymbol{p}_t - \boldsymbol{p}_{t-1}\|_1^2 + \sum_{i=1}^{N} p_{t,i}\|\mathbf{x}_{t,i} - \mathbf{x}_{t-1,i}\|^2, \tag{3.1}$$

where the first part represents the meta learner's stability while the second one is a weighted version of the base learners' stability. In Section 3.2, we cancel the two parts respectively.

## 3.2 Negative Terms for Cancellation

In this part, we propose negative terms to cancel the two parts of (3.1). Specifically, Section 3.2.1 analyzes the *endogenous* negative stability terms of the meta learner to handle the first part, and Section 3.2.2 proposes artificially-injected cascaded corrections to cancel the second part *exogenously*.

### 3.2.1 Endogenous Negativity: Stability Analysis of Meta Algorithms

In this part, we aim to control the meta learner's stability, measured by $\|\boldsymbol{p}_t - \boldsymbol{p}_{t-1}\|_1^2$. To this end, we leverage the two-layer meta algorithm proposed by Chen et al. [2021], where each layer runs the MSMwC algorithm [Chen et al., 2021] but with different parameter configurations. Specifically, a single MSMwC updates via the following rule:

$$\boldsymbol{p}_t = \arg\min_{\boldsymbol{p}\in\Delta_d} \{\langle \boldsymbol{m}_t, \boldsymbol{p}\rangle + \mathcal{D}_{\psi_t}(\boldsymbol{p}, \widehat{\boldsymbol{p}}_t)\}, \quad \widehat{\boldsymbol{p}}_{t+1} = \arg\min_{\boldsymbol{p}\in\Delta_d} \{\langle \boldsymbol{\ell}_t + \boldsymbol{a}_t, \boldsymbol{p}\rangle + \mathcal{D}_{\psi_t}(\boldsymbol{p}, \widehat{\boldsymbol{p}}_t)\}, \tag{3.2}$$

where $\Delta_d$ denotes a $d$-dimensional simplex, $\psi_t(\boldsymbol{p}) = \sum_{i=1}^d \eta_{t,i}^{-1} p_i \ln p_i$ is the weighted negative entropy regularizer with time-coordinate-varying step size $\eta_{t,i}$, $\mathcal{D}_{\psi_t}(\boldsymbol{p}, \boldsymbol{q}) = \psi_t(\boldsymbol{p}) - \psi_t(\boldsymbol{q}) - \langle\nabla\psi_t(\boldsymbol{q}), \boldsymbol{p} - \boldsymbol{q}\rangle$ is the induced Bregman divergence for any $\boldsymbol{p}, \boldsymbol{q} \in \Delta_d$, $\boldsymbol{m}_t$ is the optimism, $\boldsymbol{\ell}_t$ is the loss vector and $\boldsymbol{a}_t$ is a bias term.

It is worth noting that MSMwC is based on OMD, which is well-studied and proved to enjoy negative stability terms in analysis. However, the authors omitted them, which turns out to be crucial for our purpose. In Lemma 2 below, we extend Lemma 1 of Chen et al. [2021] by explicitly exhibiting the negative terms in MSMwC. The proof is deferred to Appendix B.2.

**Lemma 2.** *If* $\max_{t\in[T],i\in[d]}\{|\ell_{t,i}|, |m_{t,i}|\} \leq 1$, *then* MSMwC (3.2) *with time-invariant step sizes (i.e., $\eta_{t,i} = \eta_i$ for any $t \in [T]$)[3] enjoys the following guarantee if $\eta_i \leq 1/32$*

$$\sum_{t=1}^{T}\langle\boldsymbol{\ell}_t, \boldsymbol{p}_t\rangle - \sum_{t=1}^{T}\ell_{t,i^\star} \leq \frac{1}{\eta_{i^\star}}\ln\frac{1}{\widehat{p}_{1,i^\star}} + \sum_{i=1}^{d}\frac{\widehat{p}_{1,i}}{\eta_i} - 8\sum_{t=1}^{T}\sum_{i=1}^{d}\eta_i p_{t,i}(\ell_{t,i} - m_{t,i})^2$$

$$+16\eta_{i^\star}\sum_{t=1}^{T}(\ell_{t,i^\star} - m_{t,i^\star})^2 - 4\sum_{t=2}^{T}\|\boldsymbol{p}_t - \boldsymbol{p}_{t-1}\|_1^2.$$

The two-layer meta algorithm is constructed by MSMwC in the following way. Briefly, both layers run MSMwC, but with different parameter configurations. Specifically, the top MSMwC (indicated by MSMwC-TOP) connects with $K = \mathcal{O}(\log T)$ MSMwCs (indicated by MSMwC-MID), and each MSMwC-MID is further connected with $N$ base learners (as specified in Section 2). The specific parameter configurations will be illuminated later. The two-layer meta algorithm is provable to enjoy a regret guarantee of $\mathcal{O}(\sqrt{V_\star \ln V_\star})$ (Theorem 5 of Chen et al. [2021]), where $V_\star$ is analogous to $\sum_t(\ell_{t,i^\star} - m_{t,i^\star})^2$, but within a multi-layer context (a formal definition will be shown later). By choosing the optimism as $m_{t,i} = \langle\boldsymbol{\ell}_t, \boldsymbol{p}_t\rangle$,[4] it can recover the guarantee of ADAPT-ML-PROD,

---

[3]We only focus on the proof with fixed learning rate, since it is sufficient for our analysis.

[4]Although $\boldsymbol{p}_t$ is unknown when using $m_{t,i}$ due to (3.2), $m_{t,i} = \langle\boldsymbol{\ell}_t, \boldsymbol{p}_t\rangle$ remains the same for all dimension $i \in [d]$ and can thus omitted in the algorithm, i.e., it only suffices to update as $\boldsymbol{p}_t = \arg\min_{\boldsymbol{p}\in\Delta_d}\mathcal{D}_{\psi_t}(\boldsymbol{p}, \widehat{\boldsymbol{p}}_t)$. Interested readers can refer to Section 2.1 of Chen et al. [2021] for more details.

Table 2: Notations of the three-layer structure. The first column presents the index of layers. The rest columns illuminate the notations for the algorithms, losses, optimisms, decisions and outputs of each layer.

| Layer | Algorithm | Loss | Optimism | Decision | Output |
|---|---|---|---|---|---|
| Top (Meta) | MSMwC | $\boldsymbol{\ell}_t$ | $\boldsymbol{m}_t$ | $\boldsymbol{q}_t \in \Delta_K$ | $\mathbf{x}_t = \sum_k q_{t,k}\mathbf{x}_{t,k}$ |
| Middle (Meta) | MSMwC | $\boldsymbol{\ell}_{t,k}$ | $\boldsymbol{m}_{t,k}$ | $\boldsymbol{p}_{t,k} \in \Delta_N$ | $\mathbf{x}_{t,k} = \sum_i p_{t,k,i}\mathbf{x}_{t,k,i}$ |
| Bottom (Base) | Optimistic OMD | $f_t(\cdot)$ | $\nabla f_{t-1}(\cdot)$ | $\mathbf{x}_{t,k,i} \in \mathcal{X}$ | $\mathbf{x}_{t,k,i}$ |

although up to an $\mathcal{O}(\ln V_\star)$ factor (leading to an $\widehat{\mathcal{O}}(\sqrt{V_T})$ meta regret), but with additional negative terms in analysis. Note that single MSMwC enjoys an $\mathcal{O}(\sqrt{V_\star \ln T})$ bound, where the extra $\ln T$ factor would ruin the desired $\mathcal{O}(\log V_T)$ bound for exp-concave and strongly convex functions. This is the reason for choosing a second-layer meta algorithm, overall resulting in a *three-layer* structure.

For clarity, we summarize the notations of the three-layer structure in Table 2. Besides, previous notions need to be extended analogously. Specifically, instead of (2.2), regret is now decomposed as

$$\text{REG}_T = \left[ \sum_{t=1}^T f_t(\mathbf{x}_t) - \sum_{t=1}^T f_t(\mathbf{x}_{t,k^\star,i^\star}) \right] + \left[ \sum_{t=1}^T f_t(\mathbf{x}_{t,k^\star,i^\star}) - \min_{\mathbf{x} \in \mathcal{X}} \sum_{t=1}^T f_t(\mathbf{x}) \right].$$

The quantity $V_\star$ is defined as $V_\star \triangleq \sum_t (\ell_{t,k^\star,i^\star} - m_{t,k^\star,i^\star})^2$, where $\ell_{t,k,i} = \langle \nabla f_t(\mathbf{x}_t), \mathbf{x}_{t,k,i} \rangle$ and $m_{t,k,i} = \langle \nabla f_t(\mathbf{x}_t), \mathbf{x}_t \rangle - \langle \nabla f_{t-1}(\mathbf{x}_{t-1}), \mathbf{x}_{t-1} - \mathbf{x}_{t-1,k,i} \rangle$ follows the same sprit as Lemma 1.

At the end of this part, we explain why we choose MSMwC as the meta algorithm. Apparently, a direct try is to keep using ADAPT-ML-PROD following Zhang et al. [2022a]. However, it is still an open problem to determine whether ADAPT-ML-PROD contains negative stability terms in the analysis, which is essential to realize effective cancellation in our problem. Another try is to explore the titled exponentially weighted average (TEWA) as the meta algorithm, following another line of research [van Erven and Koolen, 2016], as introduced in Section 1.1. Unfortunately, its stability property is also unclear. Investigating the negative stability terms in these algorithms is an important open problem, but beyond the scope of this work.

### 3.2.2 Exogenous Negativity: Cascaded Correction Terms

In this part, we aim to deal with the second term in (3.1) (i.e., $\sum_i p_{t,i} \|\mathbf{x}_{t,i} - \mathbf{x}_{t-1,i}\|^2$). Inspired by the work of Zhao et al. [2021] on the gradient-variation dynamic regret in non-stationary online learning,[5] we incorporate exogenous correction terms for cancellation. Concretely, we inject *cascaded correction terms* to both top and middle layers. Specifically, the loss $\boldsymbol{\ell}_t$ and the optimism $\boldsymbol{m}_t$ of MSMwC-TOP are chosen as

$$\ell_{t,k} \triangleq \langle \nabla f_t(\mathbf{x}_t), \mathbf{x}_{t,k} \rangle + \lambda_1 \|\mathbf{x}_{t,k} - \mathbf{x}_{t-1,k}\|^2, m_{t,k} \triangleq \langle \widehat{\boldsymbol{m}}_{t,k}, \boldsymbol{p}_{t,k} \rangle + \lambda_1 \|\mathbf{x}_{t,k} - \mathbf{x}_{t-1,k}\|^2. \quad (3.3)$$

The loss $\boldsymbol{\ell}_{t,k}$ and optimism $\boldsymbol{m}_{t,k}$ of MSMwC-MID are chosen similarly as

$$\ell_{t,k,i} \triangleq \langle \nabla f_t(\mathbf{x}_t), \mathbf{x}_{t,k,i} \rangle + \lambda_2 \|\mathbf{x}_{t,k,i} - \mathbf{x}_{t-1,k,i}\|^2, m_{t,k,i} \triangleq \widehat{m}_{t,k,i} + \lambda_2 \|\mathbf{x}_{t,k,i} - \mathbf{x}_{t-1,k,i}\|^2, \quad (3.4)$$

where $\widehat{\boldsymbol{m}}_{t,k} \triangleq (\widehat{m}_{t,k,1}, \ldots, \widehat{m}_{t,k,N})$ and $\widehat{m}_{t,k,i} = \langle \nabla f_t(\mathbf{x}_t), \mathbf{x}_t \rangle - \langle \nabla f_{t-1}(\mathbf{x}_{t-1}), \mathbf{x}_{t-1} - \mathbf{x}_{t-1,k,i} \rangle$ denotes our optimism, which uses the same idea as Lemma 1 in Section 3.1.

To see how the correction term works, consider a simpler problem with regret $\sum_t \langle \boldsymbol{\ell}_t, \boldsymbol{q}_t \rangle - \sum_t \ell_{t,k^\star}$. If we instead optimize the corrected loss $\boldsymbol{\ell}_t + \boldsymbol{b}_t$ and obtain a regret bound of $R_T$, then moving the correction terms to the right-hand side, the original regret is at most $R_T - \sum_t \sum_k q_{t,k} b_{t,k} + \sum_t b_{t,k^\star}$, where the negative term of $-\sum_t \sum_k q_{t,k} b_{t,k}$ can be leveraged for cancellation. Meanwhile, the algorithm is required to handle an extra term of $\sum_t b_{t,k^\star}$, which only relies on the $k^\star$-th dimension. In the next part, we will discuss about how cascaded corrections and negative stability terms of meta algorithms realize effective collaboration for cancellation in the multi-layer online ensemble.

It is noteworthy that our work is the *first* to introduce correction mechanisms to universal online learning, whereas prior works use it for different purposes, such as non-stationary online learn-

---

[5]This work proposes an improved dynamic regret minimization algorithm compared to its conference version [Zhao et al., 2020], which introduces the correction terms to the meta-base online ensemble structure and thus improves the gradient query complexity from $\mathcal{O}(\log T)$ to 1 within each round.

---

**Algorithm 1** Universal OCO with Gradient-variation Guarantees

---

**Input:** Curvature coefficient pool $\mathcal{H}$, MSMWC-MID number $K$, base learner number $N$

1: **Initialize**: Top layer: $\mathcal{A}^{\text{top}}$ — MSMWC-TOP with $\eta_k = (C_0 \cdot 2^k)^{-1}$ and $\widehat{q}_{1,k} = \eta_k^2 / \sum_{k=1}^K \eta_k^2$
                Middle layer: $\{\mathcal{A}_k^{\text{mid}}\}_{k \in [K]}$ — MSMWC-MID with step size $2\eta_k$ and $\widehat{p}_{1,k,i} = {}^1/N$
                Bottom layer: $\{\mathcal{B}_{k,i}\}_{k \in [K], i \in [N]}$ — base learners as specified in Section 2

2: **for** $t = 1$ **to** $T$ **do**

3:     Receive $\mathbf{x}_{t,k,i}$ from $\mathcal{B}_{k,i}$, obtain $\mathbf{x}_{t,k} = \sum_i p_{t,k,i} \mathbf{x}_{t,k,i}$ and submit $\mathbf{x}_t = \sum_k q_{t,k} \mathbf{x}_{t,k}$

4:     Suffer $f_t(\mathbf{x}_t)$ and observe the gradient information $\nabla f_t(\cdot)$

5:     Construct $(\boldsymbol{\ell}_t, \boldsymbol{m}_t)$ (3.3) for $\mathcal{A}^{\text{top}}$ and $(\boldsymbol{\ell}_{t,k}, \boldsymbol{m}_{t,k})$ (3.4) for $\mathcal{A}_k^{\text{mid}}$

6:     $\mathcal{A}^{\text{top}}$ updates to $\boldsymbol{q}_{t+1}$ and $\mathcal{A}_k^{\text{mid}}$ updates to $\boldsymbol{p}_{t+1,k}$

7:     **Multi-gradient feedback model:**

8:         Send gradient $\nabla f_t(\cdot)$ to $\mathcal{B}_{k,i}$ for update               $\triangleright \mathcal{O}(\log^2 T)$ gradient queries

9:     **One-gradient feedback model:**

10:        Construct surrogates $h_{t,i}^{\text{sc}}(\cdot), h_{t,i}^{\text{exp}}(\cdot), h_{t,i}^{\text{c}}(\mathbf{x})$ using only $\nabla f_t(\mathbf{x}_t)$

11:        Send the surrogate functions to $\mathcal{B}_{k,i}$ for update     $\triangleright$ Only *one* gradient query

12: **end for**

---

ing [Zhao et al., 2021] and the multi-scale expert problem [Chen et al., 2021]. Distinctively, different from the conventional two-layer algorithmic frameworks seen in prior studies, deploying this technique to a three-layer structure necessitates a comprehensive use and extensive adaptation of it.

### 3.3 Overall Algorithm: A Multi-layer Online Ensemble Structure

In this part, we conclude our *three-layer online ensemble* approach in Algorithm 1. In Line 3, the decisions are aggregated from bottom to top for the final output. In Line 4, the learner suffers the loss of the decision, and the environments return the gradient information of the loss function. In Line 5, the algorithm constructs the surrogate losses and optimisms for the two-layer meta learner. In Line 6-11, the update is conducted from top to bottom. Note that in Line 8, our algorithm requires multiple, concretely $\mathcal{O}(\log^2 T)$, gradient queries per round since it needs to query $\nabla f_t(\mathbf{x}_{t,k,i})$ for each base learner, making it inefficient when the gradient evaluation is costly. To this end, in Section 4, we improve the algorithm's gradient query complexity to 1 per round, corresponding to Line 10-11, via a novel regret decomposition and carefully designed surrogate loss functions.

In Figure 1, we illustrate the detailed procedure of the collaboration in our multi-layer online ensemble approach. Specifically, we aim to deal with the positive term of $\|\mathbf{x}_t - \mathbf{x}_{t-1}\|^2$, which stems from the universal optimism design proposed in Section 3.1. Since $\mathbf{x}_t = \sum_k q_{t,k} \mathbf{x}_{t,k}$, the positive term can be further decomposed into two parts: $\|\boldsymbol{q}_t - \boldsymbol{q}_{t-1}\|_1^2$ and $\sum_k q_{t,k} \|\mathbf{x}_{t,k} - \mathbf{x}_{t-1,k}\|^2$, which can be canceled by the negative and correction terms in MSMWC-TOP, respectively. Note that the correction comes at the expense of an extra term of $\|\mathbf{x}_{t,k^\star} - \mathbf{x}_{t-1,k^\star}\|^2$, which can be decomposed similarly into two parts: $\|\boldsymbol{p}_{t,k^\star} - \boldsymbol{p}_{t-1,k^\star}\|_1^2$ and $\sum_i p_{t,k^\star,i} \|\mathbf{x}_{t,k^\star,i} - \mathbf{x}_{t-1,k^\star,i}\|^2$ because of $\mathbf{x}_{t,k} = \sum_i p_{t,k,i} \mathbf{x}_{t,k,i}$. The negative term and corrections in the middle layer (specifically, the $k^\star$-th MSMWC-MID) can be leveraged for cancellation. This correction finally generate $\|\mathbf{x}_{t,k^\star,i^\star} - \mathbf{x}_{t-1,k^\star,i^\star}\|^2$, which can be handled by choosing optimistic OMD as base algorithms.

As a final remark, although it is possible to treat the two-layer meta algorithm as a single layer, its analysis will become much more complicated than that in one layer and is unsuitable for extending to more layers. In contrast, our layer-by-layer analytical approach paves a systematic and principled way for analyzing the dynamics of the online ensemble framework with even more layers.

### 3.4 Universal Regret Guarantees

In this part, we conclude our main theoretical result and provide several implications and applications to validate its importance and practical potential. Theorem 1 summarizes the main result, universal regret guarantees in terms of the gradient variation. The proof is deferred to Appendix B.4.

**Theorem 1.** *Under Assumptions 1 and 2, Algorithm 1 obtains* $\mathcal{O}(\log V_T)$, $\mathcal{O}(d \log V_T)$ *and* $\widehat{\mathcal{O}}(\sqrt{V_T})$ *regret bounds for strongly convex, exp-concave and convex functions, respectively.*

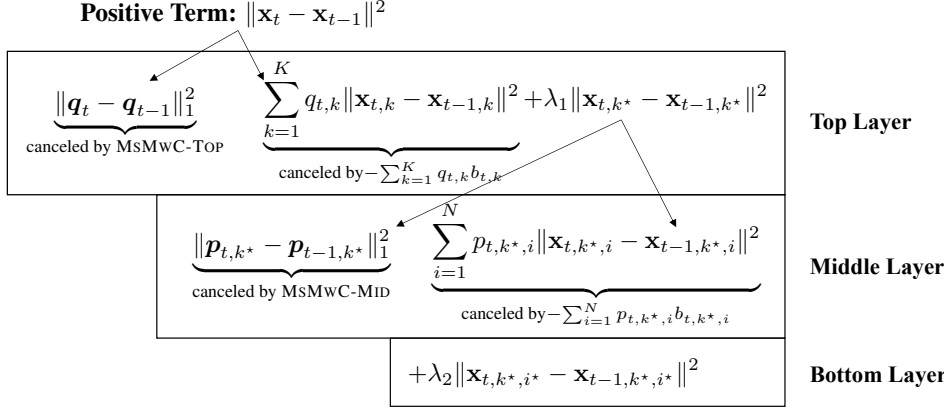

Figure 1: Decomposition of the positive term $\|\mathbf{x}_t - \mathbf{x}_{t-1}\|^2$ and how it is handled by the multi-layer online ensemble via endogenous negativity from meta algorithm and exogenous negativity from cascaded corrections.

Theorem 1 improves the results of Zhang et al. [2022a] by not only maintaining the optimal rates for strongly convex and exp-concave functions but also taking advantage of the small gradient variation for convex functions when $V_T \ll F_T$. For example, if $f_1 = \ldots = f_T = f$ and $\min_{\mathbf{x} \in \mathcal{X}} f(\mathbf{x}) = 1$, our bound is much better since $V_T = 0$ while $F_T = T$. Moreover, Algorithm 1 also provably achieves universal *small-loss* guarantees *without* any algorithmic modification and thus safeguards the case when $F_T \leq V_T$. We conclude the result below and provide the proof in Appendix B.5.

**Corollary 1.** *Under Assumptions 1 and 2, if $f_t(\cdot) \geq 0$, Algorithm 1 obtains $\mathcal{O}(\log F_T)$, $\mathcal{O}(d \log F_T)$ and $\widehat{\mathcal{O}}(\sqrt{F_T})$ regret bounds for strongly convex, exp-concave and convex functions.*

Due to the connection of the gradient-variation with *stochastic and adversarial OCO* [Sachs et al., 2022] and *game theory* [Syrgkanis et al., 2015], our results can be immediately applied and achieve best known universal guarantees therein. Due to page limits, we defer applications to Appendix A.

## 4 Improved Gradient Query Complexity

Though achieving favorable theoretical guarantees in Section 3, one caveat is that our algorithm requires $\mathcal{O}(\log^2 T)$ gradient queries per round since it needs to query $\nabla f_t(\mathbf{x}_{t,k,i})$ for all $k \in [K], i \in [N]$, making it computational-inefficient when the gradient evaluation is costly, e.g., in nuclear norm optimization [Ji and Ye, 2009] and mini-batch optimization [Li et al., 2014]. The same concern also appears in the approach of Zhang et al. [2022a], who provided small-loss and worst-case regret guarantees for universal online learning. By contrast, traditional algorithms such as OGD typically work under the *one-gradient* feedback setup, namely, they only require one gradient $\nabla f_t(\mathbf{x}_t)$ for the update. In light of this, it is natural to ask whether there is a universal algorithm that can maintain the desired regret guarantees while using only one gradient query per round.

We answer the question affirmatively by reducing the gradient query complexity to 1 per round. To describe the high-level idea, we first consider the case of *known* $\lambda$-strong convexity within a two-layer structure, e.g., adaptive regret minimization for strongly convex functions [Wang et al., 2018]. The regret can be upper-bounded as $\text{REG}_T \leq [\sum_t h_t(\mathbf{x}_t) - \sum_t h_t(\mathbf{x}_{t,i^\star})] + [\sum_t h_t(\mathbf{x}_{t,i^\star}) - \sum_t h_t(\mathbf{x}^\star)]$, where $\mathbf{x}^\star \in \arg\min_{\mathbf{x} \in \mathcal{X}} \sum_t f_t(\mathbf{x})$ and $h_t(\mathbf{x}) \triangleq \langle \nabla f_t(\mathbf{x}_t), \mathbf{x} \rangle + \lambda \|\mathbf{x} - \mathbf{x}_t\|^2/2$ is a second-order surrogate of the original function $f_t$. This yields a *homogeneous* surrogate for both meta and base algorithms — the meta algorithm uses the same evaluation function across all the base learners, who in turn use this function to update their base-level decisions.

In the universal online learning (i.e., *unknown* $\lambda$), a natural adaptation would be a *heterogeneous* surrogate $h_{t,i}(\mathbf{x}) \triangleq \langle \nabla f_t(\mathbf{x}_t), \mathbf{x} \rangle + \lambda_i \|\mathbf{x} - \mathbf{x}_t\|^2/2$ for the $i$-th base learner, where $\lambda_i$ is a guess of the true curvature $\lambda$ from the candidate pool $\mathcal{H}$ (2.1). Consequently, the regret decomposition with respect to this surrogate becomes $\text{REG}_T \leq [\sum_t h_{t,i^\star}(\mathbf{x}_t) - \sum_t h_{t,i^\star}(\mathbf{x}_{t,i^\star})] + [\sum_t h_{t,i^\star}(\mathbf{x}_{t,i^\star}) - \sum_t h_{t,i^\star}(\mathbf{x}^\star)]$, where $i^\star$ denotes the index of the best base learner whose strong convexity coefficient satisfies $\lambda_{i^\star} \leq \lambda \leq 2\lambda_{i^\star}$. This admits *heterogeneous* surrogates for both meta and base regret, which

necessitates designing expert-tracking algorithms with heterogeneous inputs for different experts as required in MetaGrad [van Erven and Koolen, 2016], which is not flexible enough.

To this end, we propose a novel regret decomposition that admits *homogeneous* surrogates for the meta regret, making our algorithm as flexible as Zhang et al. [2022a], and *heterogeneous* surrogate for the base regret, making it as efficient as van Erven and Koolen [2016]. Specifically, we remain the heterogeneous surrogates $h_{t,i}(\mathbf{x})$ defined above for the base regret. For the meta regret, we define a homogeneous linear surrogate $\ell_t(\mathbf{x}) \triangleq \langle \nabla f_t(\mathbf{x}_t), \mathbf{x} \rangle$ such that the meta regret is bounded by $\sum_t \ell_t(\mathbf{x}_t) - \sum_t \ell_t(\mathbf{x}_{t,i^\star}) - \lambda_{i^\star} \sum_t \|\mathbf{x}_t - \mathbf{x}_{t,i^\star}\|^2 / 2$. As long as we can obtain a second-order bound for the regret defined on this surrogate loss, i.e., $\sum_t \ell_t(\mathbf{x}_t) - \sum_t \ell_t(\mathbf{x}_{t,i^\star})$, it can be canceled by the negative term from strong convexity. Overall, we decompose the regret in the following way:

$$\mathrm{REG}_T \leq \left[ \sum_{t=1}^T \ell_t(\mathbf{x}_t) - \sum_{t=1}^T \ell_t(\mathbf{x}_{t,i^\star}) - \frac{\lambda_{i^\star}}{2} \sum_{t=1}^T \|\mathbf{x}_t - \mathbf{x}_{t,i^\star}\|^2 \right] + \left[ \sum_{t=1}^T h_{t,i^\star}(\mathbf{x}_{t,i^\star}) - \sum_{t=1}^T h_{t,i^\star}(\mathbf{x}^\star) \right].$$

For clarity, we denote this surrogate for strongly convex functions by $h_{t,i}^{\mathrm{sc}}(\mathbf{x})$. Similarly, we define the surrogates $h_{t,i}^{\exp}(\mathbf{x}) \triangleq \langle \nabla f_t(\mathbf{x}_t), \mathbf{x} \rangle + \alpha_i \langle \nabla f_t(\mathbf{x}_t), \mathbf{x} - \mathbf{x}_t \rangle^2 / 2$ and $h_{t,i}^{\mathrm{c}}(\mathbf{x}) \triangleq \langle \nabla f_t(\mathbf{x}_t), \mathbf{x} \rangle$ for $\alpha$-exp-concave and convex functions, respectively. These surrogates require only *one* gradient $\nabla f_t(\mathbf{x}_t)$ within each round, thus successfully reducing the gradient query complexity.

Note that the base regret optimization requires controlling the algorithmic stability, because the empirical gradient variation $\nabla h_{t,i}^{\mathrm{sc}}(\mathbf{x}_{t,i}) - \nabla h_{t-1,i}^{\mathrm{sc}}(\mathbf{x}_{t-1,i}) = \nabla f_t(\mathbf{x}_t) + \lambda_i(\mathbf{x}_{t,i} - \mathbf{x}_t) - \nabla f_{t-1}(\mathbf{x}_{t-1}) - \lambda_i(\mathbf{x}_{t-1,i} - \mathbf{x}_{t-1})$ not only contains the desired gradient variation, but also includes the positive stability terms of base and final decisions. Fortunately, as discussed earlier, these stability terms can be effectively addressed through our cancellation mechanism within the multi-layer online ensemble. This stands in contrast to previous two-layer algorithms with worst-case regret bounds [Zhang et al., 2018, Wang et al., 2018], where the algorithmic stability is not examined.

The efficient version is concluded in Algorithm 1 with Line 10-11, which uses only one gradient $\nabla f_t(\mathbf{x}_t)$. The only algorithmic modification is that base learners update on the carefully designed surrogate functions, not the original one. We provide the regret guarantee below, which achieves the same guarantees as Theorem 1 with only one gradient per round. The proof is in Appendix C.2.

**Theorem 2.** *Under Assumptions 1 and 2, efficient Algorithm 1 enjoys $\mathcal{O}(\log V_T)$, $\mathcal{O}(d \log V_T)$ and $\widehat{\mathcal{O}}(\sqrt{V_T})$ for strongly convex, exp-concave and convex functions, using only one gradient per round.*

As a byproduct, we show that this idea can be used to recover the optimal worst-case universal guarantees using one gradient with a simpler approach and analysis, with proof in Appendix C.1.

**Proposition 1.** *Under Assumption 1, using the above surrogate loss functions for base learners, and running* ADAPT-ML-PROD *as the meta learner guarantees $\mathcal{O}(\log T)$, $\mathcal{O}(d \log T)$ and $\mathcal{O}(\sqrt{T})$ regret bounds for strongly convex, exp-concave and convex functions, using one gradient per round.*

## 5 Conclusion

In this paper, we obtain universal gradient-variation guarantees via a multi-layer online ensemble approach. We first propose a novel optimism design to unify various kinds of functions. Then we analyze the negative terms of the meta algorithm MSMWC and inject cascaded correction terms to improve the algorithmic stability to realize effective cancellations in the multi-layer structure. Furthermore, we provide a novel regret decomposition combined with carefully designed surrogate functions to achieve one gradient query per round. Finally, we deploy the our approach into two applications, including the stochastically extended adversarial (SEA) model and two-player zero-sum games, to validate its effectiveness, and obtain best known universal guarantees therein. Due to page limits, the applications are deferred to Appendix A. Two byproducts rise in our work. The first one is negative stability terms in the analysis of MSMWC. And the second one contains a simple approach and analysis for the optimal worst-case universal guarantees, using one gradient per round.

An important open problem lies in optimality and efficiency. In the convex case, our results still exhibit an $\mathcal{O}(\log V_T)$ gap from the optimal $\mathcal{O}(\sqrt{V_T})$ result. Moreover, our algorithm necessitates $\mathcal{O}(\log^2 T)$ base learners, as opposed to $\mathcal{O}(\log T)$ base learners in two-layer structures. Whether it is possible to achieve the optimal results for all kinds of functions (strongly convex, exp-concave, convex) using a two-layer algorithm remains as an important problem for future investigation.

## Acknowledgements

This research was supported by National Key R&D Program of China (2022ZD0114800), NSFC (62206125, 61921006), and JiangsuSF (BK20220776). The authors would thank Reviewer #ZhYs of OpenReview for the thorough review and insightful suggestions.

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

# A Applications

In this section, we validate the importance and the practical potential of our approach by applying it in two problems, including stochastically extended adversarial (SEA) model in Appendix A.1 and two-player zero-sum games in Appendix A.2.

## A.1 Application I: Stochastically Extended Adversarial Model

Stochastically extended adversarial (SEA) model [Sachs et al., 2022] serves as an interpolation between stochastic and adversarial online convex optimization, where the environments choose the loss function $f_t$ from a distribution $\mathcal{D}_t$ (i.e., $f_t \sim \mathcal{D}_t$). They proposed cumulative stochastic variance $\sigma_{1:T}^2$, the gap between $f_t$ and its expected version $F_t(\cdot) \triangleq \mathbb{E}_{f_t \sim \mathcal{D}_t}[f_t(\cdot)]$, and cumulative adversarial variation $\Sigma_{1:T}^2$, the gap between $F_t(\cdot)$ and $F_{t-1}(\cdot)$, to bound the expected regret. Formally,

$$\sigma_{1:T}^2 \triangleq \sum_{t=1}^{T} \max_{\mathbf{x} \in \mathcal{X}} \mathbb{E}_{f_t \sim \mathcal{D}_t} \left[ \|\nabla f_t(\mathbf{x}) - \nabla F_t(\mathbf{x})\|^2 \right], \Sigma_{1:T}^2 \triangleq \mathbb{E} \left[ \sum_{t=2}^{T} \sup_{\mathbf{x} \in \mathcal{X}} \|\nabla F_t(\mathbf{x}) - \nabla F_{t-1}(\mathbf{x})\|^2 \right].$$

By specializing different $\sigma_{1:T}^2$ and $\Sigma_{1:T}^2$, the SEA model can recover the adversarial and stochastic OCO setup, respectively. Specifically, setting $\sigma_{1:T}^2 = 0$ recovers the adversarial setup, where $\Sigma_{1:T}^2$ equals to the gradient variation $V_T$. Besides, choosing $\Sigma_{1:T}^2 = 0$ recovers the stochastic setup, where $\sigma_{1:T}^2$ stands for the variance in stochastic optimization.

For smooth expected function $F_t(\cdot)$, Sachs et al. [2022] obtained an optimal $\mathcal{O}(\sqrt{\sigma_{1:T}^2 + \Sigma_{1:T}^2})$ expected regret for convex functions and an $\mathcal{O}((\sigma_{\max}^2 + \Sigma_{\max}^2) \log T)$ regret for strongly convex functions, where $\sigma_{\max}^2 \triangleq \max_{t \in [T]} \max_{\mathbf{x} \in \mathcal{X}} \mathbb{E}_{f_t \sim \mathcal{D}_t}[\|\nabla f_t(\mathbf{x}) - \nabla F_t(\mathbf{x})\|^2]$ and $\Sigma_{\max}^2 \triangleq \max_{t \in [T]} \sup_{\mathbf{x} \in \mathcal{X}} \|\nabla F_t(\mathbf{x}) - \nabla F_{t-1}(\mathbf{x})\|^2$. Later, the results are improved by Chen et al. [2023b], where the authors achieved a better $\mathcal{O}((\sigma_{\max}^2 + \Sigma_{\max}^2) \log(\sigma_{1:T}^2 + \Sigma_{1:T}^2))$ regret[6] for strongly convex functions and a new $\mathcal{O}(d \log(\sigma_{1:T}^2 + \Sigma_{1:T}^2))$ regret for exp-concave functions, while maintaining the optimal guarantee for convex functions. Their key observation is that the gradient variation in optimistic algorithms encodes the information of the cumulative stochastic variance $\sigma_{1:T}^2$ and cumulative adversarial variation $\Sigma_{1:T}^2$, restated in Lemma 5.

A major open problem remains in the work of Chen et al. [2023b] that whether it is possible to get rid of different parameter configurations and obtain universal guarantees in the problem. In the following, we show that our approach can be directly applied and achieves almost the same guarantees as those in Chen et al. [2023b], up to a logarithmic factor in leading term. Notably, our algorithm does not require different parameter setups, thus resolving the open problem proposed by the authors. We conclude our results in Theorem 3 and the proof can be found in Appendix D.1.

**Theorem 3.** *Under the same assumptions as Chen et al. [2023b], the efficient version of Algorithm 1 obtains $\mathcal{O}((\sigma_{\max}^2 + \Sigma_{\max}^2) \log(\sigma_{1:T}^2 + \Sigma_{1:T}^2))$ regret for strongly convex functions, $\mathcal{O}(d \log(\sigma_{1:T}^2 + \Sigma_{1:T}^2))$ regret for exp-concave functions and $\widehat{\mathcal{O}}(\sqrt{\sigma_{1:T}^2 + \Sigma_{1:T}^2})$ regret for convex functions, where the $\widehat{\mathcal{O}}(\cdot)$-notation omits logarithmic factors in leading terms.*

Note that Sachs et al. [2023] also considered the problem of universal learning and obtained an $\mathcal{O}(\sqrt{T \log T})$ regret for convex functions and an $\mathcal{O}((\sigma_{\max}^2 + \Sigma_{\max}^2 + D^2 L^2) \log^2 T)$ regret for strongly convex functions simultaneously. We conclude the existing results in Table 3. Our results are better than theirs in two aspects: *(i)* for strongly convex and convex functions, our guarantees are adaptive with the problem-dependent quantities $\sigma_{1:T}$ and $\Sigma_{1:T}$ while theirs depends on the time horizon $T$; and *(ii)* our algorithm achieves an additional guarantee for exp-concave functions.

## A.2 Application II: Two-player Zero-sum Games

Multi-player online games [Cesa-Bianchi and Lugosi, 2006] is a versatile model that depicts the interaction of multiple players over time. Since each player is facing similar players like herself, the theoretical guarantees, e.g., the summation of all players' regret, can be better than the minimax optimal $\sqrt{T}$ bound in adversarial environments, thus achieving fast rates.

---

[6]This bound improves the result of $\mathcal{O}(\min\{G^2 \log(\sigma_{1:T}^2 + \Sigma_{1:T}^2), (\sigma_{\max}^2 + \Sigma_{\max}^2) \log T\})$ in their conference version [Chen et al., 2023a].

Table 3: Comparisons of our results with existing ones. The second column presents the regret bounds for various kinds of loss functions, where $\sigma_{1:T}^2$ and $\Sigma_{1:T}^2$ are the cumulative stochastic variance and adversarial variation, which are at most $\mathcal{O}(T)$ and can be much smaller in benign environments. The $\widehat{\mathcal{O}}(\cdot)$-notation omits logarithmic factors on leading terms. The third column indicates whether the results for different kinds of functions are achieved by a single algorithm.

| Works | Regret Bounds | | | Single Algorithm? |
|---|---|---|---|---|
| | Strongly Convex | Exp-concave | Convex | |
| Sachs et al. [2022] | $\mathcal{O}((\sigma_{\max}^2 + \Sigma_{\max}^2)\log T)$ | ✗ | $\mathcal{O}(\sqrt{\sigma_{1:T}^2 + \Sigma_{1:T}^2})$ | No |
| Chen et al. [2023b] | $\mathcal{O}((\sigma_{\max}^2 + \Sigma_{\max}^2)\log(\sigma_{1:T}^2 + \Sigma_{1:T}^2))$ | $\mathcal{O}(d\log(\sigma_{1:T}^2 + \Sigma_{1:T}^2))$ | $\mathcal{O}(\sqrt{\sigma_{1:T}^2 + \Sigma_{1:T}^2})$ | No |
| Sachs et al. [2023] | $\mathcal{O}((\sigma_{\max}^2 + \Sigma_{\max}^2 + D^2L^2)\log^2 T)$ | ✗ | $\widehat{\mathcal{O}}(\sqrt{T})$ | Yes |
| **Ours** | $\mathcal{O}((\sigma_{\max}^2 + \Sigma_{\max}^2)\log(\sigma_{1:T}^2 + \Sigma_{1:T}^2))$ | $\mathcal{O}(d\log(\sigma_{1:T}^2 + \Sigma_{1:T}^2))$ | $\widehat{\mathcal{O}}(\sqrt{\sigma_{1:T}^2 + \Sigma_{1:T}^2})$ | Yes |

Table 4: Comparisons of our results with existing ones. In the honest case, the results are measured by the summation of all players' regret and in the dishonest case, the results are in terms of the individual regret of each player. Bilinear and strongly convex-concave games are considered inside each case. • denotes the best result in each case (row).

| | Games | Syrgkanis et al. [2015] | Zhang et al. [2022a] | Ours |
|---|---|---|---|---|
| **Honest** | bilinear | $\mathcal{O}(1)$ • | $\mathcal{O}(\sqrt{T})$ | $\mathcal{O}(1)$ • |
| | strongly convex-concave | $\mathcal{O}(1)$ • | $\mathcal{O}(\log T)$ | $\mathcal{O}(1)$ • |
| **Dishonest** | bilinear | $\mathcal{O}(\sqrt{T})$ • | $\mathcal{O}(\sqrt{T})$ • | $\widehat{\mathcal{O}}(\sqrt{T})$ • |
| | strongly convex-concave | $\mathcal{O}(\sqrt{T})$ | $\mathcal{O}(\log T)$ • | $\mathcal{O}(\log T)$ • |

The pioneering works of Rakhlin and Sridharan [2013b], Syrgkanis et al. [2015] investigated optimistic algorithms in multi-player online games and illuminated the importance of the gradient variation. Specifically, Syrgkanis et al. [2015] proved that when each player runs an optimistic algorithm (optimistic OMD or optimistic follow the regularized leader), the summation of the regret, which serves as an upper bound for some performance measures in games, can be bounded by $\mathcal{O}(1)$. The above results assume that the players are honest, i.e., they agree to run the same algorithm distributedly. In the dishonest case, where there exist players who do not follow the agreed protocol, the guarantees will degenerate to the minimax result of $\mathcal{O}(\sqrt{T})$.

In this part, we consider the simple two-player zero-sum games as an example to validate the effectiveness of our proposed algorithm. The game can be formulated as a min-max optimization problem of $\min_{\mathbf{x}\in\mathcal{X}} \max_{\mathbf{y}\in\mathcal{Y}} f(\mathbf{x}, \mathbf{y})$. We consider the case that the game is either bilinear (i.e., $f(\mathbf{x}, \mathbf{y}) = \mathbf{x}^\top A\mathbf{y}$), or strongly convex-concave (i.e., $f(\mathbf{x}, \mathbf{y})$ is $\lambda$-strongly convex in $\mathbf{x}$ and $\lambda$-strongly concave in $\mathbf{y}$). Our algorithm can guarantee the regret summation in the honest case and the individual regret of each player in the dishonest case, without knowing the type of games in advance. We conclude our results in Theorem 4, and the proof can be found in Appendix D.2.

**Theorem 4.** *Under Assumptions 1 and 2, for bilinear and strongly convex-concave games, the efficient version of Algorithm 1 enjoys $\mathcal{O}(1)$ regret summation in the honest case, $\widehat{\mathcal{O}}(\sqrt{T})$ and $\mathcal{O}(\log T)$ bounds in the dishonest case, where the $\widehat{\mathcal{O}}(\cdot)$-notation omits logarithmic factors in leading terms.*

Table 4 compares our approach with Syrgkanis et al. [2015], Zhang et al. [2022a]. Specifically, ours is better than Syrgkanis et al. [2015] in the strongly convex-concave games in the dishonest case due to its universality, and better than Zhang et al. [2022a] in the honest case since our approach enjoys gradient-variation bounds that are essential in achieving fast rates for regret summation.

# B  Proofs for Section 3

In this section, we provide proofs for Section 3, including Lemma 1, Lemma 2 and Lemma 3. For simplicity, we introduce the following notations denoting the stability of the final and intermediate decisions of the algorithm. Specifically, for any $k \in [K], i \in [N]$, we define

$$S_T^{\mathbf{x}} \triangleq \sum_{t=2}^{T} \|\mathbf{x}_t - \mathbf{x}_{t-1}\|^2, \ S_{T,k}^{\mathbf{x}} \triangleq \sum_{t=2}^{T} \|\mathbf{x}_{t,k} - \mathbf{x}_{t-1,k}\|^2, \ S_{T,k,i}^{\mathbf{x}} \triangleq \sum_{t=2}^{T} \|\mathbf{x}_{t,k,i} - \mathbf{x}_{t-1,k,i}\|^2,$$

$$S_T^{\boldsymbol{q}} \triangleq \sum_{t=2}^{T} \|\boldsymbol{q}_t - \boldsymbol{q}_{t-1}\|_1^2, \text{ and } S_{T,k}^{\boldsymbol{p}} \triangleq \sum_{t=2}^{T} \|\boldsymbol{p}_{t,k} - \boldsymbol{p}_{t-1,k}\|_1^2. \tag{B.1}$$

## B.1 Proof of Lemma 1

*Proof.* For simplicity, we define $\mathbf{g}_t \triangleq \nabla f_t(\mathbf{x}_t)$. For exp-concave functions, we have

$$\sum_{t=1}^{T} (r_{t,i^\star} - m_{t,i^\star})^2 = \sum_{t=1}^{T} (\langle \mathbf{g}_t, \mathbf{x}_t - \mathbf{x}_{t,i^\star} \rangle - \langle \mathbf{g}_{t-1}, \mathbf{x}_{t-1} - \mathbf{x}_{t-1,i^\star} \rangle)^2$$

$$\leq 2 \sum_{t=1}^{T} \langle \mathbf{g}_t, \mathbf{x}_t - \mathbf{x}_{t,i^\star} \rangle^2 + 2 \sum_{t=1}^{T} \langle \mathbf{g}_{t-1}, \mathbf{x}_{t-1} - \mathbf{x}_{t-1,i^\star} \rangle^2$$

$$\leq 4 \sum_{t=1}^{T} \langle \mathbf{g}_t, \mathbf{x}_t - \mathbf{x}_{t,i^\star} \rangle^2 + \mathcal{O}(1).$$

For convex function, it holds that

$$\sum_{t=1}^{T} (r_{t,i^\star} - m_{t,i^\star})^2 = \sum_{t=1}^{T} (\langle \mathbf{g}_t, \mathbf{x}_t - \mathbf{x}_{t,i^\star} \rangle - \langle \mathbf{g}_{t-1}, \mathbf{x}_{t-1} - \mathbf{x}_{t-1,i^\star} \rangle)^2$$

$$\leq 2 \sum_{t=1}^{T} \langle \mathbf{g}_t - \mathbf{g}_{t-1}, \mathbf{x}_t - \mathbf{x}_{t,i^\star} \rangle^2 + 2 \sum_{t=1}^{T} \langle \mathbf{g}_{t-1}, \mathbf{x}_t - \mathbf{x}_{t-1} + \mathbf{x}_{t,i^\star} - \mathbf{x}_{t-1,i^\star} \rangle^2$$

$$\leq 2D^2 \sum_{t=1}^{T} \|\mathbf{g}_t - \mathbf{g}_{t-1}\|^2 + 2G^2 \sum_{t=1}^{T} \|\mathbf{x}_t - \mathbf{x}_{t-1} + \mathbf{x}_{t,i^\star} - \mathbf{x}_{t-1,i^\star}\|^2 \qquad \text{(by Assumption 1)}$$

$$\leq 4D^2 V_T + 4(D^2 L^2 + G^2) \sum_{t=2}^{T} \|\mathbf{x}_t - \mathbf{x}_{t-1}\|^2 + 4G^2 \sum_{t=2}^{T} \|\mathbf{x}_{t,i^\star} - \mathbf{x}_{t-1,i^\star}\|^2,$$

where the last step is due to the definition of the gradient variation, finishing the proof. $\square$

## B.2 Proof of Lemma 2

In this part, we analyze the negative stability terms in the MsMwC algorithm [Chen et al., 2021]. For self-containedness, we restate the update rule of MsMwC in the following:

$$\boldsymbol{p}_t = \underset{\boldsymbol{p} \in \Delta_d}{\arg\min} \ \{\langle \boldsymbol{m}_t, \boldsymbol{p} \rangle + \mathcal{D}_{\psi_t}(\boldsymbol{p}, \widehat{\boldsymbol{p}}_t)\}, \quad \widehat{\boldsymbol{p}}_{t+1} = \underset{\boldsymbol{p} \in \Delta_d}{\arg\min} \ \{\langle \boldsymbol{\ell}_t + \boldsymbol{a}_t, \boldsymbol{p} \rangle + \mathcal{D}_{\psi_t}(\boldsymbol{p}, \widehat{\boldsymbol{p}}_t)\},$$

where the bias term $a_{t,i} = 16\eta_{t,i}(\ell_{t,i} - m_{t,i})^2$. Below, we give a detailed proof of Lemma 2, following a similar logic flow as Lemma 1 of Chen et al. [2021], while illustrating the negative stability terms. Moreover, for generality, we investigate a more general setting of an arbitrary comparator $\boldsymbol{u} \in \Delta_d$ and changing step sizes $\eta_{t,i}$. This was done hoping that the negative stability term analysis would be comprehensive enough for readers interested solely in the MsMwC algorithm.

*Proof of Lemma 2.* To begin with, the regret with correction can be analyzed as follows:

$$\sum_{t=1}^{T} \langle \boldsymbol{\ell}_t + \boldsymbol{a}_t, \boldsymbol{p}_t - \boldsymbol{e}_{i^\star} \rangle \leq \sum_{t=1}^{T} \left( \mathcal{D}_{\psi_t}(\boldsymbol{u}, \widehat{\boldsymbol{p}}_t) - \mathcal{D}_{\psi_t}(\boldsymbol{u}, \widehat{\boldsymbol{p}}_{t+1}) \right) + \sum_{t=1}^{T} \langle \boldsymbol{\ell}_t + \boldsymbol{a}_t - \boldsymbol{m}_t, \boldsymbol{p}_t - \widehat{\boldsymbol{p}}_{t+1} \rangle$$

$$- \sum_{t=1}^{T} \left( \mathcal{D}_{\psi_t}(\widehat{\boldsymbol{p}}_{t+1}, \boldsymbol{p}_t) + \mathcal{D}_{\psi_t}(\boldsymbol{p}_t, \widehat{\boldsymbol{p}}_t) \right)$$

$$\leq \underbrace{\sum_{t=1}^{T} \left( \mathcal{D}_{\psi_t}(\boldsymbol{u}, \widehat{\boldsymbol{p}}_t) - \mathcal{D}_{\psi_t}(\boldsymbol{u}, \widehat{\boldsymbol{p}}_{t+1}) \right)}_{\text{TERM (A)}} + \underbrace{\sum_{t=1}^{T} \left( \langle \boldsymbol{\ell}_t + \boldsymbol{a}_t - \boldsymbol{m}_t, \boldsymbol{p}_t - \widehat{\boldsymbol{p}}_{t+1} \rangle - \frac{1}{2} \mathcal{D}_{\psi_t}(\widehat{\boldsymbol{p}}_{t+1}, \boldsymbol{p}_t) \right)}_{\text{TERM (B)}}$$

$$-\frac{1}{2}\sum_{t=1}^{T}\left(\mathcal{D}_{\psi_t}(\widehat{\boldsymbol{p}}_{t+1},\boldsymbol{p}_t)+\mathcal{D}_{\psi_t}(\boldsymbol{p}_t,\widehat{\boldsymbol{p}}_t)\right),$$
$$\underbrace{\hphantom{-\frac{1}{2}\sum_{t=1}^{T}\left(\mathcal{D}_{\psi_t}(\widehat{\boldsymbol{p}}_{t+1},\boldsymbol{p}_t)+\mathcal{D}_{\psi_t}(\boldsymbol{p}_t,\widehat{\boldsymbol{p}}_t)\right),}}_{\text{TERM (C)}}$$

where the first step follows the standard analysis of optimistic OMD, e.g., Theorem 1 of Zhao et al. [2021]. One difference of our analysis from the previous one lies in the second step, where previous work dropped the $\mathcal{D}_{\psi_t}(\boldsymbol{p}_t,\widehat{\boldsymbol{p}}_t)$ term while we keep it to generate the desired negative terms.

To begin with, we require an upper bound of $\eta_{t,i}\leq 1/32$ for the step sizes. To give a lower bound for TERM (C), we notice that for any $\boldsymbol{a},\boldsymbol{b}\in\Delta_d$,

$$\mathcal{D}_{\psi_t}(\boldsymbol{a},\boldsymbol{b})=\sum_{i=1}^{d}\frac{1}{\eta_{t,i}}\left(a_i\ln\frac{a_i}{b_i}-a_i+b_i\right)=\sum_{i=1}^{d}\frac{b_i}{\eta_{t,i}}\left(\frac{a_i}{b_i}\ln\frac{a_i}{b_i}-\frac{a_i}{b_i}+1\right)$$

$$\geq\min_{t,i}\frac{1}{\eta_{t,i}}\sum_{i=1}^{d}\left(a_i\ln\frac{a_i}{b_i}-a_i+b_i\right)\geq 32\mathrm{KL}(\boldsymbol{a},\boldsymbol{b}),\qquad\text{(by }\eta_{t,i}\leq 1/32\text{)}$$

where the first inequality is due to $x\ln x-x+1\geq 0$ for all $x>0$, leading to

$$\text{TERM (C)}\geq 32\sum_{t=1}^{T}(\mathrm{KL}(\widehat{\boldsymbol{p}}_{t+1},\boldsymbol{p}_t)+\mathrm{KL}(\boldsymbol{p}_t,\widehat{\boldsymbol{p}}_t))\geq\frac{32}{2\ln 2}\sum_{t=1}^{T}\left(\|\widehat{\boldsymbol{p}}_{t+1}-\boldsymbol{p}_t\|_1^2+\|\boldsymbol{p}_t-\widehat{\boldsymbol{p}}_t\|_1^2\right)$$

$$\geq 16\sum_{t=2}^{T}\left(\|\widehat{\boldsymbol{p}}_t-\boldsymbol{p}_{t-1}\|_1^2+\|\boldsymbol{p}_t-\widehat{\boldsymbol{p}}_t\|_1^2\right)\geq 8\sum_{t=2}^{T}\|\boldsymbol{p}_t-\boldsymbol{p}_{t-1}\|_1^2,$$

where the first step is from the above derivation, the second step is due to the Pinsker's inequality [Pinsker, 1964]: $\mathrm{KL}(\boldsymbol{a},\boldsymbol{b})\geq\frac{1}{2\ln 2}\|\boldsymbol{a}-\boldsymbol{b}\|_1^2$ for any $\boldsymbol{a},\boldsymbol{b}\in\Delta_d$ and the last step is true because of $2x^2+2y^2\geq(x+y)^2$ for any $x,y\in\mathbb{R}$ and $\|\boldsymbol{a}+\boldsymbol{b}\|\leq\|\boldsymbol{a}\|+\|\boldsymbol{b}\|$ for any $\boldsymbol{a},\boldsymbol{b}\in\mathbb{R}^d$.

For TERM (B), the proof is similar to the previous work, where only some constants are modified. For self-containedness, we give the analysis below. Treating $\widehat{\boldsymbol{p}}_{t+1}$ as a free variable and defining

$$\boldsymbol{p}^\star\in\arg\max_{\boldsymbol{p}}\langle\boldsymbol{\ell}_t+\boldsymbol{a}_t-\boldsymbol{m}_t,\boldsymbol{p}_t-\boldsymbol{p}\rangle-\frac{1}{2}\mathcal{D}_{\psi_t}(\boldsymbol{p},\boldsymbol{p}_t),$$

by the optimality of $\boldsymbol{p}^\star$, we have

$$\boldsymbol{\ell}_t+\boldsymbol{a}_t-\boldsymbol{m}_t=\frac{1}{2}(\nabla\psi_t(\boldsymbol{p}_t)-\nabla\psi_t(\boldsymbol{p}^\star)).$$

Since $[\nabla\psi_t(\boldsymbol{p})]_i=\frac{1}{\eta_{t,i}}(\ln p_i+1)$, it holds that

$$\ell_{t,i}-m_{t,i}+a_{t,i}=\frac{1}{2\eta_{t,i}}\ln\frac{p_{t,i}}{p_i^\star}\Leftrightarrow p_i^\star=p_{t,i}\exp(-2\eta_{t,i}(\ell_{t,i}-m_{t,i}+a_{t,i})).$$

Therefore we have

$$\langle\boldsymbol{\ell}_t+\boldsymbol{a}_t-\boldsymbol{m}_t,\boldsymbol{p}_t-\widehat{\boldsymbol{p}}_{t+1}\rangle-\frac{1}{2}\mathcal{D}_{\psi_t}(\widehat{\boldsymbol{p}}_{t+1},\boldsymbol{p}_t)\leq\langle\boldsymbol{\ell}_t+\boldsymbol{a}_t-\boldsymbol{m}_t,\boldsymbol{p}_t-\boldsymbol{p}^\star\rangle-\frac{1}{2}\mathcal{D}_{\psi_t}(\boldsymbol{p}^\star,\boldsymbol{p}_t)$$

$$=\frac{1}{2}\langle\nabla\psi_t(\boldsymbol{p}_t)-\nabla\psi_t(\boldsymbol{p}^\star),\boldsymbol{p}_t-\boldsymbol{p}^\star\rangle-\frac{1}{2}\mathcal{D}_{\psi_t}(\boldsymbol{p}^\star,\boldsymbol{p}_t)=\frac{1}{2}\mathcal{D}_{\psi_t}(\boldsymbol{p}_t,\boldsymbol{p}^\star)\qquad\text{(by definition)}$$

$$=\frac{1}{2}\sum_{i=1}^{d}\frac{1}{\eta_{t,i}}\left(p_{t,i}\ln\frac{p_{t,i}}{p_i^\star}-p_{t,i}+p_i^\star\right)$$

$$=\frac{1}{2}\sum_{i=1}^{d}\frac{p_{t,i}}{\eta_{t,i}}\left(2\eta_{t,i}(\ell_{t,i}-m_{t,i}+a_{t,i})-1+\exp(-2\eta_{t,i}(\ell_{t,i}-m_{t,i}+a_{t,i}))\right)$$

$$\leq\frac{1}{2}\sum_{i=1}^{d}\frac{p_{t,i}}{\eta_{t,i}}4\eta_{t,i}^2(\ell_{t,i}-m_{t,i}+a_{t,i})^2=2\sum_{i=1}^{d}\eta_{t,i}p_{t,i}(\ell_{t,i}-m_{t,i}+a_{t,i})^2,$$

where the first and second steps use the optimality of $\boldsymbol{p}^\star$, the last inequality uses $e^{-x} - 1 + x \leq x^2$ for all $x \geq -1$, requiring $|2\eta_{t,i}(\ell_{t,i} - m_{t,i} + a_{t,i})| \leq 1$. It can be satisfied by $\eta_{t,i} \leq 1/32$ and $|\ell_{t,i} - m_{t,i} + a_{t,i}| \leq 16$, where the latter requirement can be satisfied by setting $a_{t,i} = 16\eta_{t,i}(\ell_{t,i} - m_{t,i})^2$:

$$|\ell_{t,i} - m_{t,i} + a_{t,i}| \leq 2 + 16 \cdot \frac{1}{32}(2\ell_{t,i}^2 + 2m_{t,i}^2) \leq 4 \leq 16.$$

As a result, we have

$$(\ell_{t,i} - m_{t,i} + a_{t,i})^2 = \left(\ell_{t,i} - m_{t,i} + 16\eta_{t,i}(\ell_{t,i} - m_{t,i})^2\right)^2 \leq 4(\ell_{t,i} - m_{t,i})^2,$$

where the last step holds because $|\ell_{t,i}|, |m_{t,i}| \leq 1$ and $\eta_{t,i} \leq 1/32$. Finally, it holds that

$$\text{TERM (B)} \leq 2\sum_{t=1}^{T}\sum_{i=1}^{d} \eta_{t,i} p_{t,i}(\ell_{t,i} - m_{t,i} + a_{t,i})^2 \leq 8\sum_{t=1}^{T}\sum_{i=1}^{d} \eta_{t,i} p_{t,i}(\ell_{t,i} - m_{t,i})^2.$$

As for TERM (A), following the same argument as Lemma 1 of Chen et al. [2021], we have

$$\text{TERM (A)} \leq \sum_{i=1}^{d} \frac{1}{\eta_{1,i}} f_{\text{KL}}(u_i, \widehat{p}_{1,i}) + \sum_{t=2}^{T}\sum_{i=1}^{d} \left(\frac{1}{\eta_{t,i}} - \frac{1}{\eta_{t-1,i}}\right) f_{\text{KL}}(u_i, \widehat{p}_{t,i}),$$

where $f_{\text{KL}}(a, b) \triangleq a \ln(a/b) - a + b$. Combining all three terms, we have

$$\sum_{t=1}^{T}\langle \boldsymbol{\ell}_t + \boldsymbol{a}_t, \boldsymbol{p}_t - \boldsymbol{u}\rangle \leq \sum_{i=1}^{d} \frac{1}{\eta_{1,i}} f_{\text{KL}}(u_i, \widehat{p}_{1,i}) + \sum_{t=2}^{T}\sum_{i=1}^{d} \left(\frac{1}{\eta_{t,i}} - \frac{1}{\eta_{t-1,i}}\right) f_{\text{KL}}(u_i, \widehat{p}_{t,i})$$

$$+ 8\sum_{t=1}^{T}\sum_{i=1}^{d} \eta_{t,i} p_{t,i}(\ell_{t,i} - m_{t,i})^2 - 4\sum_{t=2}^{T} \|\boldsymbol{p}_t - \boldsymbol{p}_{t-1}\|_1^2.$$

Moving the correction term $\sum_{t=1}^{T}\langle \boldsymbol{a}_t, \boldsymbol{p}_t - \boldsymbol{u}\rangle$ to the right-hand side gives:

$$\sum_{t=1}^{T}\langle \boldsymbol{\ell}_t, \boldsymbol{p}_t - \boldsymbol{u}\rangle \leq \sum_{i=1}^{d} \frac{1}{\eta_{1,i}} f_{\text{KL}}(u_i, \widehat{p}_{1,i}) + \sum_{t=2}^{T}\sum_{i=1}^{d} \left(\frac{1}{\eta_{t,i}} - \frac{1}{\eta_{t-1,i}}\right) f_{\text{KL}}(u_i, \widehat{p}_{t,i})$$

$$- 8\sum_{t=1}^{T}\sum_{i=1}^{d} \eta_{t,i} p_{t,i}(\ell_{t,i} - m_{t,i})^2 + 16\sum_{t=1}^{T}\sum_{i=1}^{d} \eta_{t,i} u_i(\ell_{t,i} - m_{t,i})^2 - 4\sum_{t=2}^{T} \|\boldsymbol{p}_t - \boldsymbol{p}_{t-1}\|_1^2.$$

Finally, choosing $\boldsymbol{u} = \boldsymbol{e}_{i^\star}$ and $\eta_{t,i} = \eta_i$ for all $t \in [T]$ finishes the proof. $\qquad\square$

### B.3  Proof of Lemma 3

In this part, we give a self-contained analysis of the two-layer meta learner, which mainly follows the Theorem 4 and Theorem 5 of Chen et al. [2021], but with additional negative stability terms.

**Lemma 3.** *If* $|\ell_{t,k}|, |m_{t,k}|, |\ell_{t,k,i}|, |m_{t,k,i}| \leq 1$ *and* $(\ell_{t,k} - m_{t,k})^2 = \langle \boldsymbol{\ell}_{t,k} - \boldsymbol{m}_{t,k}, \boldsymbol{p}_{t,k}\rangle^2$ *for any* $k \in [K]$ *and* $i \in [N]$, *then the two-layer* MsMwC *algorithm satisfies*

$$\sum_{t=1}^{T}\langle \boldsymbol{\ell}_t, \boldsymbol{q}_t - \boldsymbol{e}_{k^\star}\rangle + \sum_{t=1}^{T}\langle \boldsymbol{\ell}_{t,k^\star}, \boldsymbol{p}_{t,k^\star} - \boldsymbol{e}_{i^\star}\rangle \leq \frac{1}{\eta_{k^\star}} \ln \frac{N}{3C_0^2 \eta_{k^\star}^2} + 32\eta_{k^\star} V_\star - \frac{C_0}{2} S_T^{\boldsymbol{q}} - \frac{C_0}{4} S_{T,k^\star}^{\boldsymbol{p}},$$

*where* $V_\star \triangleq \sum_t (\ell_{t,k^\star,i^\star} - m_{t,k^\star,i^\star})^2$, $S_T^{\boldsymbol{q}} \triangleq \sum_{t=2}^{T} \|\boldsymbol{q}_t - \boldsymbol{q}_{t-1}\|_1^2$ *and* $S_{T,k}^{\boldsymbol{p}} \triangleq \sum_{t=2}^{T} \|\boldsymbol{p}_{t,k} - \boldsymbol{p}_{t-1,k}\|_1^2$ *measure the stability of* MsMwC-TOP *and* MsMwC-MID.

Note that we leverage a condition of $(\ell_{t,k} - m_{t,k})^2 = \langle \boldsymbol{\ell}_{t,k} - \boldsymbol{m}_{t,k}, \boldsymbol{p}_{t,k}\rangle^2$ in Lemma 3 only to ensure a self-contained result. When using Lemma 3 (more specifically, in Theorem 2), we will verify that this condition is inherently satisfied by our algorithm.

*Proof.* Using Lemma 2, the regret of MsMwC-TOP can be bounded as

$$\sum_{t=1}^{T}\langle \boldsymbol{\ell}_t, \boldsymbol{q}_t - \boldsymbol{e}_{k^\star}\rangle \leq \left(\frac{1}{\eta_{k^\star}} \ln \frac{1}{\widehat{q}_{1,k^\star}} + \sum_{k=1}^{K} \frac{\widehat{q}_{1,k}}{\eta_k}\right) + 0 + 16\eta_{k^\star} \sum_{t=1}^{T} (\ell_{t,k^\star} - m_{t,k^\star})^2 - \min_{k \in [K]} \frac{1}{4\eta_k} S_T^{\boldsymbol{q}},$$

where the first step comes from $f_{\mathrm{KL}}(a, b) = a \ln a/b - a + b \leq a \ln a/b + b$ for $a, b > 0$. The second term above (corresponding to the second term in Lemma 2) is zero since the step size is time-invariant. The first term above can be further bounded as

$$\frac{1}{\eta_{k^\star}} \ln \frac{1}{\widehat{q}_{1,k^\star}} + \sum_{k=1}^{K} \frac{\widehat{q}_{1,k}}{\eta_k} = \frac{1}{\eta_{k^\star}} \ln \frac{\sum_{k=1}^{K} \eta_k^2}{\eta_{k^\star}^2} + \frac{\sum_{k=1}^{K} \eta_k}{\sum_{k=1}^{K} \eta_k^2} \leq \frac{1}{\eta_{k^\star}} \ln \frac{1}{3C_0^2 \eta_{k^\star}^2} + 4C_0,$$

where the first step is due to the initialization of $\widehat{q}_{1,k} = \eta_k^2 / \sum_{k=1}^{K} \eta_k^2$. Plugging in the setting of $\eta_k = 1/(C_0 \cdot 2^k)$, the second step holds since

$$\frac{K^2}{4C_0^2} \leq \sum_{k=1}^{K} \eta_k^2 \leq \sum_{k=1}^{K} \frac{1}{C_0^2 \cdot 4^k} \leq \frac{1}{3C_0^2}, \quad \sum_{k=1}^{K} \eta_k \leq \frac{1}{C_0 \cdot 2^k} \leq \frac{1}{C_0}.$$

Since $1/\eta_k = C_0 \cdot 2^k \geq 2C_0$, the regret of MSMWC-TOP can be bounded by

$$\sum_{t=1}^{T} \langle \boldsymbol{\ell}_t, \boldsymbol{q}_t - \boldsymbol{e}_{k^\star} \rangle \leq \frac{1}{\eta_{k^\star}} \ln \frac{1}{3C_0^2 \eta_{k^\star}^2} + 16\eta_{k^\star} \sum_{t=1}^{T} (\ell_{t,k^\star} - m_{t,k^\star})^2 - \frac{C_0}{2} S_T^{\boldsymbol{q}} + \mathcal{O}(1).$$

Next, using Lemma 2 again, the regret of the $k^\star$-th MSMWC-MID, whose step size is $2\eta_{k^\star}$, can be bounded as

$$\sum_{t=1}^{T} \langle \boldsymbol{\ell}_{t,k^\star}, \boldsymbol{p}_{t,k^\star} - \boldsymbol{e}_{i^\star} \rangle \leq \frac{1}{2\eta_{k^\star}} \ln N + 32\eta_{k^\star} \sum_{t=1}^{T} (\ell_{t,k^\star,i^\star} - m_{t,k^\star,i^\star})^2 - \frac{C_0}{4} S_{T,k^\star}^{\boldsymbol{p}}$$

$$- 16\eta_{k^\star} \sum_{t=1}^{T} \sum_{i=1}^{N} p_{t,k^\star,i} (\ell_{t,k,i} - m_{t,k,i})^2,$$

where the first step is due to the initialization of $\widehat{p}_{1,k,i} = 1/N$. Based on the observation of

$$(\ell_{t,k^\star} - m_{t,k^\star})^2 = \langle \boldsymbol{\ell}_{t,k^\star} - \boldsymbol{m}_{t,k^\star}, \boldsymbol{p}_{t,k^\star} \rangle^2 \leq \sum_{i=1}^{N} p_{t,k^\star,i} (\ell_{t,k,i} - m_{t,k,i})^2,$$

where the last step use Cauchy-Schwarz inequality, combining the regret of MSMWC-TOP and the $k^\star$-th MSMWC-MID finishes the proof. $\square$

## B.4  Proof of Theorem 1

In Appendix C.2, we provide the proof of Theorem 2, the gradient-variation guarantees of the efficient version of Algorithm 1. Therefore, the proof of Theorem 1 can be directly extracted from that of Theorem 2. In the following, we give a proof sketch.

*Proof Sketch.* The proof of the meta regret is the same as Theorem 2 since the meta learners (top and middle layer) are both optimizing the linearized regret of $\sum_{t=1}^{T} \langle \nabla f_t(\mathbf{x}_t), \mathbf{x}_t - \mathbf{x}_{t,k^\star,i^\star} \rangle$. The only difference lies in the base regret. In Theorem 1, since the base regret is defined on the original function $f_t$, the gradient-variation bound of the base learner can be directly obtained via a black-box analysis of the optimistic OMD algorithm, e.g., Theorem B.1 of Zhang et al. [2022a] (strongly convex), Theorem 15 (exp-concave) and Theorem 11 (convex) of Chiang et al. [2012]. The only requirement on the base learners is that they need to cancel the positive term of $\|\mathbf{x}_{t,k^\star,i^\star} - \mathbf{x}_{t-1,k^\star,i^\star}\|^2$ caused by the universal optimism design (Lemma 1) and cascaded correction terms (Figure 1). The above base algorithms satisfy the requirement. The negative stability term in the optimistic OMD analysis is provided in Appendix E.2. $\square$

## B.5  Proof of Corollary 1

In Appendix C.3, we provide Corollary 2, the small-loss guarantees of the efficient version of Algorithm 1. The proof of Corollary 1 can be directly extracted from that of Corollary 2. We give a proof sketch below.

*Proof Sketch.* The proof sketch is similar to that of Theorem 1 in Appendix B.4. Specifically, the meta regret can be bounded in the same way. Since gradient-variation bounds naturally implies small-loss ones due to (E.3), the small-loss bound of the base learner can be directly obtained via a black-box analysis of the base algorithms mentioned in Appendix B.4, for different kinds of functions. The requirement of being capable of handling $\|\mathbf{x}_{t,k^\star,i^\star} - \mathbf{x}_{t-1,k^\star,i^\star}\|^2$ can be still satisfied, which finishes the proof sketch. $\square$

## C  Proofs for Section 4

In this section, we give the proofs for Section 4, including Proposition 1, Theorem 2 and Corollary 2.

### C.1  Proof of Proposition 1

*Proof.* To handle the meta regret, we use ADAPT-ML-PROD [Gaillard et al., 2014] to optimize the linear loss $\boldsymbol{\ell}_t = (\ell_{t,1}, \ldots, \ell_{t,N})$, where $\ell_{t,i} \triangleq \langle \nabla f_t(\mathbf{x}_t), \mathbf{x}_{t,i} \rangle$, and obtain the following second-order bound by Corollary 4 of Gaillard et al. [2014],

$$\sum_{t=1}^{T} \langle \nabla f_t(\mathbf{x}_t), \mathbf{x}_t - \mathbf{x}_{t,i^\star} \rangle \lesssim \sqrt{\ln\ln T \sum_{t=1}^{T} \langle \nabla f_t(\mathbf{x}_t), \mathbf{x}_t - \mathbf{x}_{t,i^\star} \rangle^2}.$$

For $\alpha$-exp-concave functions, it holds that

$$\text{META-REG} \lesssim \sqrt{\ln\ln T \sum_{t=1}^{T} \langle \nabla f_t(\mathbf{x}_t), \mathbf{x}_t - \mathbf{x}_{t,i^\star} \rangle^2} - \frac{\alpha_{i^\star}}{2} \sum_{t=1}^{T} \langle \nabla f_t(\mathbf{x}_t), \mathbf{x}_t - \mathbf{x}_{t,i^\star} \rangle^2$$

$$\leq \frac{\ln\ln T}{2\alpha_{i^\star}} \leq \frac{\ln\ln T}{\alpha}, \qquad\qquad\qquad (\text{by } \alpha_{i^\star} \leq \alpha \leq 2\alpha_{i^\star})$$

where the second step uses AM-GM inequality: $\sqrt{xy} \leq \frac{ax}{2} + \frac{y}{2a}$ for any $x, y, a > 0$ with $a = \alpha_{i^\star}$. To handle the base regret, by optimizing the surrogate loss function $h_{t,i^\star}^{\exp}$ using online Newton step [Hazan et al., 2007], it holds that

$$\text{BASE-REG} \lesssim \frac{dDG_{\exp}}{\alpha_{i^\star}} \log T \leq \frac{2dD(G + GD)}{\alpha} \log T,$$

where $G_{\exp} \triangleq \max_{\mathbf{x} \in \mathcal{X}, t \in [T], i \in [N]} \|\nabla h_{t,i}^{\exp}(\mathbf{x})\| \leq G + GD$ represents the maximum gradient norm the last step is because $\alpha \leq 2\alpha_{i^\star}$. Combining the meta and base regret, the regret can be bounded by $\mathcal{O}(d \log T)$. For $\lambda$-strongly convex functions, since it is also $\alpha = \lambda/G^2$ exp-concave under Assumption 1 [Hazan et al., 2007, Section 2.2], the above meta regret analysis is still applicable. To optimize the base regret, by optimizing the surrogate loss function $h_{t,i^\star}^{\text{sc}}$ using online gradient descent [Hazan et al., 2007], it holds that

$$\text{BASE-REG} \leq \frac{G_{\text{sc}}^2}{\lambda_{i^\star}}(1 + \log T) \leq \frac{2(G+D)^2}{\lambda}(1 + \log T),$$

where $G_{\text{sc}} \triangleq \max_{\mathbf{x} \in \mathcal{X}, t \in [T], i \in [N]} \|\nabla h_{t,i}^{\text{sc}}(\mathbf{x})\| \leq G + D$ represents the maximum gradient norm and the last step is because $\lambda \leq 2\lambda_{i^\star}$. Thus the overall regret can be bounded by $\mathcal{O}(\log T)$. For convex functions, the meta regret can be bounded by $\mathcal{O}(\sqrt{T \ln\ln T})$, where the $\ln\ln T$ factor can be omitted in the $\mathcal{O}(\cdot)$-notation, and the base regret can be bounded by $\mathcal{O}(\sqrt{T})$ using OGD, resulting in an $\mathcal{O}(\sqrt{T})$ regret overall, which completes the proof. $\square$

### C.2  Proof of Theorem 2

*Proof.* We first give different decompositions for the regret, then analyze the meta regret, and finally provide the proofs for different kinds of loss functions. Some abbreviations of the stability terms are defined in (B.1).

**Regret Decomposition.** Denoting by $\mathbf{x}^\star \in \arg\min_{\mathbf{x} \in \mathcal{X}} \sum_{t=1}^{T} f_t(\mathbf{x})$, for exp-concave functions,

$$
\sum_{t=1}^{T} f_t(\mathbf{x}_t) - \sum_{t=1}^{T} f_t(\mathbf{x}^\star) \leq \sum_{t=1}^{T} \langle \nabla f_t(\mathbf{x}_t), \mathbf{x}_t - \mathbf{x}^\star \rangle - \frac{\alpha}{2} \sum_{t=1}^{T} \langle \nabla f_t(\mathbf{x}_t), \mathbf{x}_t - \mathbf{x}^\star \rangle^2
$$

$$
\leq \sum_{t=1}^{T} \langle \nabla f_t(\mathbf{x}_t), \mathbf{x}_t - \mathbf{x}^\star \rangle - \frac{\alpha_{i^\star}}{2} \sum_{t=1}^{T} \langle \nabla f_t(\mathbf{x}_t), \mathbf{x}_t - \mathbf{x}^\star \rangle^2 \qquad \text{(by } \alpha_{i^\star} \leq \alpha \leq 2\alpha_{i^\star})
$$

$$
= \underbrace{\sum_{t=1}^{T} \langle \nabla f_t(\mathbf{x}_t), \mathbf{x}_t - \mathbf{x}_{t,k^\star,i^\star} \rangle - \frac{\alpha_{i^\star}}{2} \langle \nabla f_t(\mathbf{x}_t), \mathbf{x}_t - \mathbf{x}_{t,k^\star,i^\star} \rangle^2}_{\text{META-REG}} + \underbrace{\sum_{t=1}^{T} h_{t,i^\star}^{\exp}(\mathbf{x}_{t,k^\star,i^\star}) - \sum_{t=1}^{T} h_{t,i^\star}^{\exp}(\mathbf{x}^\star)}_{\text{BASE-REG}},
$$

where the second step uses the strong exp-concavity and the last step holds by defining surrogate loss functions $h_{t,i}^{\exp}(\mathbf{x}) \triangleq \langle \nabla f_t(\mathbf{x}_t), \mathbf{x} \rangle + \frac{\alpha_i}{2} \langle \nabla f_t(\mathbf{x}_t), \mathbf{x} - \mathbf{x}_t \rangle^2$. Similarly, for strongly convex functions, the regret can be upper-bounded by

$$
\underbrace{\sum_{t=1}^{T} \langle \nabla f_t(\mathbf{x}_t), \mathbf{x}_t - \mathbf{x}_{t,k^\star,i^\star} \rangle - \frac{\lambda_{i^\star}}{2} \|\mathbf{x}_t - \mathbf{x}_{t,k^\star,i^\star}\|^2}_{\text{META-REG}} + \underbrace{\sum_{t=1}^{T} h_{t,i^\star}^{\mathrm{sc}}(\mathbf{x}_{t,k^\star,i^\star}) - \sum_{t=1}^{T} h_{t,i^\star}^{\mathrm{sc}}(\mathbf{x}^\star)}_{\text{BASE-REG}},
$$

by defining surrogate loss functions $h_{t,i}^{\mathrm{sc}}(\mathbf{x}) \triangleq \langle \nabla f_t(\mathbf{x}_t), \mathbf{x} \rangle + \frac{\lambda_i}{2} \|\mathbf{x} - \mathbf{x}_t\|^2$. For convex functions, the regret can be decomposed as:

$$
\text{REG}_T \leq \underbrace{\sum_{t=1}^{T} \langle \nabla f_t(\mathbf{x}_t), \mathbf{x}_t - \mathbf{x}_{t,k^\star,i^\star} \rangle}_{\text{META-REG}} + \underbrace{\sum_{t=1}^{T} \langle \nabla f_t(\mathbf{x}_t), \mathbf{x}_{t,k^\star,i^\star} - \mathbf{x}^\star \rangle}_{\text{BASE-REG}}.
$$

**Meta Regret Analysis.** For the meta regret, we focus on the linearized term since the negative term by exp-concavity or strong convexity only exists in analysis. Specifically,

$$
\sum_{t=1}^{T} \langle \nabla f_t(\mathbf{x}_t), \mathbf{x}_t - \mathbf{x}_{t,k^\star,i^\star} \rangle = \sum_{t=1}^{T} \langle \nabla f_t(\mathbf{x}_t), \mathbf{x}_t - \mathbf{x}_{t,k^\star} \rangle + \sum_{t=1}^{T} \langle \nabla f_t(\mathbf{x}_t), \mathbf{x}_{t,k^\star} - \mathbf{x}_{t,k^\star,i^\star} \rangle
$$

$$
= \sum_{t=1}^{T} \langle \boldsymbol{\ell}_t, \boldsymbol{q}_t - \boldsymbol{e}_{k^\star} \rangle + \sum_{t=1}^{T} \langle \boldsymbol{\ell}_{t,k^\star}, \boldsymbol{p}_{t,k^\star} - \boldsymbol{e}_{i^\star} \rangle - \lambda_1 \sum_{t=1}^{T} \sum_{k=1}^{K} q_{t,k} \|\mathbf{x}_{t,k} - \mathbf{x}_{t-1,k}\|^2
$$

$$
- \lambda_2 \sum_{t=1}^{T} \sum_{i=1}^{N} p_{t,k^\star,i} \|\mathbf{x}_{t,k^\star,i} - \mathbf{x}_{t-1,k^\star,i}\|^2 + \lambda_1 S_{T,k^\star}^{\mathbf{x}} + \lambda_2 S_{T,k^\star,i^\star}^{\mathbf{x}} \quad \text{(by definition of } \boldsymbol{\ell}_t, \boldsymbol{\ell}_{t,k})
$$

$$
\leq \sum_{t=1}^{T} \langle \boldsymbol{\ell}_t, \boldsymbol{q}_t - \boldsymbol{e}_{k^\star} \rangle + \sum_{t=1}^{T} \langle \boldsymbol{\ell}_{t,k^\star}, \boldsymbol{p}_{t,k^\star} - \boldsymbol{e}_{i^\star} \rangle - \lambda_1 \sum_{t=1}^{T} \sum_{k=1}^{K} q_{t,k} \|\mathbf{x}_{t,k} - \mathbf{x}_{t-1,k}\|^2 + \lambda_2 S_{T,k^\star,i^\star}^{\mathbf{x}}
$$

$$
+ (2\lambda_1 - \lambda_2) \sum_{t=1}^{T} \sum_{i=1}^{N} p_{t,k^\star,i} \|\mathbf{x}_{t,k^\star,i} - \mathbf{x}_{t-1,k^\star,i}\|^2 + 2\lambda_1 D^2 S_{T,k^\star}^{\boldsymbol{p}} \qquad \text{(by Lemma 6)}
$$

$$
\leq \sum_{t=1}^{T} \langle \boldsymbol{\ell}_t, \boldsymbol{q}_t - \boldsymbol{e}_{k^\star} \rangle + \sum_{t=1}^{T} \langle \boldsymbol{\ell}_{t,k^\star}, \boldsymbol{p}_{t,k^\star} - \boldsymbol{e}_{i^\star} \rangle - \lambda_1 \sum_{t=1}^{T} \sum_{k=1}^{K} q_{t,k} \|\mathbf{x}_{t,k} - \mathbf{x}_{t-1,k}\|^2
$$

$$
+ 2\lambda_1 D^2 S_{T,k^\star}^{\boldsymbol{p}} + \lambda_2 S_{T,k^\star,i^\star}^{\mathbf{x}}. \qquad \text{(requiring } \lambda_2 \geq 2\lambda_1)
$$

Next, we investigate the regret of the two-layer meta algorithm (i.e., the first two terms above). To begin with, we validate the conditions of Lemma 3. First, since

$$
|\ell_{t,k,i}| \leq GD + \lambda_2 D^2, \quad |m_{t,k,i}| \leq 2GD + \lambda_2 D^2,
$$

$$
|\ell_{t,k}| \leq GD + \lambda_1 D^2, \quad |m_{t,k}| \leq 2GD + (\lambda_1 + \lambda_2)D^2,
$$

rescaling the ranges of losses and optimisms to $[-1, 1]$ only add a constant multiplicative factors on the final result ($\lambda_1$ and $\lambda_2$ only consist of constants). Second, by the definition of the losses and the optimisms, it holds that

$$(\ell_{t,k^\star} - m_{t,k^\star})^2 = (\langle \nabla f_t(\mathbf{x}_t), \mathbf{x}_{t,k^\star} \rangle - \langle \widehat{\boldsymbol{m}}_{t,k^\star}, \boldsymbol{p}_{t,k^\star} \rangle)^2$$
$$= \langle \widehat{\boldsymbol{\ell}}_{t,k^\star} - \widehat{\boldsymbol{m}}_{t,k^\star}, \boldsymbol{p}_{t,k^\star} \rangle^2 = \langle \boldsymbol{\ell}_{t,k^\star} - \boldsymbol{m}_{t,k^\star}, \boldsymbol{p}_{t,k^\star} \rangle^2$$

where $\widehat{\ell}_{t,k,i} \triangleq \langle \nabla f_t(\mathbf{x}_t), \mathbf{x}_{t,k,i} \rangle$. Using Lemma 3, it holds that

$$\sum_{t=1}^{T} \langle \nabla f_t(\mathbf{x}_t), \mathbf{x}_t - \mathbf{x}_{t,k^\star,i^\star} \rangle \le \frac{1}{\eta_{k^\star}} \ln \frac{N}{\eta_{k^\star}^2} + 32\eta_{k^\star} V_\star + \lambda_2 S_{T,k^\star,i^\star}^{\mathbf{x}} \qquad \text{(requiring } C_0 \ge 1\text{)}$$

$$- \frac{C_0}{2} S_T^{\boldsymbol{q}} - \lambda_1 \sum_{t=2}^{T} \sum_{k=1}^{K} q_{t,k} \|\mathbf{x}_{t,k} - \mathbf{x}_{t-1,k}\|^2 + \left( 2\lambda_1 D^2 - \frac{C_0}{4} \right) S_{T,k^\star}^{\boldsymbol{p}}$$

$$\le \frac{1}{\eta_{k^\star}} \ln \frac{N}{\eta_{k^\star}^2} + 32\eta_{k^\star} V_\star + \lambda_2 S_{T,k^\star,i^\star}^{\mathbf{x}} - \frac{C_0}{2} S_T^{\boldsymbol{q}} - \lambda_1 \sum_{t=2}^{T} \sum_{k=1}^{K} q_{t,k} \|\mathbf{x}_{t,k} - \mathbf{x}_{t-1,k}\|^2,$$

where the last step holds by requiring $C_0 \ge 4\lambda_1 D^2$.

**Exp-concave Functions.** Using Lemma 1, it holds that

$$\frac{1}{\eta_{k^\star}} \ln \frac{N}{\eta_{k^\star}^2} + 32\eta_{k^\star} V_\star \le \frac{1}{\eta_{k^\star}} \ln \frac{N}{\eta_{k^\star}^2} + 128\eta_{k^\star} \sum_{t=1}^{T} \langle \nabla f_t(\mathbf{x}_t), \mathbf{x}_{t,k^\star,i^\star} - \mathbf{x}_t \rangle^2 + \mathcal{O}(1).$$

As a result, the meta regret can be bounded by

$$\text{META-REG} \le \frac{1}{\eta_{k^\star}} \ln \frac{N}{\eta_{k^\star}^2} + \left( 128\eta_{k^\star} - \frac{\alpha_{i^\star}}{2} \right) \sum_{t=1}^{T} \langle \nabla f_t(\mathbf{x}_t), \mathbf{x}_{t,k^\star,i^\star} - \mathbf{x}_t \rangle^2 + \lambda_2 S_{T,k^\star,i^\star}^{\mathbf{x}}$$

$$- \frac{C_0}{2} S_T^{\boldsymbol{q}} - \lambda_1 \sum_{t=2}^{T} \sum_{k=1}^{K} q_{t,k} \|\mathbf{x}_{t,k} - \mathbf{x}_{t-1,k}\|^2$$

$$\le 2C_0 \ln(4C_0^2 N) + \frac{512}{\alpha} \ln \frac{2^{20} N}{\alpha^2} + \lambda_2 S_{T,k^\star,i^\star}^{\mathbf{x}} - \frac{C_0}{2} S_T^{\boldsymbol{q}} \qquad \text{(by Lemma 8 with } \eta_\star \triangleq \frac{\alpha_{i^\star}}{256}\text{)}$$

$$- \lambda_1 \sum_{t=2}^{T} \sum_{k=1}^{K} q_{t,k} \|\mathbf{x}_{t,k} - \mathbf{x}_{t-1,k}\|^2, \qquad\qquad\qquad (\text{C.1})$$

where the last step holds by $\alpha_{i^\star} \le \alpha \le 2\alpha_{i^\star}$. For the base regret, according to Lemma 11, the $i^\star$-th base learner guarantees

$$\sum_{t=1}^{T} h_{t,i^\star}^{\exp}(\mathbf{x}_{t,k^\star,i^\star}) - \sum_{t=1}^{T} h_{t,i^\star}^{\exp}(\mathbf{x}^\star) \le \frac{16d}{\alpha_{i^\star}} \ln \left( 1 + \frac{\alpha_{i^\star}}{8\gamma d} \bar{V}_{T,k^\star,i^\star} \right) + \frac{1}{2} \gamma D^2 - \frac{\gamma}{4} S_{T,k^\star,i^\star}^{\mathbf{x}}, \quad (\text{C.2})$$

where $\bar{V}_{T,k^\star,i^\star} = \sum_{t=2}^{T} \|\nabla h_{t,i^\star}^{\exp}(\mathbf{x}_{t,k^\star,i^\star}) - \nabla h_{t-1,i^\star}^{\exp}(\mathbf{x}_{t-1,k^\star,i^\star})\|^2$ denotes the empirical gradient variation of the $i^\star$-th base learner. For simplicity, we denote by $\mathbf{g}_t \triangleq \nabla f_t(\mathbf{x}_t)$, and this term can be further decomposed and bounded as:

$$\bar{V}_{T,k^\star,i^\star} = \sum_{t=2}^{T} \|(\mathbf{g}_t + \alpha_{i^\star} \mathbf{g}_t \mathbf{g}_t^\top (\mathbf{x}_{t,k^\star,i^\star} - \mathbf{x}_t)) - (\mathbf{g}_{t-1} + \alpha_{i^\star} \mathbf{g}_{t-1} \mathbf{g}_{t-1}^\top (\mathbf{x}_{t-1,k^\star,i^\star} - \mathbf{x}_{t-1}))\|^2$$

$$\le 2 \sum_{t=2}^{T} \|\mathbf{g}_t - \mathbf{g}_{t-1}\|^2 + 2\alpha_{i^\star}^2 \sum_{t=2}^{T} \|\mathbf{g}_t \mathbf{g}_t^\top (\mathbf{x}_{t,k^\star,i^\star} - \mathbf{x}_t) - \mathbf{g}_{t-1} \mathbf{g}_{t-1}^\top (\mathbf{x}_{t-1,k^\star,i^\star} - \mathbf{x}_{t-1})\|^2$$

$$\le 4V_T + 4L^2 S_T^{\mathbf{x}} + 4D^2 \sum_{t=2}^{T} \|\mathbf{g}_t \mathbf{g}_t^\top - \mathbf{g}_{t-1} \mathbf{g}_{t-1}^\top\|^2 \qquad\qquad (\text{by (E.2)})$$

$$
+ 4G^4 \sum_{t=2}^{T} \|(\mathbf{x}_{t,k^\star,i^\star} - \mathbf{x}_t) - (\mathbf{x}_{t-1,k^\star,i^\star} - \mathbf{x}_{t-1})\|^2 \qquad \text{(by } \alpha \in [1/T, 1])
$$

$$
\leq C_1 V_T + C_2 S_T^{\mathbf{x}} + 8G^4 S_{T,k^\star,i^\star}^{\mathbf{x}},
$$

where the first step uses the definition of $\nabla h_{t,i}^{\exp}(\mathbf{x}) = \mathbf{g}_t + \alpha_i \mathbf{g}_t \mathbf{g}_t^\top (\mathbf{x} - \mathbf{x}_t)$ and the last step holds by setting $C_1 = 4 + 32D^2 G^2$ and $C_2 = 4L^2 + 32D^2 G^2 L^2 + 8G^4$. Then we obtain

$$
\ln\left(1 + \frac{\alpha_{i^\star}}{8\gamma d} \bar{V}_{T,k^\star,i^\star}\right) \leq \ln\left(1 + \frac{\alpha_{i^\star} C_1}{8\gamma d} V_T + \frac{\alpha_{i^\star} C_2}{8\gamma d} S_T^{\mathbf{x}} + \frac{\alpha_{i^\star} G^4}{\gamma d} S_{T,k^\star,i^\star}^{\mathbf{x}}\right)
$$

$$
= \ln\left(1 + \frac{\alpha_{i^\star} C_1}{8\gamma d} V_T\right) + \ln\left(1 + \frac{\frac{\alpha_{i^\star} C_2}{8\gamma d} S_T^{\mathbf{x}} + \frac{\alpha_{i^\star} G^4}{\gamma d} S_{T,k^\star,i^\star}^{\mathbf{x}}}{1 + \frac{\alpha_{i^\star} C_1}{8\gamma d} V_T}\right)
$$

$$
\leq \ln\left(1 + \frac{\alpha_{i^\star} C_1}{8\gamma d} V_T\right) + \frac{\alpha_{i^\star} C_2}{8\gamma d} S_T^{\mathbf{x}} + \frac{\alpha_{i^\star} G^4}{\gamma d} S_{T,k^\star,i^\star}^{\mathbf{x}}. \qquad (\ln(1+x) \leq x \text{ for } x \geq 0)
$$

Plugging the above result and due to $\alpha_{i^\star} \leq \alpha \leq 2\alpha_{i^\star}$, the base regret can be bounded by

$$
\textsc{Base-Reg} \leq \frac{32d}{\alpha} \ln\left(1 + \frac{\alpha C_1}{8\gamma d} V_T\right) + \frac{2C_2}{\gamma} S_T^{\mathbf{x}} + \left(\frac{16G^4}{\gamma} - \frac{\gamma}{4}\right) S_{T,k^\star,i^\star}^{\mathbf{x}} + \frac{1}{2}\gamma D^2.
$$

Combining the meta regret and base regret, the overall regret can be bounded by

$$
\textsc{Reg}_T \leq \frac{512}{\alpha} \ln\frac{2^{20} N}{\alpha^2} + 2C_0 \ln(4C_0^2 N) + \frac{32d}{\alpha} \ln\left(1 + \frac{\alpha C_1}{8\gamma d} V_T\right)
$$

$$
+ \left(\frac{4C_2 D^2}{\gamma} - \frac{C_0}{2}\right) S_T^{\boldsymbol{q}} + \left(\frac{4C_2}{\gamma} - \lambda_1\right) \sum_{t=2}^{T} \sum_{k=1}^{K} q_{t,k} \|\mathbf{x}_{t,k} - \mathbf{x}_{t-1,k}\|^2
$$

$$
+ \left(\lambda_2 + \frac{16G^4}{\gamma} - \frac{\gamma}{4}\right) S_{T,k^\star,i^\star}^{\mathbf{x}} + \frac{1}{2}\gamma D^2
$$

$$
\leq \mathcal{O}\left(\frac{d}{\alpha} \ln V_T\right). \qquad \text{(requiring } C_0 \geq 8C_2 D^2/\gamma, \ \lambda_1 \geq 4C_2/\gamma, \ \gamma \geq 8G^2 + \lambda_2)
$$

**Strongly Convex Functions.** Since a $\lambda$-strongly convex function is also $\alpha = \lambda/G^2$ exp-concave under Assumption 1 [Hazan et al., 2007, Section 2.2], plugging $\alpha = \lambda/G^2$ into the above analysis, the meta regret can be bounded by

$$
\textsc{Meta-Reg} \leq \frac{512G^2}{\lambda} \ln\frac{2^{20} N G^4}{\lambda^2} + 2C_0 \ln(4C_0^2 N) + \lambda_2 S_{T,k^\star,i^\star}^{\mathbf{x}}
$$
$$
- \frac{C_0}{2} S_T^{\boldsymbol{q}} - \lambda_1 \sum_{t=2}^{T} \sum_{k=1}^{K} q_{t,k} \|\mathbf{x}_{t,k} - \mathbf{x}_{t-1,k}\|^2. \qquad \text{(C.3)}
$$

For the base regret, according to Lemma 12, the $i^\star$-th base learner guarantees

$$
\sum_{t=1}^{T} h_{t,i^\star}^{\mathrm{sc}}(\mathbf{x}_{t,k^\star,i^\star}) - \sum_{t=1}^{T} h_{t,i^\star}^{\mathrm{sc}}(\mathbf{x}^\star) \leq \frac{16G_{\mathrm{sc}}^2}{\lambda_{i^\star}} \ln\left(1 + \lambda_{i^\star} \bar{V}_{T,k^\star,i^\star}\right) + \frac{1}{4}\gamma D^2 - \frac{\gamma}{8} S_{T,k^\star,i^\star}^{\mathbf{x}}, \quad \text{(C.4)}
$$

where $\bar{V}_{T,k^\star,i^\star} = \sum_{t=2}^{T} \|\nabla h_{t,i^\star}^{\mathrm{sc}}(\mathbf{x}_{t,k^\star,i^\star}) - \nabla h_{t-1,i^\star}^{\mathrm{sc}}(\mathbf{x}_{t-1,k^\star,i^\star})\|^2$ is the empirical gradient variation and $G_{\mathrm{sc}} \triangleq \max_{\mathbf{x} \in \mathcal{X}, t \in [T], i \in [N]} \|\nabla h_{t,i}^{\mathrm{sc}}(\mathbf{x})\| \leq G + D$ represents the maximum gradient norm. For simplicity, we denote by $\mathbf{g}_t \triangleq \nabla f_t(\mathbf{x}_t)$, and the empirical gradient variation $\bar{V}_{T,k^\star,i^\star}$ can be further decomposed as

$$
\bar{V}_{T,k^\star,i^\star} = \sum_{t=2}^{T} \|(\mathbf{g}_t + \lambda_{i^\star}(\mathbf{x}_{t,k^\star,i^\star} - \mathbf{x}_t)) - (\mathbf{g}_{t-1} + \lambda_{i^\star}(\mathbf{x}_{t-1,k^\star,i^\star} - \mathbf{x}_{t-1}))\|^2
$$

$$
\leq 2 \sum_{t=2}^{T} \|\mathbf{g}_t - \mathbf{g}_{t-1}\|^2 + 2\lambda_{i^\star}^2 \sum_{t=2}^{T} \|(\mathbf{x}_{t,k^\star,i^\star} - \mathbf{x}_t) - (\mathbf{x}_{t-1,k^\star,i^\star} - \mathbf{x}_{t-1})\|^2
$$

$$\leq 4V_T + (4 + 4L^2)S_T^{\mathbf{x}} + 4S_{T,k^\star,i^\star}^{\mathbf{x}}, \qquad\qquad (\text{by } \lambda \in [1/T, 1])$$

where the first step uses the definition of $\nabla h^{\mathrm{sc}}(\mathbf{x}) = \mathbf{g}_t + \lambda_i(\mathbf{x} - \mathbf{x}_t)$. Consequently,

$$\ln\left(1 + \lambda_{i^\star}\bar{V}_{T,k^\star,i^\star}\right) \leq \ln(1 + 4\lambda_{i^\star}V_T + (4 + 4L^2)\lambda_{i^\star}S_T^{\mathbf{x}} + 4\lambda_{i^\star}S_{T,k^\star,i^\star}^{\mathbf{x}})$$

$$= \ln(1 + 4\lambda_{i^\star}V_T) + \ln\left(1 + \frac{(4 + 4L^2)\lambda_{i^\star}S_T^{\mathbf{x}} + 4\lambda_{i^\star}S_{T,k^\star,i^\star}^{\mathbf{x}}}{1 + 4\lambda_{i^\star}V_T}\right)$$

$$\leq \ln(1 + 4\lambda_{i^\star}V_T) + (4 + 4L^2)\lambda_{i^\star}S_T^{\mathbf{x}} + 4\lambda_{i^\star}S_{T,k^\star,i^\star}^{\mathbf{x}}. \qquad (\ln(1+x) \leq x \text{ for } x \geq 0)$$

Plugging the above result and due to $\lambda_{i^\star} \leq \lambda \leq 2\lambda_{i^\star}$, the base regret can be bounded by

$$\text{BASE-REG} \leq \frac{32G_{\mathrm{sc}}^2}{\lambda}\ln(1 + 4\lambda V_T) + 16C_3 S_T^{\mathbf{x}} + \left(64G_{\mathrm{sc}}^2 - \frac{\gamma}{8}\right)S_{T,k^\star,i^\star}^{\mathbf{x}} + \frac{1}{4}\gamma D^2,$$

where $C_3 = (4 + 4L^2)G_{\mathrm{sc}}^2$. Combining the meta regret and base regret, the overall regret satisfies

$$\text{REG}_T \leq \frac{512G^2}{\lambda}\ln\frac{2^{20}NG^4}{\lambda^2} + 2C_0\ln(4C_0^2 N) + \frac{32(G + D)^2}{\lambda}\ln(1 + 4\lambda V_T)$$

$$+ \left(32D^2 C_3 - \frac{C_0}{2}\right)S_T^q + (32C_3 - \lambda_1)\sum_{t=2}^{T}\sum_{k=1}^{K} q_{t,k}\|\mathbf{x}_{t,k} - \mathbf{x}_{t-1,k}\|^2$$

$$+ \left(\lambda_2 + 64G_{\mathrm{sc}}^2 - \frac{\gamma}{8}\right)S_{T,k^\star,i^\star}^{\mathbf{x}} + \frac{1}{4}\gamma D^2$$

$$\leq \mathcal{O}\left(\frac{1}{\lambda}\ln V_T\right). \qquad (\text{requiring } C_0 \geq 64D^2 C_3, \lambda_1 \geq 32C_3, \gamma \geq 8\lambda_2 + 512G_{\mathrm{sc}}^2)$$

**Convex Functions.** Using Lemma 1, it holds that

$$\frac{1}{\eta_{k^\star}}\ln\frac{N}{\eta_{k^\star}^2} + 32\eta_{k^\star}V_\star \leq \frac{1}{\eta_{k^\star}}\ln\frac{N}{\eta_{k^\star}^2} + 128\eta_{k^\star}D^2 V_T + 2(D^2 L^2 + G^2)S_T^{\mathbf{x}} + 2G^2 S_{T,k^\star,i^\star}^{\mathbf{x}}$$

$$\leq 2C_0\ln(4C_0^2 N) + 32D\sqrt{2V_T\ln(512ND^2 V_T)} + 2(D^2 L^2 + G^2)S_T^{\mathbf{x}} + 2G^2 S_{T,k^\star,i^\star}^{\mathbf{x}}$$

where the first step is due to $\eta_k = 1/(C_0 \cdot 2^k) \leq 1/(2C_0)$ and requiring $C_0 \geq 1$ and the last step is by Lemma 7 and requiring $C_0 \geq 8D$. Thus the meta regret can be bounded by

$$\text{META-REG} \leq 2C_0\ln(4C_0^2 N) + \mathcal{O}(\sqrt{V_T\ln V_T}) + 2(D^2 L^2 + G^2)S_T^{\mathbf{x}}$$

$$+ (\lambda_2 + 2G^2)S_{T,k^\star,i^\star}^{\mathbf{x}} - \frac{C_0}{2}S_T^q - \lambda_1\sum_{t=2}^{T}\sum_{k=1}^{K} q_{t,k}\|\mathbf{x}_{t,k} - \mathbf{x}_{t-1,k}\|^2$$

For the base regret, according to Lemma 10, the convex base learner guarantees

$$\text{BASE-REG} = \sum_{t=1}^{T}\langle\nabla f_t(\mathbf{x}_t), \mathbf{x}_{t,k^\star,i^\star} - \mathbf{x}^\star\rangle \leq 5D\sqrt{1 + \bar{V}_{T,k^\star,i^\star}} + \gamma D^2 - \frac{\gamma}{4}S_{T,k^\star,i^\star}^{\mathbf{x}}, \qquad (\text{C.5})$$

where $\bar{V}_{T,k^\star,i^\star} = \sum_{t=2}^{T}\|\nabla f_t(\mathbf{x}_t) - \nabla f_{t-1}(\mathbf{x}_{t-1})\|^2$ denotes the empirical gradient variation of the base learner. Via (E.2), the base regret can be bounded by

$$\text{BASE-REG} \leq 5D\sqrt{1 + 2V_T} + 10DL^2 S_T^{\mathbf{x}} + \gamma D^2 - \frac{\gamma}{4}S_{T,k^\star,i^\star}^{\mathbf{x}}.$$

Combining the meta regret and base regret, the overall regret can be bounded by

$$\text{REG}_T \leq 2C_0\ln(4C_0^2 N) + \mathcal{O}(\sqrt{V_T\ln V_T}) + 5D\sqrt{1 + 2V_T}$$

$$+ \left(2DC_4 - \frac{C_0}{2}\right)S_T^q + (2C_4 - \lambda_1)\sum_{t=2}^{T}\sum_{k=1}^{K} q_{t,k}\|\mathbf{x}_{t,k} - \mathbf{x}_{t-1,k}\|^2$$

$$+ \left(\lambda_2 + 2G^2 - \frac{\gamma}{4}\right)S_{T,k^\star,i^\star}^{\mathbf{x}} + \gamma D^2$$

$$\leq \mathcal{O}(\sqrt{V_T \ln V_T}), \qquad \text{(requiring } C_0 \geq 4DC_4, \ \lambda_1 \geq 2C_4, \ \gamma \geq 4\lambda_2 + 8G^2)$$

where $C_4 = 2D^2 L^2 + 2G^2 + 10DL^2$.

**Overall.** At last, we determine the specific values of $C_0$, $\lambda_1$ and $\lambda_2$. These parameters need to satisfy the following requirements:

$$C_0 \geq 1, \ C_0 \geq 4\lambda_1 D^2, \ C_0 \geq 128, \ C_0 \geq \frac{8C_2 D^2}{8G^2 + \lambda_2}, \ C_0 \geq 64D^2 C_3, \ C_0 \geq 8D, \ C_0 \geq 4DC_4,$$

$$\lambda_1 \geq \frac{4C_2}{8G^2 + \lambda_2}, \ \lambda_1 \geq 32C_3, \ \lambda_1 \geq 2C_4, \ \lambda_2 \geq 2\lambda_1.$$

As a result, we set $C_0, \lambda_1, \lambda_2$ to be the minimum constants satisfying the above conditions., which completes the proof. $\qquad \square$

### C.3 Proof of Corollary 2

In this part, we provide Corollary 2, which obtains the same small-loss bounds as Corollary 1, but only requires one gradient query within each round.

**Corollary 2.** *Under Assumptions 1 and 2, if $f_t(\cdot) \geq 0$ for all $t \in [T]$, the efficient version of Algorithm 1 obtains $\mathcal{O}(\log F_T)$, $\mathcal{O}(d \log F_T)$ and $\widehat{\mathcal{O}}(\sqrt{F_T})$ regret bounds for strongly convex, exp-concave and convex functions, using only one gradient per round.*

*Proof.* For simplicity, we define $F_T^{\mathbf{x}} \triangleq \sum_{t=1}^{T} f_t(\mathbf{x}_t)$. Some abbreviations are given in (B.1).

**Exp-concave Functions.** The meta regret can be bounded the same as Theorem 1 (especially (C.1)), and thus we directly move on to the base regret. For simplicity, we denote by $\mathbf{g}_t \triangleq \nabla f_t(\mathbf{x}_t)$ and give a different decomposition for $\bar{V}_{T,k^\star,i^\star} = \sum_{t=2}^{T} \|\nabla h_{t,i^\star}^{\exp}(\mathbf{x}_{t,k^\star,i^\star}) - \nabla h_{t-1,i^\star}^{\exp}(\mathbf{x}_{t-1,k^\star,i^\star})\|^2$, the empirical gradient variation of the base learner.

$$\bar{V}_{T,k^\star,i^\star} = \sum_{t=2}^{T} \|(\mathbf{g}_t + \alpha_{i^\star} \mathbf{g}_t \mathbf{g}_t^\top (\mathbf{x}_{t,k^\star,i^\star} - \mathbf{x}_t)) - (\mathbf{g}_{t-1} + \alpha_{i^\star} \mathbf{g}_{t-1} \mathbf{g}_{t-1}^\top (\mathbf{x}_{t-1,k^\star,i^\star} - \mathbf{x}_{t-1}))\|^2$$

$$\leq 2 \sum_{t=2}^{T} \|\mathbf{g}_t - \mathbf{g}_{t-1}\|^2 + 2\alpha_{i^\star}^2 \sum_{t=2}^{T} \|\mathbf{g}_t \mathbf{g}_t^\top (\mathbf{x}_{t,k^\star,i^\star} - \mathbf{x}_t) - \mathbf{g}_{t-1} \mathbf{g}_{t-1}^\top (\mathbf{x}_{t-1,k^\star,i^\star} - \mathbf{x}_{t-1})\|^2$$

$$\leq 2 \sum_{t=2}^{T} \|\mathbf{g}_t - \mathbf{g}_{t-1}\|^2 + 4D^2 \sum_{t=2}^{T} \|\mathbf{g}_t \mathbf{g}_t^\top - \mathbf{g}_{t-1} \mathbf{g}_{t-1}^\top\|^2 + 8G^4 S_T^{\mathbf{x}} + 8G^4 S_{T,k^\star,i^\star}^{\mathbf{x}}$$

$$\leq (2 + 16D^2 G^2) \sum_{t=2}^{T} \|\mathbf{g}_t - \mathbf{g}_{t-1}\|^2 + 8G^4 S_T^{\mathbf{x}} + 8G^4 S_{T,k^\star,i^\star}^{\mathbf{x}}$$

$$\leq C_5 F_T^{\mathbf{x}} + 8G^4 S_T^{\mathbf{x}} + 8G^4 S_{T,k^\star,i^\star}^{\mathbf{x}}, \qquad \text{(by (E.3))}$$

where the first step by using the definition of $\nabla h_{t,i}^{\exp}(\mathbf{x}) = \mathbf{g}_t + \alpha_i \mathbf{g}_t \mathbf{g}_t^\top (\mathbf{x} - \mathbf{x}_t)$, the third step uses the assumption that $\alpha \in [1/T, 1]$ without loss of generality, and the last step sets $C_5 = 16L(2 + 16D^2 G^2)$. Consequently, it holds that

$$\ln\left(1 + \frac{\alpha_{i^\star}}{8\gamma d} \bar{V}_{T,k^\star,i^\star}\right) \leq \ln\left(1 + \frac{\alpha_{i^\star} C_5}{8\gamma d} F_T^{\mathbf{x}}\right) + \frac{\alpha_{i^\star} G^4}{\gamma d} S_T^{\mathbf{x}} + \frac{\alpha_{i^\star} G^4}{\gamma d} S_{T,k^\star,i^\star}^{\mathbf{x}}.$$

Plugging the above result and due to $\alpha_{i^\star} \leq \alpha \leq 2\alpha_{i^\star}$, the base regret can be bounded by

$$\text{BASE-REG} \leq \frac{32d}{\alpha} \ln\left(1 + \frac{\alpha C_5}{8\gamma d} F_T^{\mathbf{x}}\right) + \frac{16G^4}{\gamma} S_T^{\mathbf{x}} + \left(\frac{16G^4}{\gamma} - \frac{\gamma}{4}\right) S_{T,k^\star,i^\star}^{\mathbf{x}} + \frac{1}{2}\gamma D^2.$$

Combining the meta regret (C.1) and base regret, the overall regret can be bounded by

$$\text{REG}_T \leq \frac{512}{\alpha} \ln \frac{2^{20} N}{\alpha^2} + 2C_0 \ln(4C_0^2 N) + \frac{32d}{\alpha} \ln\left(1 + \frac{\alpha C_5}{8\gamma d} F_T^{\mathbf{x}}\right)$$

$$+ \left( \frac{32D^2G^4}{\gamma} - \frac{C_0}{2} \right) S_T^{\boldsymbol{q}} + \left( \frac{32G^4}{\gamma} - \lambda_1 \right) \sum_{t=2}^{T} \sum_{k=1}^{K} q_{t,k} \|\mathbf{x}_{t,k} - \mathbf{x}_{t-1,k}\|^2$$

$$+ \left( \lambda_2 + \frac{16G^4}{\gamma} - \frac{\gamma}{4} \right) S_{T,k^\star,i^\star}^{\mathbf{x}}$$

$$\leq \mathcal{O} \left( \frac{d}{\alpha} \ln F_T^{\mathbf{x}} \right) \qquad \text{(requiring } C_0 \geq 64D^2G^4/\gamma,\ \lambda_1 \geq 32G^4/\gamma,\ \gamma \geq 8G^2 + \lambda_2)$$

$$\leq \mathcal{O} \left( \frac{d}{\alpha} \ln F_T \right),$$

where the last uses the following lemma.

**Lemma 4** (Corollary 5 of Orabona et al. [2012])**.** *If* $a, b, c, d, x > 0$ *satisfy* $x - d \leq a \ln(bx + c)$,

$$x - d \leq a \ln \left( 2ab \ln \frac{2ab}{e} + 2bd + 2c \right).$$

**Strongly Convex Functions.** The meta regret can be bounded the same as Theorem 1 (especially (C.3)), and thus we directly move on to the base regret. For simplicity, we denote by $\mathbf{g}_t \triangleq \nabla f_t(\mathbf{x}_t)$ and give a different decomposition for $\bar{V}_{T,k^\star,i^\star} = \sum_{t=2}^{T} \|\nabla h_{t,i^\star}^{\mathrm{sc}}(\mathbf{x}_{t,k^\star,i^\star}) - \nabla h_{t-1,i^\star}^{\mathrm{sc}}(\mathbf{x}_{t-1,k^\star,i^\star})\|^2$, the empirical gradient variation of the base learner.

$$\bar{V}_{T,k^\star,i^\star} = \sum_{t=2}^{T} \|(\mathbf{g}_t + \lambda_{i^\star}(\mathbf{x}_{t,k^\star,i^\star} - \mathbf{x}_t)) - (\mathbf{g}_{t-1} + \lambda_{i^\star}(\mathbf{x}_{t-1,k^\star,i^\star} - \mathbf{x}_{t-1}))\|^2$$

$$\leq 2 \sum_{t=2}^{T} \|\mathbf{g}_t - \mathbf{g}_{t-1}\|^2 + 2\lambda_{i^\star}^2 \sum_{t=2}^{T} \|(\mathbf{x}_{t,k^\star,i^\star} - \mathbf{x}_t) - (\mathbf{x}_{t-1,k^\star,i^\star} - \mathbf{x}_{t-1})\|^2$$

$$\leq 32LF_T^{\mathbf{x}} + 4S_T^{\mathbf{x}} + 4S_{T,k^\star,i^\star}^{\mathbf{x}}, \qquad \text{(by (E.3) and } \lambda \in [1/T, 1])$$

where the first step uses the definition of $\nabla h_{t,i}^{\mathrm{sc}}(\mathbf{x}) = \mathbf{g}_t + \lambda_i(\mathbf{x} - \mathbf{x}_t)$. Consequently,

$$\frac{16G_{\mathrm{sc}}^2}{\lambda_{i^\star}} \ln \left( 1 + \lambda_{i^\star} \bar{V}_{T,k^\star,i^\star} \right) \leq \frac{16G_{\mathrm{sc}}^2}{\lambda_{i^\star}} \ln(1 + 32\lambda_{i^\star} LF_T^{\mathbf{x}}) + 64G_{\mathrm{sc}}^2(S_T^{\mathbf{x}} + S_{T,k^\star,i^\star}^{\mathbf{x}}).$$

Plugging the above result and due to $\lambda_{i^\star} \leq \lambda \leq 2\lambda_{i^\star}$, the base regret can be bounded by

$$\textsc{Base-Reg} \leq \frac{32G_{\mathrm{sc}}^2}{\lambda} \ln(1 + 32\lambda LF_T^{\mathbf{x}}) + 64G_{\mathrm{sc}}^2 S_T^{\mathbf{x}} + \left( 64G_{\mathrm{sc}}^2 - \frac{\gamma}{8} \right) S_{T,k^\star,i^\star}^{\mathbf{x}} + \frac{1}{4}\gamma D^2.$$

Combining the meta regret (C.3) and base regret, the overall regret can be bounded by

$$\textsc{Reg}_T \leq \frac{512G^2}{\lambda} \ln \frac{2^{20} NG^4}{\lambda^2} + 2C_0 \ln(4C_0^2 N) + \frac{32G_{\mathrm{sc}}^2}{\lambda} \ln(1 + 32\lambda LF_T^{\mathbf{x}})$$

$$+ \left( 128D^2G_{\mathrm{sc}}^2 - \frac{C_0}{2} \right) S_T^{\boldsymbol{q}} + \left( 128G_{\mathrm{sc}}^2 - \lambda_1 \right) \sum_{t=2}^{T} \sum_{k=1}^{K} q_{t,k} \|\mathbf{x}_{t,k} - \mathbf{x}_{t-1,k}\|^2$$

$$+ \left( \lambda_2 + 64G_{\mathrm{sc}}^2 - \frac{\gamma}{8} \right) S_{T,k^\star,i^\star}^{\mathbf{x}}$$

$$\leq \mathcal{O} \left( \frac{1}{\lambda} \ln F_T^{\mathbf{x}} \right) \qquad \text{(requiring } C_0 \geq 256D^2G_{\mathrm{sc}}^2,\ \lambda_1 \geq 128G_{\mathrm{sc}}^2,\ \gamma \geq 8\lambda_2 + 512G_{\mathrm{sc}}^2)$$

$$\leq \mathcal{O} \left( \frac{1}{\lambda} \ln F_T \right). \qquad \text{(by Lemma 4)}$$

**Convex Functions.** We first give a different analysis for $V_\star \triangleq \sum_{t=1}^{T}(\ell_{t,k^\star,i^\star} - m_{t,k^\star,i^\star})^2$. From Lemma 1, it holds that

$$V_\star \leq 2D^2 \sum_{t=1}^{T} \|\nabla f_t(\mathbf{x}_t) - \nabla f_{t-1}(\mathbf{x}_{t-1})\|^2 + 2G^2 \sum_{t=1}^{T} \|\mathbf{x}_{t,k^\star,i^\star} - \mathbf{x}_{t-1,k^\star,i^\star} + \mathbf{x}_{t-1} - \mathbf{x}_t\|^2$$

$$\leq 32D^2 L F_T^{\mathbf{x}} + 4G^2 S_{T,k^\star,i^\star}^{\mathbf{x}} + 4G^2 S_T^{\mathbf{x}}. \tag{by (E.3)}$$

Plugging the above analysis back to the meta regret, we obtain

$$\text{META-REG} \leq \frac{1}{\eta_{k^\star}} \ln \frac{N}{\eta_{k^\star}^2} + 1024\eta_{k^\star} D^2 L F_T^{\mathbf{x}} + 2G^2 S_T^{\mathbf{x}} + (\lambda_2 + 2G^2) S_{T,k^\star,i^\star}^{\mathbf{x}}$$
$$- \frac{C_0}{2} S_T^{\boldsymbol{q}} - \lambda_1 \sum_{t=2}^T \sum_{k=1}^K q_{t,k} \|\mathbf{x}_{t,k} - \mathbf{x}_{t-1,k}\|^2.$$

For the base regret, using Lemma 10 and (E.3), it holds that

$$\text{BASE-REG} \leq 20D\sqrt{L F_T^{\mathbf{x}}} + \gamma D^2 - \frac{\gamma}{4} S_{T,k^\star,i^\star}^{\mathbf{x}} + \mathcal{O}(1).$$

Combining the meta regret and base regret, the overall regret can be bounded by

$$\text{REG}_T \leq \frac{1}{\eta_{k^\star}} \ln \frac{N}{\eta_{k^\star}^2} + 1024\eta_{k^\star} D^2 L F_T^{\mathbf{x}} + 20D\sqrt{L F_T^{\mathbf{x}}} + \left(\lambda_2 + 2G^2 - \frac{\gamma}{4}\right) S_{T,k^\star,i^\star}^{\mathbf{x}}$$
$$+ \left(4D^2 G^2 - \frac{C_0}{2}\right) S_T^{\boldsymbol{q}} + \left(4G^2 - \lambda_1\right) \sum_{t=2}^T \sum_{k=1}^K q_{t,k} \|\mathbf{x}_{t,k} - \mathbf{x}_{t-1,k}\|^2$$
$$\leq \frac{1}{\eta_{k^\star}} \ln \frac{N}{\eta_{k^\star}^2} + 1024\eta_{k^\star} D^2 L F_T^{\mathbf{x}} + 20D\sqrt{L F_T^{\mathbf{x}}} \leq \frac{1}{\eta_{k^\star}} \ln \frac{Ne^5}{\eta_{k^\star}^2} + 1044\eta_{k^\star} D^2 L F_T^{\mathbf{x}}$$
$$\leq \frac{2}{\eta_{k^\star}} \ln \frac{Ne^5}{\eta_{k^\star}^2} + 2088\eta_{k^\star} D^2 L F_T$$
$$\leq 4C_0 \ln(4C_0^2 N) + \mathcal{O}\left(\sqrt{F_T \ln F_T}\right), \tag{by Lemma 7}$$

where the second step holds by requiring $C_0 \geq 8D^2 G^2$, $\lambda_1 \geq 4G^2$ and $\gamma \geq 4\lambda_2 + 8G^2$, and the third step uses AM-GM inequality: $\sqrt{xy} \leq \frac{ax}{2} + \frac{y}{2a}$ for any $x, y, a > 0$ with $a = 1/(2D\eta_{k^\star})$. The fourth step is by requiring $1 - 1044\eta_{k^\star} D^2 L \geq 1/2$, i.e., $\eta_k \leq 1/(2088D^2 L)$ for any $k \in [K]$, which can be satisfied by requiring $C_0 \geq 1044D^2 L$.

**Overall.** At last, we determine the specific values of $C_0, \lambda_1$ and $\lambda_2$. These parameters need to satisfy the following requirements:

$$C_0 \geq 1, \ C_0 \geq 4\lambda_1 D^2, \ C_0 \geq 128, \ C_0 \geq \frac{64D^2 G^4}{8G^2 + \lambda_2}, \ C_0 \geq 256D^2 G_{\text{sc}}^2, \ C_0 \geq 8D^2 G^2,$$
$$C_0 \geq 1044D^2 L, \ \lambda_1 \geq \frac{32G^4}{8G^2 + \lambda_2}, \ \lambda_1 \geq 128G_{\text{sc}}^2, \ \lambda_1 \geq 4G^2, \ \lambda_2 \geq 2\lambda_1.$$

As a result, we set $C_0, \lambda_1, \lambda_2$ to be the minimum constants satisfying the above conditions. Besides, since the absolute values of the surrogate losses and the optimisms are bounded by problem-independent constants, as shown in the proof of Theorem 1, rescaling them to $[-1, 1]$ only add a constant multiplicative factors on the final result. $\square$

## D  Proofs for Appendix A

This section provides the proofs for Appendix A, including Theorem 3 and Theorem 4.

### D.1  Proof of Theorem 3

A key result in the work of Chen et al. [2023b] is the following decomposition for the empirical gradient variation $\bar{V}_T \triangleq \sum_{t=2}^T \|\nabla f_t(\mathbf{x}_t) - \nabla f_{t-1}(\mathbf{x}_{t-1})\|^2$, restated as follows.

**Lemma 5** (Lemma 4 of Chen et al. [2023b])**.** *Under the same assumptions as Chen et al. [2023b],*

$$\mathbb{E}\left[\sum_{t=2}^T \|\nabla f_t(\mathbf{x}_t) - \nabla f_{t-1}(\mathbf{x}_{t-1})\|^2\right] \leq 4L^2 \sum_{t=2}^T \|\mathbf{x}_t - \mathbf{x}_{t-1}\|^2 + 8\sigma_{1:T}^2 + 4\Sigma_{1:T}^2 + \mathcal{O}(1).$$

*Proof of Theorem 3.* The analysis is almost the same as the proof of Theorem 2. Some abbreviations of the stability terms are defined in (B.1).

**Exp-concave Functions.** The meta regret remains the same as (C.1) and the only difference is a slightly different decomposition of the empirical gradient variation of the base learner, i.e., $\bar{V}_{T,k^\star,i^\star}$ in (C.2). Specifically, it holds that

$$
\bar{V}_{T,k^\star,i^\star} = \sum_{t=2}^{T} \|(\mathbf{g}_t + \alpha_{i^\star}\mathbf{g}_t\mathbf{g}_t^\top(\mathbf{x}_{t,k^\star,i^\star} - \mathbf{x}_t)) - (\mathbf{g}_{t-1} + \alpha_{i^\star}\mathbf{g}_{t-1}\mathbf{g}_{t-1}^\top(\mathbf{x}_{t-1,k^\star,i^\star} - \mathbf{x}_{t-1}))\|^2
$$

$$
\leq (2 + 16D^2G^2)\sum_{t=2}^{T}\|\mathbf{g}_t - \mathbf{g}_{t-1}\|^2 + 8G^4 S_T^{\mathbf{x}} + 8G^4 S_{T,k^\star,i^\star}^{\mathbf{x}}
$$

$$
\leq 2C_5\sigma_{1:T}^2 + C_5\Sigma_{1:T}^2 + C_6 S_T^{\mathbf{x}} + 8G^4 S_{T,k^\star,i^\star}^{\mathbf{x}},
$$

where $C_5 = 8(1+8D^2G^2)$, $C_6 = 8L^2 + 64D^2G^2L^2 + 8G^4$ and the last step is by taking expectation on both sides and Lemma 5, leading to the following upper bound of the base algorithm:

$$
\mathbb{E}[\textsc{Base-Reg}] \leq \mathcal{O}\left(\frac{d}{\alpha}\ln\left(\sigma_{1:T}^2 + \Sigma_{1:T}^2\right)\right) + \frac{2C_6}{\gamma}S_T^{\mathbf{x}} + \left(\frac{16G^4}{\gamma} - \frac{\gamma}{4}\right)S_{T,k^\star,i^\star}^{\mathbf{x}} + \frac{1}{2}\gamma D^2.
$$

Combining the meta regret and base regret, the overall regret can be bounded by

$$
\mathbb{E}[\textsc{Reg}_T] \leq \frac{512}{\alpha}\ln\frac{2^{20}N}{\alpha^2} + 2C_0\ln(4C_0^2N) + \mathcal{O}\left(\frac{d}{\alpha}\ln\left(\sigma_{1:T}^2 + \Sigma_{1:T}^2\right)\right)
$$

$$
+ \left(\frac{4C_6D^2}{\gamma} - \frac{C_0}{2}\right)S_T^{\boldsymbol{q}} + \left(\frac{4C_6}{\gamma} - \lambda_1\right)\sum_{t=2}^{T}\sum_{k=1}^{K}q_{t,k}\|\mathbf{x}_{t,k} - \mathbf{x}_{t-1,k}\|^2
$$

$$
+ \left(\lambda_2 + \frac{16G^4}{\gamma} - \frac{\gamma}{4}\right)S_{T,k^\star,i^\star}^{\mathbf{x}} + \frac{1}{2}\gamma D^2 \leq \mathcal{O}\left(\frac{d}{\alpha}\ln(\sigma_{1:T}^2 + \Sigma_{1:T}^2)\right),
$$

where the last step holds by requiring $C_0 \geq 8C_6D^2/\gamma$, $\lambda_1 \geq 4C_6/\gamma$ and $\gamma \geq \lambda_2 + 8G^2$.

**Strongly Convex Functions.** The meta regret remains the same as (C.3) and the only difference is a slightly different decomposition of the empirical gradient variation of the base learner:

$$
\mathbb{E}\left[\|\nabla h_{t,i^\star}^{\mathrm{sc}}(\mathbf{x}_{t,k^\star,i^\star}) - \nabla h_{t-1,i^\star}^{\mathrm{sc}}(\mathbf{x}_{t-1,k^\star,i^\star})\|^2\right]
$$

$$
= \mathbb{E}\left[\|(\mathbf{g}_t + \lambda_{i^\star}(\mathbf{x}_{t,k^\star,i^\star} - \mathbf{x}_t)) - (\mathbf{g}_{t-1} + \lambda_{i^\star}(\mathbf{x}_{t-1,k^\star,i^\star} - \mathbf{x}_{t-1}))\|^2\right]
$$

$$
\leq 2\mathbb{E}\left[\|\mathbf{g}_t - \mathbf{g}_{t-1}\|^2\right] + 2\lambda_{i^\star}^2\|(\mathbf{x}_{t,k^\star,i^\star} - \mathbf{x}_t) - (\mathbf{x}_{t-1,k^\star,i^\star} - \mathbf{x}_{t-1})\|^2
$$

$$
\leq 16\sigma_t^2 + 8\Sigma_t^2 + (4 + 8L^2)\|\mathbf{x}_t - \mathbf{x}_{t-1}\|^2 + 4\|\mathbf{x}_{t,k^\star,i^\star} - \mathbf{x}_{t-1,k^\star,i^\star}\|^2,
$$

where $\sigma_t^2 \triangleq \max_{\mathbf{x}\in\mathcal{X}}\mathbb{E}_{f_t\sim\mathcal{D}_t}[\|\nabla f_t(\mathbf{x}) - \nabla F_t(\mathbf{x})\|^2]$, $\Sigma_t^2 \triangleq \mathbb{E}[\sup_{\mathbf{x}\in\mathcal{X}}\|\nabla F_t(\mathbf{x}) - \nabla F_{t-1}(\mathbf{x})\|^2]$, and the last step is by taking expectation on both sides and Lemma 5. Plugging the above empirical gradient variation decomposition into Lemma 12, we aim to control the following term:

$$
\mathbb{E}\left[\sum_{t=2}^{T}\frac{1}{\lambda_{i^\star}t}\|\nabla h_{t,i^\star}^{\mathrm{sc}}(\mathbf{x}_{t,k^\star,i^\star}) - \nabla h_{t-1,i^\star}^{\mathrm{sc}}(\mathbf{x}_{t-1,k^\star,i^\star})\|^2\right]
$$

$$
\leq 8\sum_{t=2}^{T}\frac{2\sigma_t^2 + \Sigma_t^2}{\lambda_{i^\star}t} + 4\sum_{t=2}^{T}\frac{(1 + 2L^2)\|\mathbf{x}_t - \mathbf{x}_{t-1}\|^2 + \|\mathbf{x}_{t,k^\star,i^\star} - \mathbf{x}_{t-1,k^\star,i^\star}\|^2}{\lambda_{i^\star}t}
$$

$$
\leq \mathcal{O}\left(\frac{1}{\lambda}(\sigma_{\max}^2 + \Sigma_{\max}^2)\ln(\sigma_{1:T}^2 + \Sigma_{1:T}^2)\right) + \frac{8D^2(L^2 + 1)}{\lambda_{i^\star}}\ln(1 + \lambda_{i^\star}(1 + 2L^2)S_T^{\mathbf{x}} + \lambda_{i^\star}S_{T,k^\star,i^\star}^{\mathbf{x}}).
$$

The last step is due to Lemma 9, which is a generalization of Lemma 5 of Chen et al. [2023b]. Combining the meta regret and base regret, the overall regret can be bounded by

$$
\mathbb{E}[\textsc{Reg}_T] \leq \frac{512G^2}{\lambda}\ln\frac{2^{20}NG^4}{\lambda^2} + 2C_0\ln(4C_0^2N) + \mathcal{O}\left(\frac{1}{\lambda}(\sigma_{\max}^2 + \Sigma_{\max}^2)\ln(\sigma_{1:T}^2 + \Sigma_{1:T}^2)\right)
$$

$$
+ \left(2C_7D^2 - \frac{C_0}{2}\right)S_T^{\boldsymbol{q}} + (2C_7 - \lambda_1)\sum_{t=2}^{T}\sum_{k=1}^{K}q_{t,k}\|\mathbf{x}_{t,k} - \mathbf{x}_{t-1,k}\|^2
$$

$$+ \left( \lambda_2 + 32D^2(L^2+1) - \frac{\gamma}{8} \right) S^{\mathbf{x}}_{T,k^\star,i^\star} + \frac{1}{4}\gamma D^2$$

$$\leq \mathcal{O}\left( \frac{1}{\lambda}(\sigma^2_{\max} + \Sigma^2_{\max})\ln(\sigma^2_{1:T} + \Sigma^2_{1:T}) \right),$$

where $C_7 = 64D^2(L^2+1)^2$ and the last step holds by requiring $C_0 \geq 4C_7D^2$, $\lambda_1 \geq 2C_7$ and $\gamma \geq 8\lambda_2 + 256D^2(L^2+1)$.

**Convex Functions.** We first give a slightly different decomposition for $V_\star \triangleq \sum_t (\ell_{t,k^\star,i^\star} - m_{t,k^\star,i^\star})^2$. Starting from Lemma 5, it holds that

$$V_\star \leq 2D^2 \sum_{t=1}^{T} \|\nabla f_t(\mathbf{x}_t) - \nabla f_{t-1}(\mathbf{x}_{t-1})\|^2 + 2G^2 \sum_{t=1}^{T} \|\mathbf{x}_{t,k^\star,i^\star} - \mathbf{x}_{t-1,k^\star,i^\star} + \mathbf{x}_{t-1} - \mathbf{x}_t\|^2$$

$$\leq 8D^2(2\sigma^2_{1:T} + \Sigma^2_{1:T}) + (8D^2L^2 + 4G^2)S^{\mathbf{x}}_T + 4G^2 S^{\mathbf{x}}_{T,k^\star,i^\star} + \mathcal{O}(1),$$

where the last step by taking expectation on both sides. Following the same proof as in Theorem 2, taking expectation on both sides, the expected meta regret can be bounded by

$$\mathbb{E}[\text{META-REG}] \leq \frac{1}{\eta_{k^\star}} \ln \frac{N}{\eta^2_{k^\star}} + 128D^2 \eta_{k^\star}(2\sigma^2_{1:T} + \Sigma^2_{1:T}) + 128(2D^2L^2 + G^2)S^{\mathbf{x}}_T$$

$$+ (\lambda_2 + 128G^2)S^{\mathbf{x}}_{T,k^\star,i^\star} - \frac{C_0}{2}S^{\boldsymbol{q}}_T - \lambda_1 \sum_{t=2}^{T}\sum_{k=1}^{K} q_{t,k}\|\mathbf{x}_{t,k} - \mathbf{x}_{t-1,k}\|^2$$

$$\leq 2C_0 \ln(4C_0^2 N) + \mathcal{O}\left( \sqrt{(\sigma^2_{1:T} + \Sigma^2_{1:T})\ln(\sigma^2_{1:T} + \Sigma^2_{1:T})} \right)$$

$$+ 128(2D^2L^2 + G^2)S^{\mathbf{x}}_T + (\lambda_2 + 128G^2)S^{\mathbf{x}}_{T,k^\star,i^\star} - \frac{C_0}{2}S^{\boldsymbol{q}}_T - \lambda_1 \sum_{t=2}^{T}\sum_{k=1}^{K} q_{t,k}\|\mathbf{x}_{t,k} - \mathbf{x}_{t-1,k}\|^2,$$

where the last step is due to Lemma 7 by requiring $C_0 \geq 8D$.

For the base regret, the only difference is a slightly different decomposition of the empirical gradient variation of the base learner, i.e., $\bar{V}_{T,k^\star,i^\star}$ in (C.5). Specifically, using Lemma 5, it holds that

$$\mathbb{E}[\text{BASE-REG}] \leq 10D\sqrt{(2\sigma^2_{1:T} + \Sigma^2_{1:T})} + 10DLS^{\mathbf{x}}_T + \gamma D^2 - \frac{\gamma}{4}S^{\mathbf{x}}_{T,k^\star,i^\star} + \mathcal{O}(1).$$

Combining the meta regret and base regret, the overall regret can be bounded by

$$\mathbb{E}[\text{REG}_T] \leq \mathcal{O}\left( \sqrt{(\sigma^2_{1:T} + \Sigma^2_{1:T})\ln(\sigma^2_{1:T} + \Sigma^2_{1:T})} \right) + 2C_0 \ln(4C_0^2 N)$$

$$+ \left( 2D^2 C_9 - \frac{C_0}{2} \right) S^{\boldsymbol{q}}_T + (2C_9 - \lambda_1) \sum_{t=2}^{T}\sum_{k=1}^{K} q_{t,k}\|\mathbf{x}_{t,k} - \mathbf{x}_{t-1,k}\|^2$$

$$+ \left( \lambda_2 + 128G^2 - \frac{\gamma}{4} \right) S^{\mathbf{x}}_{T,k^\star,i^\star} + \gamma D^2$$

$$\leq \mathcal{O}\left( \sqrt{(\sigma^2_{1:T} + \Sigma^2_{1:T})\ln(\sigma^2_{1:T} + \Sigma^2_{1:T})} \right)$$

where $C_9 = 128(2D^2L^2 + G^2) + 10DL$ and the last step holds by requiring $C_0 \geq 4D^2 C_9$, $\lambda_1 \geq 2C_9$ and $\gamma \geq 4\lambda_2 + 512G^2$.

**Overall.** At last, we determine the specific values of $C_0$, $\lambda_1$ and $\lambda_2$. These parameters need to satisfy the following requirements:

$$C_0 \geq 1, \ C_0 \geq 4\lambda_1 D^2, \ C_0 \geq 128, \ C_0 \geq \frac{8C_6 D^2}{8G^2 + \lambda_2}, \ C_0 \geq 4C_7 D^2, \ C_0 \geq \frac{4D^2 C_8}{16 + \lambda_2}, \ C_0 \geq 8D,$$

$$C_0 \geq 4D^2 C_9, \ \lambda_1 \geq \frac{4C_6}{8G^2 + \lambda_2}, \ \lambda_1 \geq 2C_7, \ \lambda_1 \geq \frac{2C_8}{16 + \lambda_2}, \ \lambda_1 \geq 2C_9, \ \lambda_1 \geq 4G^2, \ \lambda_2 \geq 2\lambda_1.$$

As a result, we set $C_0, \lambda_1, \lambda_2$ to be the minimum constants satisfying the above conditions. Besides, since the absolute values of the surrogate losses and the optimisms are bounded by problem-independent constants, as shown in the proof of Theorem 1, rescaling them to $[-1,1]$ only add a constant multiplicative factors on the final result. $\square$

## D.2 Proof of Theorem 4

*Proof.* The proof of the dishonest case is straightforward by directly applying Theorem 2. In the following, we mainly focus on the honest case. Consider a bilinear game of $f(\mathbf{x}, \mathbf{y}) \triangleq \mathbf{x}^\top A \mathbf{y}$ and denote by $\mathbf{g}_t^{\mathbf{x}} \triangleq A\mathbf{y}_t$, $\mathbf{g}_t^{\mathbf{y}} \triangleq A\mathbf{x}_t$ the gradients received by the $\mathbf{x}$-player and $\mathbf{y}$-player. The only difference from the proof of Theorem 2 is that the gradient variation can be now decomposed

$$\|\mathbf{g}_t^{\mathbf{x}} - \mathbf{g}_{t-1}^{\mathbf{x}}\|^2 = \|A\mathbf{y}_t - A\mathbf{y}_{t-1}\|^2 \le \|\mathbf{y}_t - \mathbf{y}_{t-1}\|^2,$$

where the last step holds under the mild assumption of $\|A\| \le 1$. The gradient variation of the $\mathbf{x}$-player is associated with the stability term of the $\mathbf{y}$-player. When summing the regret of the players, the negative terms in the $\mathbf{x}$-player's algorithm can be leveraged to cancel the gradient variation of the $\mathbf{y}$-player and vise versa. As a result, all gradient variations can be canceled and the summation of regret is bounded by $\mathcal{O}(1)$. As for strongly convex-concave games, since it is a special case of the bilinear games, the above derivations still hold, which completes the proof. $\square$

## E Supporting Lemmas

In this section, we present several supporting lemmas used in proving our theoretical results. In Appendix E.1, we provide useful lemmas for the decomposition of two combined decisions and the parameter tuning. And in Appendix E.2, we analyze the stability of the base algorithms for different kinds of loss functions.

### E.1 Useful Lemmas

In this part, we conclude some useful lemmas for bounding the gap between two combined decisions (Lemma 6), tuning the parameter (Lemma 7 and Lemma 8), and a useful summation (Lemma 9).

**Lemma 6.** *Under Assumption 1, if* $\mathbf{x} = \sum_{i=1}^N p_i \mathbf{x}_i, \mathbf{y} = \sum_{i=1}^N q_i \mathbf{y}_i$, *where* $\boldsymbol{p}, \boldsymbol{q} \in \Delta_N, \mathbf{x}_i, \mathbf{y}_i \in \mathcal{X}$ *for any* $i \in [N]$, *then it holds that*

$$\|\mathbf{x} - \mathbf{y}\|^2 \le 2 \sum_{i=1}^N p_i \|\mathbf{x}_i - \mathbf{y}_i\|^2 + 2D^2 \|\boldsymbol{p} - \boldsymbol{q}\|_1^2.$$

**Lemma 7.** *For a step size pool of* $\mathcal{H}_\eta = \{\eta_k\}_{k \in [K]}$, *where* $\eta_1 = \frac{1}{2C_0} \ge \ldots \ge \eta_K = \frac{1}{2C_0 T}$, *if* $C_0 \ge \frac{\sqrt{X}}{2T}$, *there exists* $\eta \in \mathcal{H}_\eta$ *such that*

$$\frac{1}{\eta} \ln \frac{Y}{\eta^2} + \eta X \le 2C_0 \ln(4YC_0^2) + 4\sqrt{X \ln(4XY)}.$$

**Lemma 8.** *Denoting by* $\eta_\star$ *the optimal step size, for a step size pool of* $\mathcal{H}_\eta = \{\eta_k\}_{k \in [K]}$, *where* $\eta_1 = \frac{1}{2C_0} \ge \ldots \ge \eta_K = \frac{1}{2C_0 T}$, *if* $C_0 \ge \frac{1}{2\eta_\star T}$, *there exists* $\eta \in \mathcal{H}_\eta$ *such that*

$$\frac{1}{\eta} \ln \frac{Y}{\eta^2} \le 2C_0 \ln(4YC_0^2) + \frac{2}{\eta_\star} \ln \frac{4Y}{\eta_\star^2}.$$

**Lemma 9.** *For a sequence of* $\{a_t\}_{t=1}^T$ *and* $b$, *where* $a_t, b > 0$ *for any* $t \in [T]$, *denoting by* $a_{\max} \triangleq \max_t a_t$ *and* $A \triangleq \lceil b \sum_{t=1}^T a_t \rceil$, *we have*

$$\sum_{t=1}^T \frac{a_t}{bt} \le \frac{a_{\max}}{b}(1 + \ln A) + \frac{1}{b^2}.$$

*Proof of Lemma 6.* The term of $\|\mathbf{x} - \mathbf{y}\|^2$ can be decomposed as follows:

$$\|\mathbf{x} - \mathbf{y}\|^2 = \left\| \sum_{i=1}^N p_i \mathbf{x}_i - \sum_{i=1}^N q_i \mathbf{y}_i \right\|^2 = \left\| \sum_{i=1}^N p_i \mathbf{x}_i - \sum_{i=1}^N p_i \mathbf{y}_i + \sum_{i=1}^N p_i \mathbf{y}_i - \sum_{i=1}^N q_i \mathbf{y}_i \right\|^2$$

$$\le 2 \left\| \sum_{i=1}^N p_i(\mathbf{x}_i - \mathbf{y}_i) \right\|^2 + 2 \left\| \sum_{i=1}^N (p_i - q_i)\mathbf{y}_i \right\|^2$$

$$\leq 2 \left( \sum_{i=1}^{N} p_i \|\mathbf{x}_i - \mathbf{y}_i\| \right)^2 + 2 \left( \sum_{i=1}^{N} |p_i - q_i| \|\mathbf{y}_i\| \right)^2$$

$$\leq 2 \sum_{i=1}^{N} p_i \|\mathbf{x}_i - \mathbf{y}_i\|^2 + 2D^2 \|\boldsymbol{p} - \boldsymbol{q}\|_1^2,$$

where the first inequality is due to $(a+b)^2 \leq 2a^2 + 2b^2$ for any $a, b \in \mathbb{R}$, and the last step is due to Cauchy-Schwarz inequality, Assumption 1 and the definition of $\ell_1$-norm, finishing the proof. $\qquad \square$

*Proof of Lemma 7.* Denoting the optimal step size by $\eta_\star \triangleq \sqrt{\ln(4XY)/X}$, if the optimal step size satisfies $\eta \leq \eta_\star \leq 2\eta$, where $\eta \leq \eta_\star$ can be guaranteed if $C_0 \geq \frac{\sqrt{X}}{2T}$, then it holds that

$$\frac{1}{\eta} \ln \frac{Y}{\eta^2} + \eta X \leq \frac{2}{\eta_\star} \ln \frac{4Y}{\eta_\star^2} + \eta_\star X \leq 3\sqrt{X \ln(4XY)}.$$

Otherwise, if the optimal step size is greater than the maximum step size in the parameter pool, i.e., $\eta_\star \geq (\eta = \eta_1 = \frac{1}{2C_0})$, then we have

$$\frac{1}{\eta} \ln \frac{Y}{\eta^2} + \eta X \leq \frac{1}{\eta} \ln \frac{Y}{\eta^2} + \eta_\star X \leq 2C_0 \ln(4YC_0^2) + \sqrt{X \ln(4XY)}.$$

Overall, it holds that

$$\frac{1}{\eta} \ln \frac{Y}{\eta^2} + \eta X \leq 2C_0 \ln(4YC_0^2) + 4\sqrt{X \ln(4XY)},$$

which completes the proof. $\qquad \square$

*Proof of Lemma 8.* The proof follows the same flow as Lemma 7. $\qquad \square$

*Proof of Lemma 9.* This result is inspired by Lemma 5 of Chen et al. [2023b], and we generalize it to arbitrary variables for our purpose. Specifically, we consider two cases: $A < T$ and $A \geq T$. For the first case, if $A < T$, it holds that

$$\sum_{t=1}^{T} \frac{a_t}{bt} = \sum_{t=1}^{A} \frac{a_t}{bt} + \sum_{A+1}^{T} \frac{a_t}{bt} \leq \frac{a_{\max}}{b} \sum_{t=1}^{A} \frac{1}{t} + \frac{1}{b(A+1)} \sum_{A+1}^{T} a_t \leq \frac{a_{\max}}{b}(1 + \ln A) + \frac{1}{b^2},$$

where the last step is due to $\sum_{A+1}^{T} a_t \leq \sum_{t=1}^{T} a_t \leq A/b$. The case of $A < T$ can be proved similarly, which finishes the proof. $\qquad \square$

## E.2 Stability Analysis of Base Algorithms

In this part, we analyze the negative stability terms in optimism OMD analysis, for convex, exp-concave and strongly convex functions. For simplicity, we define the *empirical gradient variation*:

$$\bar{V}_T \triangleq \sum_{t=2}^{T} \|\mathbf{g}_t - \mathbf{g}_{t-1}\|^2, \text{ where } \mathbf{g}_t \triangleq \nabla f_t(\mathbf{x}_t). \tag{E.1}$$

In the following, we provide two kinds of decompositions that are used in the analysis of gradient-variation and small-loss guarantees, respectively. First, using smoothness (Assumption 2), we have

$$\bar{V}_T = \sum_{t=2}^{T} \|\mathbf{g}_t - \nabla f_{t-1}(\mathbf{x}_t) + \nabla f_{t-1}(\mathbf{x}_t) - \mathbf{g}_{t-1}\|^2 \leq 2V_T + 2L^2 \sum_{t=2}^{T} \|\mathbf{x}_t - \mathbf{x}_{t-1}\|^2, \tag{E.2}$$

Second, for an $L$-smooth and non-negative function $f : \mathcal{X} \mapsto \mathbb{R}_+$, $\|\nabla f(\mathbf{x})\|_2^2 \leq 4Lf(\mathbf{x})$ holds for any $\mathbf{x} \in \mathcal{X}$ [Srebro et al., 2010, Lemma 3.1]. It holds that

$$\bar{V}_T = \sum_{t=2}^{T} \|\mathbf{g}_t - \mathbf{g}_{t-1}\|^2 \leq 2 \sum_{t=2}^{T} \|\mathbf{g}_t\|^2 + 2 \sum_{t=2}^{T} \|\mathbf{g}_{t-1}\|^2 \leq 4 \sum_{t=1}^{T} \|\mathbf{g}_t\|^2 \leq 16L \sum_{t=1}^{T} f_t(\mathbf{x}_t). \tag{E.3}$$

Next we provide the regret analysis in terms of the empirical gradient-variation $\bar{V}_T$, for convex (Lemma 10), exp-concave (Lemma 11), and strongly convex (Lemma 12) functions.

**Lemma 10.** *Under Assumptions 1 and 2, if the loss functions are convex, optimistic OGD with the following update rule:*

$$\mathbf{x}_t = \Pi_{\mathcal{X}}[\widehat{\mathbf{x}}_t - \eta_t \mathbf{m}_t], \quad \widehat{\mathbf{x}}_{t+1} = \Pi_{\mathcal{X}}[\widehat{\mathbf{x}}_t - \eta_t \nabla f_t(\mathbf{x}_t)], \tag{E.4}$$

*where $\Pi_{\mathcal{X}}[\mathbf{x}] \triangleq \arg\min_{\mathbf{y} \in \mathcal{X}} \|\mathbf{x} - \mathbf{y}\|$, $\mathbf{m}_t = \nabla f_{t-1}(\mathbf{x}_{t-1})$ and $\eta_t = \min\{D/\sqrt{1 + \bar{V}_{t-1}}, 1/\gamma\}$, enjoys the following empirical gradient-variation bound:*

$$\text{REG}_T \leq 5D\sqrt{1 + \bar{V}_T} + \gamma D^2 - \frac{\gamma}{4} \sum_{t=2}^{T} \|\mathbf{x}_t - \mathbf{x}_{t-1}\|^2 + \mathcal{O}(1).$$

**Lemma 11.** *Under Assumptions 1 and 2, if the loss functions are $\alpha$-exp-concave, optimistic OMD with the following update rule:*

$$\mathbf{x}_t = \arg\min_{\mathbf{x} \in \mathcal{X}} \left\{ \langle \mathbf{m}_t, \mathbf{x} \rangle + \mathcal{D}_{\psi_t}(\mathbf{x}, \widehat{\mathbf{x}}_t) \right\}, \quad \widehat{\mathbf{x}}_{t+1} = \arg\min_{\mathbf{x} \in \mathcal{X}} \left\{ \langle \nabla f_t(\mathbf{x}_t), \mathbf{x} \rangle + \mathcal{D}_{\psi_t}(\mathbf{x}, \widehat{\mathbf{x}}_t) \right\},$$

*where $\psi_t(\cdot) = \frac{1}{2}\|\cdot\|_{U_t}^2$,[7] $U_t = \gamma I + \frac{\alpha G^2}{2} I + \frac{\alpha}{2} \sum_{s=1}^{t-1} \nabla f_s(\mathbf{x}_s) \nabla f_s(\mathbf{x}_s)^\top$ and $\mathbf{m}_t = \nabla f_{t-1}(\mathbf{x}_{t-1})$, enjoys the following empirical gradient-variation bound:*

$$\text{REG}_T \leq \frac{16d}{\alpha} \ln\left(1 + \frac{\alpha}{8\gamma d} \bar{V}_T\right) + \frac{1}{2}\gamma D^2 - \frac{\gamma}{4} \sum_{t=2}^{T} \|\mathbf{x}_t - \mathbf{x}_{t-1}\|^2 + \mathcal{O}(1).$$

**Lemma 12.** *Under Assumptions 1 and 2, if the loss functions are $\lambda$-strongly convex, optimistic OGD (E.4) with $\eta_t = 2/(\gamma + \lambda t)$ and $\mathbf{m}_t = \nabla f_{t-1}(\mathbf{x}_{t-1})$, enjoys the following empirical gradient-variation bound:*

$$\text{REG}_T \leq \frac{16G^2}{\lambda} \ln\left(1 + \lambda \bar{V}_T\right) + \frac{1}{4}\gamma D^2 - \frac{\gamma}{8} \sum_{t=2}^{T} \|\mathbf{x}_t - \mathbf{x}_{t-1}\|^2 + \mathcal{O}(1).$$

*Proof of Lemma 10.* The proof mainly follows Theorem 11 of Chiang et al. [2012]. Following the standard analysis of optimistic OMD, e.g., Theorem 1 of Zhao et al. [2021], it holds that

$$\sum_{t=1}^{T} f_t(\mathbf{x}_t) - \sum_{t=1}^{T} f_t(\mathbf{x}^\star) \leq \underbrace{\sum_{t=1}^{T} \eta_t \|\nabla f_t(\mathbf{x}_t) - \mathbf{m}_t\|^2}_{\text{ADAPTIVITY}} + \underbrace{\sum_{t=1}^{T} \frac{1}{\eta_t}(\mathcal{D}_\psi(\mathbf{x}^\star, \widehat{\mathbf{x}}_t) - \mathcal{D}_\psi(\mathbf{x}^\star, \widehat{\mathbf{x}}_{t+1}))}_{\text{OPT-GAP}}$$

$$- \underbrace{\sum_{t=1}^{T} \frac{1}{\eta_t}(\mathcal{D}_\psi(\widehat{\mathbf{x}}_{t+1}, \mathbf{x}_t) + \mathcal{D}_{\psi_t}(\mathbf{x}_t, \widehat{\mathbf{x}}_t))}_{\text{STABILITY}}, \tag{E.5}$$

where $\mathbf{x}^\star \in \arg\min_{\mathbf{x} \in \mathcal{X}} \sum_{t=1}^{T} f_t(\mathbf{x})$ and $\psi(\cdot) \triangleq \frac{1}{2}\|\cdot\|^2$. The adaptivity term satisfies

$$\text{ADAPTIVITY} = \sum_{t=1}^{T} \eta_t \|\nabla f_t(\mathbf{x}_t) - \mathbf{m}_t\|^2 \leq D \sum_{t=1}^{T} \frac{\|\nabla f_t(\mathbf{x}_t) - \nabla f_{t-1}(\mathbf{x}_{t-1})\|^2}{\sqrt{1 + \sum_{s=1}^{t-1} \|\nabla f_s(\mathbf{x}_s) - \nabla f_{s-1}(\mathbf{x}_{s-1})\|^2}}$$

$$\leq 4D\sqrt{1 + \sum_{t=1}^{T} \|\nabla f_t(\mathbf{x}_t) - \nabla f_{t-1}(\mathbf{x}_{t-1})\|^2} + 4DG^2,$$

where the last step uses $\sum_{t=1}^{T} a_t / \sqrt{1 + \sum_{s=1}^{t-1} a_s} \leq 4\sqrt{1 + \sum_{t=1}^{T} a_t} + \max_{t \in [T]} a_t$ [Pogodin and Lattimore, 2019, Lemma 4.8]. Next, we move on to the optimality gap,

$$\text{OPT-GAP} = \sum_{t=1}^{T} \frac{1}{\eta_t}(\mathcal{D}_\psi(\mathbf{x}^\star, \widehat{\mathbf{x}}_t) - \mathcal{D}_\psi(\mathbf{x}^\star, \widehat{\mathbf{x}}_{t+1})) = \sum_{t=1}^{T} \frac{1}{2\eta_t}(\|\mathbf{x}^\star - \widehat{\mathbf{x}}_t\|^2 - \|\mathbf{x}^\star - \widehat{\mathbf{x}}_{t+1}\|^2)$$

---
[7]$\|\mathbf{x}\|_{U_t} \triangleq \sqrt{\mathbf{x}^\top U_t \mathbf{x}}$ refers to the matrix norm.

$$\leq \frac{\|\mathbf{x}^\star - \widehat{\mathbf{x}}_1\|^2}{2\eta_1} + \sum_{t=2}^{T} \left( \frac{1}{2\eta_t} - \frac{1}{2\eta_{t-1}} \right) \|\mathbf{x}^\star - \widehat{\mathbf{x}}_t\|^2$$

$$\leq \frac{D}{2}(1 + \gamma D) + D^2 \sum_{t=2}^{T} \left( \frac{1}{2\eta_t} - \frac{1}{2\eta_{t-1}} \right) \leq \frac{D}{2}(1 + \gamma D) + \frac{D^2}{2\eta_T}$$

$$= \gamma D^2 + \frac{D}{2}\sqrt{1 + \bar{V}_T} + \mathcal{O}(1).$$

Finally, we analyze the stability term,

$$\text{STABILITY} = \sum_{t=1}^{T} \frac{1}{2\eta_t}(\|\widehat{\mathbf{x}}_{t+1} - \mathbf{x}_t\|^2 + \|\mathbf{x}_t - \widehat{\mathbf{x}}_t\|^2) \geq \sum_{t=2}^{T} \frac{1}{2\eta_t}(\|\widehat{\mathbf{x}}_t - \mathbf{x}_{t-1}\|^2 + \|\mathbf{x}_t - \widehat{\mathbf{x}}_t\|^2)$$

$$\geq \sum_{t=2}^{T} \frac{1}{4\eta_t}\|\mathbf{x}_t - \mathbf{x}_{t-1}\|^2 \geq \frac{\gamma}{4} \sum_{t=2}^{T} \|\mathbf{x}_t - \mathbf{x}_{t-1}\|^2.$$

Combining the above inequalities completes the proof. $\qquad\square$

*Proof of Lemma 11.* The proof mainly follows Theorem 15 of Chiang et al. [2012]. Denoting by $\mathbf{x}^\star \in \arg\min_{\mathbf{x} \in \mathcal{X}} \sum_{t=1}^{T} f_t(\mathbf{x})$, it holds that

$$\sum_{t=1}^{T} f_t(\mathbf{x}_t) - \sum_{t=1}^{T} f_t(\mathbf{x}^\star) \leq \underbrace{\sum_{t=1}^{T} \|\nabla f_t(\mathbf{x}_t) - \mathbf{m}_t\|_{U_t^{-1}}^2}_{\text{ADAPTIVITY}} + \underbrace{\sum_{t=1}^{T} (\mathcal{D}_{\psi_t}(\mathbf{x}^\star, \widehat{\mathbf{x}}_t) - \mathcal{D}_{\psi_t}(\mathbf{x}^\star, \widehat{\mathbf{x}}_{t+1}))}_{\text{OPT-GAP}}$$

$$\underbrace{- \sum_{t=1}^{T} (\mathcal{D}_{\psi_t}(\widehat{\mathbf{x}}_{t+1}, \mathbf{x}_t) + \mathcal{D}_{\psi_t}(\mathbf{x}_t, \widehat{\mathbf{x}}_t))}_{\text{STABILITY}} \underbrace{- \frac{\alpha}{2} \sum_{t=1}^{T} \|\mathbf{x}_t - \mathbf{x}^\star\|_{\nabla f_t(\mathbf{x}_t)\nabla f_t(\mathbf{x}_t)^\top}^2}_{\text{NEGATIVITY}},$$

where the last term is imported by the definition of exp-concavity. First, the optimality gap satisfies

$$\text{OPT-GAP} = \frac{1}{2} \sum_{t=1}^{T} \|\mathbf{x}^\star - \widehat{\mathbf{x}}_t\|_{U_t}^2 - \frac{1}{2} \sum_{t=1}^{T} \|\mathbf{x}^\star - \widehat{\mathbf{x}}_{t+1}\|_{U_t}^2$$

$$\leq \frac{1}{2} \|\mathbf{x}^\star - \widehat{\mathbf{x}}_1\|_{V_1}^2 + \frac{1}{2} \sum_{t=1}^{T} (\|\mathbf{x}^\star - \widehat{\mathbf{x}}_{t+1}\|_{U_{t+1}}^2 - \|\mathbf{x}^\star - \widehat{\mathbf{x}}_{t+1}\|_{U_t}^2)$$

$$\leq \frac{1}{2}\gamma D^2 + \frac{\alpha G^2 D^2}{4} + \frac{\alpha}{4} \sum_{t=1}^{T} \|\mathbf{x}^\star - \widehat{\mathbf{x}}_{t+1}\|_{\nabla f_t(\mathbf{x}_t)\nabla f_t(\mathbf{x}_t)^\top}^2.$$

We handle the last term by leveraging the negative term imported by exp-concavity:

$$\text{OPT-GAP} - \text{NEGATIVITY}$$

$$\leq \frac{1}{2}\gamma D^2 + \frac{\alpha}{4} \sum_{t=1}^{T} \|\mathbf{x}^\star - \widehat{\mathbf{x}}_{t+1}\|_{\nabla f_t(\mathbf{x}_t)\nabla f_t(\mathbf{x}_t)^\top}^2 - \frac{\alpha}{2} \sum_{t=1}^{T} \|\mathbf{x}_t - \mathbf{x}^\star\|_{\nabla f_t(\mathbf{x}_t)\nabla f_t(\mathbf{x}_t)^\top}^2 + \mathcal{O}(1)$$

$$\leq \frac{1}{2}\gamma D^2 + \frac{\alpha}{2} \sum_{t=1}^{T} \|\mathbf{x}_t - \widehat{\mathbf{x}}_{t+1}\|_{\nabla f_t(\mathbf{x}_t)\nabla f_t(\mathbf{x}_t)^\top}^2 + \mathcal{O}(1),$$

where the local norm of the second term above can be transformed into $U_t$:

$$\frac{\alpha}{2} \sum_{t=1}^{T} \|\mathbf{x}_t - \widehat{\mathbf{x}}_{t+1}\|_{\nabla f_t(\mathbf{x}_t)\nabla f_t(\mathbf{x}_t)^\top}^2 \leq \frac{\alpha G^2}{2} \sum_{t=1}^{T} \|\mathbf{x}_t - \widehat{\mathbf{x}}_{t+1}\|^2 \leq \sum_{t=1}^{T} \|\mathbf{x}_t - \widehat{\mathbf{x}}_{t+1}\|_{U_t}^2.$$

Using the stability of optimistic OMD [Chiang et al., 2012, Proposition 7], the above term can be further bounded by

$$\sum_{t=1}^{T} \|\mathbf{x}_t - \widehat{\mathbf{x}}_{t+1}\|_{U_t}^2 \leq \sum_{t=1}^{T} \|\nabla f_t(\mathbf{x}_t) - \mathbf{m}_t\|_{U_t^{-1}}^2.$$

By choosing the optimism as $\mathbf{m}_t = \nabla f_{t-1}(\mathbf{x}_{t-1})$, the above term can be consequently bounded due to Lemma 19 of Chiang et al. [2012]:

$$\sum_{t=1}^{T} \|\nabla f_t(\mathbf{x}_t) - \nabla f_{t-1}(\mathbf{x}_{t-1})\|_{U_t^{-1}}^2 \leq \frac{8d}{\alpha} \ln\left(1 + \frac{\alpha}{8\gamma d} \bar{V}_T\right).$$

The last step is to analyze the negative stability term:

$$\text{STABILITY} = \sum_{t=1}^{T} (\mathcal{D}_{\psi_t}(\widehat{\mathbf{x}}_{t+1}, \mathbf{x}_t) + \mathcal{D}_{\psi_t}(\mathbf{x}_t, \widehat{\mathbf{x}}_t)) = \frac{1}{2} \sum_{t=1}^{T} \|\widehat{\mathbf{x}}_{t+1} - \mathbf{x}_t\|_{U_t}^2 + \frac{1}{2} \sum_{t=1}^{T} \|\mathbf{x}_t - \widehat{\mathbf{x}}_t\|_{U_t}^2$$

$$\geq \frac{\gamma}{2} \sum_{t=1}^{T} \|\widehat{\mathbf{x}}_{t+1} - \mathbf{x}_t\|^2 + \frac{\gamma}{2} \sum_{t=1}^{T} \|\mathbf{x}_t - \widehat{\mathbf{x}}_t\|^2 \geq \frac{\gamma}{4} \sum_{t=2}^{T} \|\mathbf{x}_t - \mathbf{x}_{t-1}\|^2.$$

Combining existing results, we have

$$\text{REG}_T \leq \frac{16d}{\alpha} \ln\left(1 + \frac{\alpha}{8\gamma d} \bar{V}_T\right) + \frac{1}{2}\gamma D^2 - \frac{\gamma}{4} \sum_{t=2}^{T} \|\mathbf{x}_t - \mathbf{x}_{t-1}\|^2 + \mathcal{O}(1),$$

which completes the proof. □

*Proof of Lemma 12.* The proof mainly follows Theorem 3 of Chen et al. [2023b]. Following the almost the same regret decomposition in Lemma 10, it holds that

$$\sum_{t=1}^{T} f_t(\mathbf{x}_t) - \sum_{t=1}^{T} f_t(\mathbf{x}^\star) \leq (\text{E.5}) \underbrace{- \frac{\lambda}{2} \sum_{t=1}^{T} \|\mathbf{x}_t - \mathbf{x}^\star\|^2}_{\text{NEGATIVITY}}.$$

First, we analyze the optimality gap,

$$\text{OPT-GAP} \leq \frac{1}{\eta_1} \mathcal{D}_\psi(\mathbf{x}^\star, \widehat{\mathbf{x}}_1) + \sum_{t=1}^{T} \left(\frac{1}{\eta_{t+1}} - \frac{1}{\eta_t}\right) \mathcal{D}_\psi(\mathbf{x}^\star, \widehat{\mathbf{x}}_{t+1})$$

$$\leq \frac{1}{4}(\gamma + \lambda)D^2 + \frac{\lambda}{4} \sum_{t=1}^{T} \|\mathbf{x}^\star - \widehat{\mathbf{x}}_{t+1}\|^2.$$

We handle the last term by leveraging the negative term imported by strong convexity:

$$\text{OPT-GAP} - \text{NEGATIVITY} \leq \frac{1}{4}(\gamma + \lambda)D^2 + \frac{\lambda}{4} \sum_{t=1}^{T} \|\mathbf{x}^\star - \widehat{\mathbf{x}}_{t+1}\|^2 - \frac{\lambda}{2} \sum_{t=1}^{T} \|\mathbf{x}_t - \mathbf{x}^\star\|^2$$

$$\leq \frac{1}{4}\gamma D^2 + \frac{\lambda}{2} \sum_{t=1}^{T} \|\mathbf{x}_t - \widehat{\mathbf{x}}_{t+1}\|^2 + \mathcal{O}(1).$$

The second term above can be bounded by the stability of optimistic OMD:

$$\frac{\lambda}{2} \sum_{t=1}^{T} \|\mathbf{x}_t - \widehat{\mathbf{x}}_{t+1}\|^2 \leq \frac{\lambda}{2} \sum_{t=1}^{T} \eta_t^2 \|\nabla f_t(\mathbf{x}_t) - \mathbf{m}_t\|^2 \leq \sum_{t=1}^{T} \eta_t \|\nabla f_t(\mathbf{x}_t) - \mathbf{m}_t\|^2.$$

Finally, we lower-bound the stability term as

$$\text{STABILITY} = \sum_{t=1}^{T} \frac{\gamma + \lambda t}{4} (\|\widehat{\mathbf{x}}_{t+1} - \mathbf{x}_t\|^2 + \|\mathbf{x}_t - \widehat{\mathbf{x}}_t\|^2) \qquad \left(\text{by } \eta_t = \frac{2}{\gamma + \lambda t}\right)$$

$$\geq \frac{\gamma}{4} \sum_{t=2}^{T} (\|\widehat{\mathbf{x}}_t - \mathbf{x}_{t-1}\|^2 + \|\mathbf{x}_t - \widehat{\mathbf{x}}_t\|^2) \geq \frac{\gamma}{8} \sum_{t=2}^{T} \|\mathbf{x}_t - \mathbf{x}_{t-1}\|^2.$$

Choosing the optimism as $\mathbf{m}_t = \nabla f_{t-1}(\mathbf{x}_{t-1})$, we have

$$\text{REG}_T \leq 2 \sum_{t=1}^{T} \eta_t \|\nabla f_t(\mathbf{x}_t) - \nabla f_{t-1}(\mathbf{x}_{t-1})\|^2 + \frac{1}{4}\gamma D^2 - \frac{\gamma}{8} \sum_{t=2}^{T} \|\mathbf{x}_t - \mathbf{x}_{t-1}\|^2 + \mathcal{O}(1).$$

To analyze the first term above, we follow the similar argument of Chen et al. [2023b]. By Lemma 9 with $a_t = \|\nabla f_t(\mathbf{x}_t) - \nabla f_{t-1}(\mathbf{x}_{t-1})\|^2$, $a_{\max} = 4G^2$, $A = \lceil \lambda \bar{V}_T \rceil$, and $b = \lambda$, it holds that

$$\sum_{t=1}^{T} \frac{1}{\lambda t} \|\mathbf{g}_t - \mathbf{g}_{t-1}\|^2 \leq \frac{4G^2}{\lambda} \ln\left(1 + \lambda \bar{V}_T\right) + \frac{4G^2}{\lambda} + \frac{1}{\lambda^2}.$$

Since $\eta_t = 2/(\gamma + \lambda t) \leq 2/(\lambda t)$, combining existing results, we have

$$\text{REG}_T \leq \frac{16G^2}{\lambda} \ln\left(1 + \lambda \bar{V}_T\right) + \frac{1}{4}\gamma D^2 - \frac{\gamma}{8} \sum_{t=2}^{T} \|\mathbf{x}_t - \mathbf{x}_{t-1}\|^2 + \mathcal{O}(1),$$

which completes the proof. $\qquad \square$

