# OpenReview forum: "Universal Online Learning with Gradient Variations: A Multi-layer Online Ensemble Approach"
_NeurIPS.cc/2023/Conference — NeurIPS 2023 spotlight_

### Official Review · Reviewer_rs9j · 2023-07-04

**Soundness:** 4 excellent
**Presentation:** 4 excellent
**Contribution:** 4 excellent
**Rating:** 8
**Confidence:** 4

**Summary:**

   In this work, the authors aim to develop an algorithm for online convex optimization that can adapt to the specific curvature of the loss function without knowing it in advance. The goal is to develop an algorithm that can adapt to strongly convex, exp-concave and convex loss functions dynamically, while also deriving some problem-dependent guarantees. They also ensure that their method is computationally efficient as it only requires one gradient query per round.

To do so, they present a three layer algorithm that estimates both the type of loss functions and its parameters. Crucially, this kind of multi-layer approach generally comes at the cost of logarithmic factors. These three algorithms follow pre existing structures (MSMWC (Chen et al. 2021) for the meta algorithms and an optimistic Online Mirror Descent for the base algorithm) but the authors rely on several tricks and refinements of the analysis to exploit negative terms in the regret analysis and obtain competitive regret bounds.

Finally, they also provide a novel decomposition of the regret which allows to reduce the number of gradients queries from $O(\log^2 T)$ to $O(1)$ at each round. This result is of independent interest, as they show that it can be used with different meta algorithms, such as Adapt-ML-Prod (van Erven and  Koolen 2016) to recover worst-case guarantees (whereas the main results of the paper are stated in terms of problem dependent terms).

**Strengths:**

This paper provides significant contributions to the field of OCO. Achieving optimal results simultaneously across several environments has been an important topic in learning theory in the past few years, and, by providing universal guarantees, this paper fits in this line of work. The presented algorithm builds upon existing results and provides non-trivial improvements to their analysis. These technical contributions are interesting and likely to be build upon in the future.

It is worth noting that this paper is particularly well written. The body of the paper focuses on providing detailed explanations for the choice of methods, and the authors take a particular care in higlighting the connections and differences with the rest of the litterature. Discussionss such as in Section 3.2.1 which highlight the different approaches that could have been considered and were discarded are not that common and should be encouraged. The proofs, deferred to the appendix, are clear and detailed. Appendix A provides several applications for the results, showing that the method can be used to bridge the gap between stochastic and adversarial online convex optimization and for two-player zero-sum games. In both cases, the proposed algorithm obtains regret bounds that are at least matching up to log-factors and sometimes improving upon the state of the art while considering a broader setting.

**Weaknesses:**

While many comparisons with other results in the litterature have been provided, it would be interesting to get comparisons with lower bounds, in particular for the small loss bounds that depend in F and V.
As the results achieved by the current method are matching these of Zhang et al. (2022), though with a better runtime complexity, it could also be interesting to get experimental results to compare them.

**Questions:**

Could you discuss if and how you think these results could be further improved?

---

> ### Author Rebuttal · Authors · 2023-08-08
>
> Thanks for the constructive feedback and appreciation of our work! Below we answer your questions on the lower bounds and the possibility of further improvements.
>
> **Q1**. It would be interesting to get comparisons with lower bounds, in particular for the small loss bounds that depend in F and V.
>
> **A1**. Thanks for the insightful suggestion. The optimal bounds for convex functions are $O(\sqrt{V_T})$ and $O(\sqrt{F_T})$. For exp-concave and strongly convex functions, the optimal results are $O(\ln V_T)$ and $O(\ln F_T)$, with an additional dimension dependence for exp-concave functions. For more details,  please refer to Lemma 9 of [1] and Corollary 3 of [2]. We will add a corresponding discussion in the next version.
>
> **References:**
>
> [1] Online Optimization with Gradual Variations, COLT 2012
>
> [2] Beyond Logarithmic Bounds in Online Learning, AISTATS 2012
>
> ---
>
> **Q2**. Could you discuss if and how you think these results could be further improved?
>
> **A2**. Thanks for the question. An open problem left in our work is to *achieve the optimal regret bound for convex functions* (by removing the extra $\ln V_T$ factor) *while attaining the current optimal rates for exp-concave and strongly convex ones.* In the future, we will investigate whether this term can be removed either by novel modifications and analysis of the MsMwC algorithm or by exploring the negative stability terms in other meta algorithms (e.g., Adapt-ML-Prod). We will include this discussion on future directions in the next version.
>
> ---
>
> We believe that our research occupies a nonnegligible position in universal online learning, especially given the significance of the gradient-variation quantity coupled with the novel techniques we've introduced. In the end, thanks again for recognizing the importance and contributions of our work.

---

> > ### Comment · Reviewer_rs9j · 2023-08-16
> >
> > Dear authors,
> >
> > Thank you for your answers, I don't have any further questions at this point.

---

### Official Review · Reviewer_osvq · 2023-07-09

**Soundness:** 4 excellent
**Presentation:** 4 excellent
**Contribution:** 4 excellent
**Rating:** 9
**Confidence:** 3

**Summary:**

This paper put together and designed new techniques to reach a high level of adaptivity in online convex optimization. It is a timely contribution that is useful to put together various algorithms optimal for different problem setups and to better adapt to the characteristic of each setup.

**Strengths:**

This is an excellent paper in presenting the results and the qualitative explanation of the techniques or proofs. The adaptivity contribution of the paper is also very satisfying in terms of general knowledge of machine learning.


**Weaknesses:**

I do not see any weakness in this paper. However, I have not checked the proofs in the appendix.

**Questions:**

No question.

**Limitations:**

None.

---

> ### Author Rebuttal · Authors · 2023-08-07
>
> Thanks for the valuable feedback and appreciation of our work! We believe that our research occupies a nonnegligible position in universal online learning, especially given the significance of the gradient-variation quantity coupled with the novel techniques we've introduced.
>
> The gradient variation is a fundamental problem-dependent quantity in modern online learning, largely due to its profound connections with both stochastic/adversarial OCO and games. As a result, our work finds broad application scope in these fields by obtaining universal algorithms with nearly optimal guarantees therein.
>
> Moreover, the techniques we've developed, such as the cascaded correction terms, the negative terms in MsMwC, and the meta-base regret decomposition with customized surrogate functions, are poised to capture the interest of the broader community and potentially spark further innovations.

---

### Official Review · Reviewer_wYJC · 2023-07-09

**Soundness:** 3 good
**Presentation:** 2 fair
**Contribution:** 3 good
**Rating:** 7
**Confidence:** 4

**Summary:**

This paper proposes algorithms that have data-dependent regrets and adaptive to the loss function classes. The algorithm improves on Zhang et al., 2022 in the sense that 1) it also achieves a gradient variation bound for convex Lipschitz losses and 2) it requires only one gradient evaluation at each round. The first is achieved by the introduction of optimism. The second is achieved by appropriately designed surrogate functions for the base learners.

---

The authors have addressed my questions. I keep the original rating.

**Strengths:**

1. The algorithm is new and improves on Zhang et al., 2022 (see the summary).
2. The paper clearly addresses the ideas behind algorithm design and analysis.

**Weaknesses:**

1. **Comparison of complexity.** Please also compare the number of the instances of the base learners required for each algorithm in Table 1. This information helps evaluate the computational complexity.
2. **Algorithm 1.** It is said that the base learners are set "as specified in Section 2," but Section 2 (preliminaries) does not give any algorithm.
1. **Presentation.**
    - Section 4 is presented in a hasty way. The reader may not be able to understand the ideas without reading Appendix C.
    - Table 1: The Lipschitzness and smoothness assumptions are missing. The assumptions may not hold in, e.g., online portfolio selection.
    - Ln. 35: That "the learner requires to know the function information (type and curvature) in advance to select suitable algorithms" only holds for textbook OCO algorithms and not general OCO.
    - The three aspects in Ln. 57--60 can be skipped as that will be addressed in detail in Ln. 79--89.
    - Ln. 58: The term "small-loss bound" has not been explained before. It is better to explain it as the term is not common sense to general machine learning researchers.
    - Ln. 125: That $N \approx 1 + \log T + \log T$ looks weird. Perhaps there is some typo.
2. **Typos.**
    - Ln. 57: aspect*s*
    - Ln. 219: *for* our purpose

**Questions:**

Please address the first two weaknesses and the possible typo in Ln. 125.

**Limitations:**

This is a theory paper. All assumptions are explicitly written. The potential issue regarding the computational complexity has been raised in the weakness block above.

---

> ### Author Rebuttal · Authors · 2023-08-08
>
> Thanks for the valuable feedback and appreciation of our work! Given that all three questions (computational complexity, base learners setup, and number $N$) relate to the base learner configuration, we provide a comprehensive explanation for it first, followed by the answers to your questions.
>
> ---
>
> **Base learner setup:** we are addressing a problem where both the type (strongly convex, exp-concave, or convex) and curvature coefficient (value of $\alpha$ or $\lambda$) of the functions are unknown. Ideally, the best base learner is the one that runs the algorithm for the right function type and the accurate guess of the curvature coefficient. Since both of them are unknown in our problem, we employ multiple base learners to hedge the uncertainty. The key to the problem is designing a meta algorithm to effectively track the best base learner.
>
> Specifically, we make *one base learner run an algorithm for convex functions* (only one is needed since there is no curvature coefficient here). For exp-concave and strongly convex functions, where the curvature coefficient's value is unknown, we employ multiple (the number will be illuminated later) base learners for each case. They all run an algorithm for the corresponding function type but with different guessed values for the curvature coefficient. Concretely, we discretize the possible range $[1/T,1]$ of $\alpha$ and $\lambda$ into $\lceil \log_2 T \rceil$ values ($\mathcal{H}$ in Line 128) and choose one of them as a guess. Overall, we need *$\lceil \log_2 T \rceil$ base learners for exp-concave and strongly convex functions*. Therefore, we use $N = 1 + \lceil \log_2 T \rceil + \lceil \log_2 T \rceil = O(\log T)$ base learners.
>
> ---
>
> **Q1**. Please also compare the number of instances of the base learners required for each algorithm in Table 1.
>
> **A1**. Thanks for the suggestion. Previous works employ a *two-layer* structure, necessitating $O(\log T)$ base learners. Our method leverages a *three-layer* structure and thus requires $O(\log^2 T)$ base learners. We will include this information in the revised version.
>
> **Q2**. It is said that the base learners are set "as specified in Section 2," but Section 2 (preliminaries) does not give any algorithm.
>
> **A2**. Thanks for the question. The base learner setup is provided in Lines 125-129 and please refer to the description above Q1 for more details. We will give a more detailed formalization in the revised version.
>
> **Q3**. Ln. 125: That $N \approx 1 + \log T + \log T$ looks weird. Perhaps there is some typo.
>
> **A3**. We appreciate your observation. It is not a typo. We represent $N$ as the *sum of three components* to denote the number of base learners needed for different kinds of functions. Specifically, one for convex functions (which don't have a curvature coefficient), and $O(\log T)$ for both exp-concave and strongly convex functions due to their unknown curvature coefficient.
>
> ---
>
> We are grateful for your meticulous review and will carefully revise the mentioned presentation issues and typos to enhance the paper's readability.

---

> > ### Comment · Reviewer_wYJC · 2023-08-14
> > **Thanks**
> >
> > Thanks for the response. I will keep the rating.
> >
> > **Q1.** Please include this information. Thanks.
> >
> > **Q2.** I raised this issue simply as a matter of presentation. The statement may confuse some readers, as Lines 125--129 are buried in a paragraph in the literature review. Please consider rewriting the related sentences to make things clearer.
> >
> > **Q3.** Please add a few words to explain the reason for the two $\log T$ terms.

---

> > > ### Author Response · Authors · 2023-08-14
> > > **Thanks for the suggestions!**
> > >
> > > **Response to Q1:** Thanks for the recommendation. We'll incorporate a comparison of base learner numbers from various methods in our revised version.
> > >
> > > **Response to Q2:** Thanks for the sincere advice. Given the extra page being permitted in the camera-ready version, we will rewrite this part to give a more detailed explanation of the base learners.
> > >
> > > **Response to Q3:** Thanks. We'll provide further clarifications in this section.
> > >
> > > Finally, thanks again for the thorough review and valuable suggestions!

---

### Official Review · Reviewer_ZhYs · 2023-07-18

**Soundness:** 3 good
**Presentation:** 1 poor
**Contribution:** 2 fair
**Rating:** 6
**Confidence:** 3

**Summary:**

A new algorithm for online learning is provided, together with improved regret bounds compared to the state-of-the-art:

The new regret guarantees are summarised in table 1 on page 2: in the case of a strongly convex function, the regret guarantee is of $O(min(\log(V_T),\log (F_T)))$ where $V_T$ and $F_T$ are the gradient variation and cumulative loss of the best comparator respectively (cf. equation on lines 53 to 54). In the case of exp-concave functions, the rate is the same, whereas in the case of a convex function, the rate is $\widetilde{O}(\min(\sqrt{V_t},\sqrt{F_T})$ instead. In terms of the regret bound, only the convex case is an improvement over the existing results ([3], $O(\sqrt{F_T})$). An additional claimed advantage is that the gradient query is reduced to $O(1)$ instead of $O(\log(T))$: this is valid for an improved algorithm (separate from the main algorithm presented in the paper) with additional surrogate losses, which is only presented in the appendix on page 18 (Algorithm 2)

The main algorithm itself is a combination of several existing techniques: at each iteration, a set of $KN$ experts $(k,i)\in [K]\times [N]$ each consisting of different gradient descent methods is maintained. Each expert computes the gradient update as per Optimistic OMD [3]. Then, for each fixed $k$, the experts $(k,i')$ for $i'\leq N$ are aggregated exactly as in MsMwC (cf. [1] and equation (2.3)), then the resulting experts are aggregated again via another use of MsMwC, with different parameters. The proof is a very long list of re-expressions of the regret via various existing results and additional calculations, drawing a lot of inspiration from [1], which itself improves on  techniques in [2].




=================Post-Rebuttal=============

After discussing with the authors, I am convinced that the presentation issues are theoretically fixable, and I still think the amount of work is adequate. Thus, I will keep my score and hope the authors fix all the minor errors as promised and try hard to make the paper more accessible to a broader audience.

==============




**References**

[1]  Sijia Chen, Wei-Wei Tu, Peng Zhao and  Lijun Zhang. Optimistic Online Mirror Descent for Bridging Stochastic and Adversarial Online Convex Optimization. ICML 2023.

[2] Pierre Gaillard,  Gilles Stoltz and Tim Van Erven. A Second-order Bound with Excess Losses. COLT 2014.

[3]  Lijun Zhang, Guanghui Wang, Jinfeng Yi and Tianbao Yang.  A Simple yet Universal Strategy for Online Convex Optimization. ICML 2022.

**Strengths:**

This is a highly technical and long paper and it is clear the authors know what they are doing. The parts of the proofs I managed to check are generally correct with only minor typos (though they are not reader friendly).

**Weaknesses:**

In my opinion, the main paper is poorly written. I understand that the authors are trying to explain the intuition for why they construct their algorithm in this way, but there isn't enough context or revision of existing methods to make the paper readable.
The paper reads like a soliloquy of a researcher trying to solve the problem at hand: for instance, in lines 140 to 148, the authors are saying "if we used Adapt-ML [2], bad things ill happen, what can be done about this?" then later in Section 3, their solution is proposed, which consists in using [3] instead, with a purportedly novel choice of optimism.

See also how the authors are using expressions like "Another try is to" on line 206. This is an issue not just in terms of grammar but also because it occurs in the context of a vague exposition that follows the stream of consciousness of the author rather than trying to present a cogent story.


Here are examples of more concrete **issues with the presentation**:


**1** There is a lot of talk in the main paper of "how to cancel out terms" which comes a long time before the part where the authors introduce their main algorithm. When they finally decide to do so on page 7, it is only the basic version of the algorithm (not the one with the $O(1)$ gradient calls) and it still vaguely refers to "the base learners described in Section 2", which I can only imagine refers to what is introduced in line 168 (section 3.1), but is still presented as new in line 177 "to unify all kinds of functions, we propose...."


**2** The concept of "optimism" isn't properly introduced either in the main or in the supplementary; the only definition is on line 72 in the intro: "a hyperparameter encoding historical information", which I am given to understand means slightly different things in different algorithms. A few more words of explanation here would be nice. Definitely, the base learners should be defined properly in the main text. In the current form, not even the reference to [3] is especially clear outside of Table 1. The same goes for the section names, terms such as "exogeneous negativity" are not very informative. In footnote 2, the authors even speak of "implementing $m_t$", which is definitely very informal and should be explained more clearly.


**3** Note also that the Bregman Divergence is note defined in lines 486 to 487 or 211-212 (page 6), despite the fact that it would make the description more complete and that it is indeed defined in the reference [1] when the same equation is introduced.

**4** In the proof of Lemma 2, the first equation is hard to understand without background in the field. the only explanation is "by standard analysis of optimistic OMD". It would be nice to say that this refers specifically to Theorem 1 of [4], which is explained better later in the appendix when the argument arises again (line 767 page 30). Since it is natural for the reader to read the proof of Lemma 2 first, the more detailed explanation should be present there as well.

**5** Similarly, the proof of Lemma 1 on page 14 uses equation E.1 from page 29, which is completely unnecessary as E.1. is derived from first principles.

**6** The proof of Lemma 2 is done for "an arbitrary comparator $u\in\Delta$, but the concept of comparator isn't really defined there, though it becomes clear from context after some effort. This also seems to run counter to the fact that in the main paper the authors write elsewhere in the paper that they "focus on the proof for fixed learning rate", which seems not to be the case in the case of the proof of Lemma 2.

**7** What does the notation $\|.\|_{U_t}$ mean on line 761? Is it explained somewhere?




I understand that I am not an expert in the field and my difficulties understanding the bigger picture is mostly due to this, but I do feel like in the case of well written papers, I am often able to make better progress, in less time, towards understanding more complex papers than this one in equally distant areas. However, I am not sure, it may be that the prerequisites are intrinsically larger.



**References**

[1]  Sijia Chen, Wei-Wei Tu, Peng Zhao and  Lijun Zhang. Optimistic Online Mirror Descent for Bridging Stochastic and Adversarial Online Convex Optimization. ICML 2023.

[2] Pierre Gaillard,  Gilles Stoltz and Tim Van Erven. A Second-order Bound with Excess Losses. COLT 2014.

[3]  Lijun Zhang, Guanghui Wang, Jinfeng Yi and Tianbao Yang.  A Simple yet Universal Strategy for Online Convex Optimization. ICML 2022.

[4]   Zhao, Peng ; Zhang, Yu-Jie ; Zhang, Lijun ; Zhou, Zhi-Hua.  Adaptivity and Non-stationarity: Problem-dependent Dynamic Regret for Online Convex Optimization. ArXiv. May 2023.

**Questions:**

**1** Can you describe the base learner from [3] concisely?  Perhaps rewrite the main paper in a more careful way?

**2** I am a bit confused about what you mean by the $O$ notation sometimes: which constants are considered to be "$O(1)$ exactly? In page 31, it seems like $\gamma$ is treated as a variable (not $O(1)$), but it is less clear in the case of $D$ and $G$: at the end of the proof of Lemma 10, **the $O(1)$ notation hides a term of $4DG^2$**. If you can absorb this into the $O(1)$ notation, why do you still need to write $\gamma D^2$? I can sort of see that the reason is that there is a $\gamma$ in there, but providing a bit of context might help.

**3** Do you really mean an equal sign at the second equality (first line) describing the Adaptivity term in page 30? it seems like if $1/\gamma<\frac{D}{\sqrt{1+\bar{V_T}}}$, **the equality wouldn't hold**. More worrying still, the inequality would be in the wrong direction.

**4** In lemma 3, you make the assumption that $(\ell_{t,k -m_{t,k}})^2 =\langle l_{t,k}-m_{t,k},p_{t,k}\rangle^2$. Could you provide more context there?

**5** Could you summarize the difference between your work and [1] more concisely? Why was the base learner [3] not used directly in [1]? What would happen if we used the correct base learner but did not use the hierarchical approach from your method?



**Minor mathematical errors:**

**1** I think the term  $-8\sum_{t=2}^T \|p_t-p_{t-1}\|$ should be $-4\sum_{t=2}^T \|p_t-p_{t-1}\|$ in Lemma 2. Indeed, the definition of "term C" excludes the factor of $1/2$ at the bottom of page 14 (as evidenced not just by the curly bracket but also by the use of Pinsker's inequality in line 498 as there would need to be a factor of 16 instead of 32 if the 1/2 were included): this factor of $1/2$ needs to be reintroduced in line 513 on page 16.  It would also be nice to quote Pinsker's inequality and perhaps add a citation at line 499 (page 15).

**2** In the calculations on page 15, lines 497-498, at the third equality on the first line, the $b_i$ should be at the numerator instead of the denominator.



**3** In the second line of the series of equations for the optimality gap in the proof of Lemma 10 on page 30, I think it should be $\leq$ instead of $=$ because there is a term of $-\|x^*-\hat{x}_{T+1}\|^2\frac{1}{2\eta_T}$ which is dropped out.


**4** In the statement of Lemma 5, it should be $\|p-q\|_1^2$ instead of  $\|p-p\|_1^2$


**5** In the proof of Lemma 1 on page 14, at the second line of equalities after line 482, I think it should be $\leq $ and not $=$ because of a missing $\langle g_T-g_{T+1},x_T-x_{T,i*}\rangle$ term which is added to make the sum look simpler.

**6** In the definitions of the iterations of $p_t$ and $q_t$ from [1] in line 486 page 14 and also in line 211 page 6, the last term should be $D(p,q_{t-1})$ instead of $D(p,q_{t})$, consistent with equation 6 in [1].  **NB**: I have used q instead of \hat{p} due to issues with markdown.




**Typos**




line 7: "the O notation" (missing "the")

line 146: constants=> constant

line 161, same thing

line 205 "it is still open that whether..>" ==> "it is still an open problem to determine whether"

line 217 "same of" ==> "same as"

line 252 "same of" ==> "same as"

line 488: "follows the similar logic of" ==> "follows a similar logic as"

line 506 "which equals to" ==> "which is equivalent to" (this somewhat changes the meaning and makes the line more understandable. Also, one might want to consider turning some of these inline equations into {align} blocks).

line 739 "it the optimal step size" ==> "if the optimal step size"

**Limitations:**

The improvement compared to [3] is somewhat disappointing compared to the much larger complexity of the algorithm: it seems that eh main improvement is to use a single algorithm for all types of functions and to reach similar rates, with a better rate in the case of convex functions. This is interesting theoretically but the algorithm seems too unwieldy to be used in either practice or future work.

I cannot fully vouch for the correctness as I often had to read the supplementary line by line without fully grasping the bigger picture. It is also hard to vouch for the originality compared to [3] as I am not familiar enough with it. Nevertheless, this seems like solid work, assuming the presentation was improved.

---

> ### Author Rebuttal · Authors · 2023-08-08
>
> Thanks for the valuable feedback and very careful check of our paper! We will carefully revise it according to the mentioned points. Due to the 6,000-character limit of this year's rebuttal, we first answer your questions about [Chen et al., 2023] and [Zhang et al., 2022] and then respond to presentation/typo issues and other questions in the next reply after the reviewer-author discussion begins.
>
> ---
>
> **Q1**. Can you describe the base learner from [Zhang et al., 2022] concisely? Perhaps rewrite the main paper in a more careful way?
>
> **A1**. We appreciate your question, and thanks for the suggestion. We will give a more detailed formalization in the revised version. Below we provide a comprehensive explanation for it.
>
> Specifically, we consider the problem where the kind (strongly convex, exp-concave, or convex) and curvature coefficient (value of $\alpha$ or $\lambda$) of the functions are unknown. Ideally, the best base learner is the one that runs the algorithm for the right function type and the accurate guess of the curvature coefficient. Since both of them are unknown in our problem, we employ multiple base learners to hedge the uncertainty. The key to the problem is designing a meta algorithm to effectively track the best base learner.
>
> Specifically, we set *one* base learner to run an algorithm for convex functions (only one is needed since there is no curvature coefficient here). For the rest two cases, e.g., for exp-concave functions, since the value of $\alpha$ is unclear, we discretize its potential range $[1/T,1]$ into $O(\log T)$ values ($\mathcal{H}$ in Line 128) and select one of them as a guess. In total, we require $O(\log T)$ base learners for the exp-concave case, all of which execute an algorithm for exp-concave functions but with *varying guesses* of the value of curvature coefficient $\alpha$. A similar arrangement also applies to strongly convex functions.
>
> ---
>
> **Q2**. Could you summarize the difference between your work and [Chen et al., 2023] more concisely? Why was the base learner [Zhang et al., 2022] not used directly in [Chen et al., 2023]? What would happen if we used the correct base learner but did not use the hierarchical approach from your method?
>
> **A2**. Thanks for the question. We compare with [Chen et al., 2023] from the following aspects:
>
> * **Difference:** the two works focus on *different problems*. Specifically, ours considers obtaining gradient-variation bounds in the universal problem. While theirs studied the *stochastically extended adversarial (SEA)* model, an interpolation between stochastic and adversarial OCO. Since the base learners in [Zhang et al., 2022] are employed to deal with the universal problem, they cannot be used directly in [Chen et al., 2023].
> * **Similarity:** one similarity is that both works consider three kinds of functions (strongly convex, exp-concave, and convex). Thus, our analysis of *base learners' negative terms* has some parallels with theirs, which are also *standard* in optimistic online learning.
> * **Application:** our method can be applied to their problem with nearly optimal universal results due to the profound connection between the gradient variation and the SEA model, resolving a *major open problem* therein (see their conclusion for the open problem).
>
> ---
>
> **Q3**. The improvement compared to [Zhang et al., 2022] is somewhat disappointing compared to the much larger complexity of the algorithm... This is interesting theoretically, but the algorithm seems too unwieldy to be used in either practice or future work.
>
> **A3**. Thanks for the feedback. While our algorithm might appear complex, it stands as the *first* to achieve gradient-variation bounds for the universal problem --- an *open problem* identified by [Zhang et al. 2022]. The improvement over [Zhang et al. 2022] is significant, not only due to the importance of the gradient-variation quantity itself but also because the improvement from $T$ to $V_T$ is *polynomial* for convex functions, whereas *logarithmic* in the rest cases (as stated in Line 69). Furthermore, our results can be applied to various learning problems, encompassing the *stochastically extended adversarial (SEA)* model and *games* (Appendix A). In doing so, we tackle a *major open question* posited by [Chen et al., 2023]. Please refer to Table 3 and Table 4 on Page 13 for an overview of our results in the aforementioned applications.
>
> The complexity of our three-layer method arises primarily because, to our knowledge, *only MsMwC* can yield the desired negative terms for cancellations (also one of our technical contributions) while securing a second-order regret bound. In the future, we will focus on obtaining the same rates with a two-level structure. A possible solution is to explore the negative stability terms in other meta algorithms (e.g., Adapt-ML-Prod). We will include a discussion on these future directions, especially concerning computational efficiency, in the updated version.
>
> ---
>
> We believe that our work offers valuable contributions to the community. We hope the above replies will address your concerns and would appreciate a reevaluation of our paper's score. And we are happy to provide further clarifications if needed in the following author-reviewer discussions.
>
> We also take this opportunity to sincerely thank you for the careful review, including the thorough examination of the proofs! Your suggestions are very insightful and important for further improving the paper.

---

> > ### Author Response · Authors · 2023-08-10
> > **Response to other questions**
> >
> > In this reply, we respond to presentation/typo issues and other questions, hoping to help you understand our work better. Since some LaTeX commends of `\hat{}`, `\boldsymbol{}` do not work well in OpenReview, we use other notations instead or simply omit them.
> >
> > ---
> >
> > In this part, we clarify the **presentation issues**.
> >
> > **Q1**. The presentation order.
> >
> > **A1**. We truly appreciate your advice. We will reorder the content in the revision to provide readers with an overview of the entire paper before delving into the specifics.
> >
> > **Q2**. The concept of "optimism"; The definition of base learners; Section names such as "exogenous negativity"; "implementing $m_t$".
> >
> > **A2**. We greatly appreciate your suggestions. We will provide more explanations about optimism when it first appears. Given that an extra page is allowed in the camera-ready version, we will include a more comprehensive formalization of base learners. And we will carefully revise the paper to remove not informative statements.
> >
> > **Q3**. The Bregman divergence is not defined.
> >
> > **A3**. Thanks. We will include the definition in the revised version.
> >
> > **Q4**. In the proof of Lemma 2, the first equation is hard to understand without a background in the field.
> >
> > **A4**. We appreciate your keen observation! We will cite Theorem 1 of Zhao et al. [2023] at this point.
> >
> > **Q5**. Referring to equation E.1 in the proof of Lemma 1 is unnecessary.
> >
> > **A5**. Thanks. We will remove it.
> >
> > **Q6**. The comparator and learning rate in Lemma 2.
> >
> > **A6**. Thanks for highlighting this. In Lemma 2, we provided a more general result with an arbitrary comparator and changing learning rates. This was done hoping that the negative stability term analysis would be comprehensive enough for readers interested solely in the MsMwC algorithm. We will state this clearly to prevent misunderstandings.
> >
> > **Q7**. What does the notation $\|\cdot\|_{U_t}$ mean on line 761? Is it explained somewhere?
> >
> > **A7**. Thanks for your sharp observation. It refers to the matrix norm: $\|\mathbf{x}\|_{U_t} = \sqrt{\mathbf{x}^\top U_t \mathbf{x}}$. We will provide its definition in the revised version.
> >
> > ---
> >
> > In this part, we answer your **questions**.
> >
> > **Q8**. The $O$-notation.
> >
> > **A8**. Thanks for raising this. The gradient norm $G$ and domain diameter $D$ are generally considered constants in constrained online learning. As a result, we incorporate them into the $O(1)$ term. Regarding $\gamma$, since it can take different values in various scenarios (e.g., please refer to Line 694 and Line 704), we do not include it in $O(1)$, even though $\gamma$ is always a constant.
> >
> > **Q9**. Do you really mean an equal sign at the second equality (first line) describing the Adaptivity term in page 30?
> >
> > **A9**. Thanks for the question. The correct derivation should use $\le$ due to $\eta_t = \min\{D/\sqrt{1+\bar{V}_t}, 1/\gamma\} \le D/\sqrt{1+\bar{V}_t}$.
> >
> > **Q10**. In lemma 3, you make the assumption that $(\ell_{t,k} - m_{t,k})^2 = \langle \ell_{t,k} - m_{t,k}, p_{t,k} \rangle$. Could you provide more context there?
> >
> > **A10**. We appreciate your insightful question. This is not an assumption but rather a *condition* that is inherently satisfied by our algorithm. For context, this condition serves to ensure a self-contained proof for our meta algorithm and has been validated, for instance, in Line 595. We appreciate this inquiry and will refine the manuscript to underscore that this is a condition, not an assumption.
> >
> > ---
> >
> > In this part, we answer your remarks on some minor **mathematical errors**.
> >
> > **Q11**. The constant before $\|p_t - p_{t-1}\|_1^2$ and the citation to Pinsker's inequality.
> >
> > **A11**. Thanks for the sharp observation! It is indeed a typo. We will rectify this typo by adjusting the constant to 4 and will cite Pinsker's inequality.
> >
> > **Q12**. In the calculations on page 15, lines 497-498, at the third equality on the first line, the $b_i$ should be at the numerator instead of the denominator.
> >
> > **A12**. Thanks. $b_i$ should be at the numerator, and we will revise it.
> >
> > **Q13**. In the second line of the series of equations for the optimality gap in the proof of Lemma 10 on page 30, I think it should be $\le$ instead of $=$.
> >
> > **A13**. Thanks. We will correct it to be $\le$.
> >
> > **Q14**. In the statement of Lemma 5, it should be $\|p - q\|_1^2$ instead of $\|p - p\|_1^2$.
> >
> > **A14**. Thanks for catching this error! It is a typo, and we will correct it.
> >
> > **Q15**. In the proof of Lemma 1 on page 14, at the second line of equalities after line 482, I think it should be $\le$ and not $=$.
> >
> > **A15**. Thanks. It is a typo, and we will revise it to be $\le$.
> >
> > **Q16**. In the definitions of the iterations of $p_t$ and $q_t$ from [1] in line 486 page 14 and also in line 211 page 6, the last term should be $D(p, q_{t-1})$, where $q$ denotes `\hat{p}`.
> >
> > **A16**. Thanks. We will revise it in the follow-up version.
> >
> > And thanks for finding some additional typos in both the main paper and the appendix. We will revise them carefully.

---

> > > ### Comment · Reviewer_ZhYs · 2023-08-18
> > > **Follow-up questions**
> > >
> > > Thanks for the many clarifications and for agreeing to correct the errors.  Please do not forget to include a clarification about which quantities you include in the O notation (Q8) and also an explanation of the difference with [Chen et al. 2023].
> > >
> > > I still have a few remaining questions:
> > >
> > > 1I am still not sure about your answer to my point about the equality at the top of page 30 (Proof of Lemma 10). I see you agree with me that it should be an inequality rather than an equality, but still, that doesn't fully explain why you are using the formula $\eta_t=\min(\frac{D}{1+\bar{V}_t},\frac{1}{\gamma})$. Indeed, it seems this condition is taken from the assumptions in Lemma 8 instead. The assumption in Lemma 10 is instead $\eta_t=\frac{2}{\gamma+\lambda_t}$. How do you reconcile this? What is the correct assumption on $\eta_t$ (and how does it ensure that the equation just before line 771 still holds)?
> > >
> > > 2. In your answer to my Q10, you claim that the condition is "inherently satisfied". Can you explain from which equation in the definition this follows ? (both in this rebuttal and later, in the paper).
> > >
> > > 3. In A2, you claim that by applying your results to Chen et al, you solve a "Major open problem". This seems like an overstatement. Was this open problem suggested at any time before the Arxiv publication of Chen et al in february 2023?

---

> > > > ### Author Response · Authors · 2023-08-19
> > > > **Response to the follow-up questions**
> > > >
> > > > Thanks for the follow-up questions and helpful comments. We will revise the paper according to your suggestions. Below we answer your technical questions.
> > > >
> > > > ---
> > > >
> > > > **Q1**. The step size issue in Lemma 8 and Lemma 10.
> > > >
> > > > **A1**. We clarify that the setup of the step size $\eta_t$ is *not* an assumption but used in the base algorithms. Lemma 8 and Lemma 10 provide the regret guarantees of base learners for *different* kinds of functions (convex and strongly convex), respectively.
> > > >
> > > > Briefly, as explained in A1 in the first response about the base learners from [Zhang et al., 2022], we employ diverse base learners to hedge the uncertainty of the unknown function type and curvature coefficient. Therefore, it is necessary to establish the regret guarantee for each function type separately. To this end, we provide Lemma 8 (for convex functions), Lemma 9 (for exp-concave functions), and Lemma 10 (for strongly convex functions). Specifically,
> > > >
> > > > - Lemma 8 provides the regret guarantee of the base learner for *convex* functions, which runs OGD with the step size $\eta_t = \min(D/\sqrt{1 + \bar{V}_t},1/\gamma)$.
> > > > - In contrast, Lemma 10 provides the regret guarantee of the base learner for *strongly convex* functions, which runs OGD but with a different step size $\eta_t = 2/(\gamma + \lambda t)$.
> > > > - Additionally, Lemma 9 provides the regret guarantee of the base learner for *exp-concave* functions, which runs the optimistic OMD with a local-norm regularizer.
> > > >
> > > > Therefore, Lemma 8 and Lemma 10 do not conflict since they refer to different base learners designed for different function families.
> > > >
> > > > ---
> > > >
> > > > **Q2**. Condition validation of Lemma 3.
> > > >
> > > > **A2**. Lemma 3 is only used in the proof of Theorem 2. Concretely, we use it in Line 595 to analyze the meta regret. Before applying Lemma 3, we have validated that its conditions can be fully satisfied, as can be seen in the second line of Line 594 (with detailed validations in Line 591-594). We will revise the paper to make it more clearly.
> > > >
> > > > ---
> > > >
> > > > **Q3**. SEA-related questions.
> > > >
> > > > **A3**. We mention that applying our universal algorithm to the SEA model directly addresses the major open problem *originally* proposed by [Chen et al., 2023]. So the open problem didn’t appear before Feb 2023. In fact, the study of SEA was initiated very recently by [Sachs et al., NeurIPS 2022], whose manuscript was initially released in Feb 2022 as an arXiv version and later published at NeurIPS 2022.
> > > >
> > > > In the following, we provide a brief review of the development of the SEA model and clarify the importance of the aforementioned open problem.
> > > >
> > > > - The SEA (Stochastically Extended Adversarial) model, serving as a natural bridge between stochastic optimization and adversarial online optimization, was recently proposed by [Sachs et al., NeurIPS 2022] and continues to be an area of active research.
> > > > - The most up-to-date findings are provided by [Chen et al., ICML 2023]. Although they introduced a unified OMD framework and achieved favorable guarantees for convex/exp-concave/strongly convex functions, respectively, their algorithms require separately designed parameter settings. Thus, a central open problem left by [Chen et al., 2023] is to design a (single) adaptive algorithm that can handle potentially different functions while maintaining optimal regret guarantees.
> > > > - This adaptivity concern has garnered interest from the community. Notably, [Sachs et al., arXiv 2023], a work concurrent to ours (their manuscript was posted to arXiv online on Mar 6, slightly ahead of NeurIPS ddl), also investigates the adaptivity of the SEA model but only attains partial adaptability. Our results are *strictly better than theirs*, achieving the state-of-the-art. For an overview of comparisons, please kindly refer to Table 3 on Page 13.
> > > >
> > > > **References:**
> > > >
> > > > [1] Sachs et al., Between Stochastic and Adversarial Online Convex Optimization: Improved Regret Bounds via Smoothness, NeurIPS 2022
> > > >
> > > > [2] Chen et al., Optimistic Online Mirror Descent for Bridging Stochastic and Adversarial Online Convex Optimization, ICML 2023
> > > >
> > > > [3] Sachs et al., Accelerated Rates between Stochastic and Adversarial Online Convex Optimization, arXiv 2023
> > > >
> > > > ---
> > > >
> > > > We greatly appreciate your thorough review and helpful feedback. We will work on refining the presentation to enhance clarity and avoid potential misunderstandings.

---

### Official Review · Reviewer_x4hL · 2023-07-25

**Soundness:** 4 excellent
**Presentation:** 4 excellent
**Contribution:** 2 fair
**Rating:** 6
**Confidence:** 3

**Summary:**

This paper proposes an online convex optimization algorithm that is adaptive in two ways to the environment.
Specifically, it can achieve logarithmic regrets for problems with good properties such as strong convexity and exp-concavity,
as well as $\sqrt{T}$ regret in the worst-case.
Further,
it achieves improved regret bounds for problems with small variation of the function gradients.
In addition, it works with only one gradient call per round.

**Strengths:**

- The paper is well structured and easy to follow.
- Existing studies are adequately reviewed and the paper is informative for the reader.
- The paper clearly explains the structure of the proposed method, the idea of the proof, and its implication in great detail.

**Weaknesses:**

Compared to the study by Zhang et al. [2022], the magnitude of the contribution of this paper appears somewhat limited.
Although the design of the algorithm is quite sophisticated, the main component is a combination of already known techniques.
It would be better to have a more detailed description of the challenges in algorithm design and analysis, as well as the new techniques proposed in this paper.

**Questions:**

Can you address the concerns described in Weaknesses above?

**Limitations:**

I have no concerns about the limitations and potential negative societal impact are adequately addressed.

---

> ### Author Rebuttal · Authors · 2023-08-08
>
> Thanks for the valuable feedback. We believe that our work has solid contributions and holds significant interest to the community. The submitted version may not have effectively conveyed our findings clearly, leading to the reviewer's perception. We will carefully revise the paper to clarify the unique contributions of our work. Herein, we elaborate on our work's results and techniques to highlight its unique significance.
>
> ---
>
> **Results:** our research offers more than an enhanced gradient-variation bound for convex functions over [Zhang et al., 2022]. Below we emphasize the importance of the problem and our results.
>
> * **Importance of problem:** obtaining universal gradient-variation bounds is highly important but challenging, left as an *open problem* by [Zhang et al., 2022] and now resolved by us. The importance comes from two aspects:
>   * The convex case, where the method of [Zhang et al., 2022] fails, is far more important than the other two cases (exp-concave and strongly convex) since the improvement from $T$ to $V_T$ is *polynomial* in the convex case, whereas *logarithmic* in the other two cases (as stated in Line 69).
>   * The gradient variation is a *fundamental* problem-dependent quantity in modern online learning due to its implications for worst-case and small-loss guarantees, and profound connections with adversarial/stochastic convex optimization and games.
> * **Importance of our results:** our results can be applied to various learning problems. In particular, applying them to the *stochastically extended adversarial (SEA)* model gives a *single* algorithm with *nearly optimal* regret bounds with different kinds of functions, thus resolving the *major open problem* left by [Chen et al., 2023] (see their conclusion for the open problem). Furthermore, our results can also be applied to *games*, providing universal guarantees therein. The applications are deferred to *Appendix A* due to page constraints. Please refer to Table 3 and Table 4 on Page 13 for an overview of our results in the aforementioned applications.
>
> ---
>
> **Techniques:** our method is not a combination of already known techniques but with some important innovations, as outlined in Lines 90-99. As far as we know, these techniques did not appear in previous works, which distinctly differentiate our method from that of [Zhang et al., 2022].
>
> * The first innovation is a *novel optimism design* (elaborated in Section 3.1), demonstrating that a straightforward modification of existing methods is insufficient to tackle this problem.
> * The second innovation involves a *novel meta-base regret decomposition* with customized surrogate losses for base learners (detailed in Section 4), reducing gradient complexity to one per round. Note that such a regret decomposition is *inapplicable* to [Zhang et al., 2022] because their method cannot deal with the positive stability terms as we do.
> * The third innovation is incorporating *cascaded correction terms* within a three-layer structure (Section 3.2.2). Although the idea of correction terms is not new to other problems, such as non-stationary online learning [Zhao et al., 2021] and the multi-scale expert problem [Chen et al., 2021], our work is the *first* to introduce it to universal online learning, and the application to a three-layer structure necessitated *an extensive adaptation* of the technique.
> * Two interesting novel byproducts arise in our techniques.
>   * The first one is the *negative terms in MsMwC* (Section 3.2.1), which may be of independent interest to the community.
>   * The second one is a simple method and analysis for the worst-case universal guarantees (detailed in Proposition 1 in Section 4) and a general regret decomposition in the online ensemble structure for exp-concave and strongly convex functions.
>
>
> ---
>
>
> If our responses have properly addressed your concerns, please consider updating your score. Thanks!

---

> > ### Comment · Reviewer_x4hL · 2023-08-16
> >
> > Thank you very much for your very thorough response.
> > I now have a better understanding of the importance of this research.
> > I maintain my positive score.

---

### Official Review · Reviewer_FwDP · 2023-07-26

**Soundness:** 2 fair
**Presentation:** 3 good
**Contribution:** 2 fair
**Rating:** 6
**Confidence:** 4

**Summary:**

Addressing to the drawback of Zhang et al. [2022], which does not enjoy gradient-variation bounds for convex functions, using an optimistic version of ADAPT-ML-PROD with a second-order bound of  [Wei et al.,  2016],three-layer online ensemble structure with two-layer meta learner running  MSMWC [Chen et al., 2021] with different parameter configurations, and each MSMWC-MID is further connected with N base learners to explore the unknown function information, is developed.$\mathcal{O}(\ln V_T)$, $\mathcal{O}(d \ln V_T)$ and $\hat{\mathcal{O}}(\sqrt{V_T})$ regret bounds for strongly convex, exp-concave and convex loss functions is obtained. two different levels of adaptivity with problem-dependent gradient variations, e.g., adaptive to unknown function information and benign Environments are fulfilled.

**Strengths:**

In Zhang et al. [2022], the convex case has not been addressed. a two-layer framework In Zhang et al. [2022] is generalized to three-layer online ensemble structure. a single algorithm with gradient-variation bounds for all kinds of functions is designed.

**Weaknesses:**

1 the derived results substantially depend on Zhang et al. [2022], [Chen et al., 2021], Zhao et al. [2021] and [Wei et al., 2016].
2 The results of the algorithm still belong to the class of MSMWC [Chen et al., 2021]. the decomposed regret and injecting cascaded correction terms to both the top and middle layer make the work replicate published findings without adding substantial knowledge, leading to lack of novelty.

**Questions:**

I believe that exists other gradient-variation bounds, which is similar to [Wei et al., 2016], Can better results be obtained by means of it?

**Limitations:**

the authors adequately addressed the limitations.

---

> ### Author Rebuttal · Authors · 2023-08-08
>
> Thanks for the insightful feedback. Below we clarify our unique contributions and differentiate our approach from existing ones, hoping to address your concerns.
>
> ---
>
> **Q1**. The derived results substantially depend on [Zhang et al., 2022], [Chen et al., 2021], [Zhao et al., 2021], and [Wei et al., 2016].
>
> **A1**. We acknowledge that some algorithm components of our approach are based on existing works, and thus the new ones might appear as "simple modifications" (depending on one's familiarity with online learning). However, effectively integrating them to obtain our results necessitates *non-trivial innovations*. We will include more elaborations on technical contributions in A2. Below, we emphasize the importance of the problem and our results.
>
> * **Importance of problem:** obtaining universal gradient-variation bounds is highly important but challenging, left as an *open problem* by [Zhang et al., 2022] and now resolved by us. The importance of the problem comes from two aspects:
>   * The convex case, where the method of [Zhang et al., 2022] fails, is far more important than the other two cases (exp-concave and strongly convex) since the improvement from $T$ to $V_T$ is *polynomial* in the convex case, whereas *logarithmic* in the other two cases (as stated in Line 69).
>   * The gradient variation is a *fundamental* problem-dependent quantity in modern online learning due to its implications for worst-case and small-loss guarantees, and profound connections with adversarial/stochastic convex optimization and games.
>
> * **Importance of our results:** our results can be applied to various learning problems. In particular, applying them to the *stochastically extended adversarial (SEA)* model gives a *single* algorithm with *nearly optimal* regret bounds for different kinds of functions, thus resolving the *major open problem* left by [Chen et al., 2023] (see their conclusion for the open problem).  Furthermore, our results can also be applied to *games*, providing universal guarantees therein. The applications are deferred to *Appendix A* due to page constraints. Please refer to Table 3 and Table 4 on Page 13 for an overview of our results in the aforementioned applications.
>
> To summarize, while some algorithm components of our method are based on existing works, we have made non-trivial usages and effectively resolved *two open problems* raised in the literature. We believe that our work has solid contributions and holds significant interest to the community.
>
> ---
>
> **Q2**. The results of the algorithm still belong to the class of MSMWC [Chen et al., 2021]. the decomposed regret and injecting cascaded correction terms to both the top and middle layer make the work replicate published findings without adding substantial knowledge, leading to a lack of novelty.
>
> **A2**. We respectfully disagree with this comment. While inspired by existing works, effectively unifying them to solve our problem requires comprehensive uses of them. Besides, there are also unique challenges in our problem, which necessitate novel technical contributions for solving them.
>
> Below we discuss the mentioned techniques (MsMwC, regret decomposition, and cascaded correction terms), hoping to address your concerns.
>
> * **MsMwC:** our meta algorithm resides within MsMwC but incorporates novel techniques, including a *novel optimism design* (detailed in Section 3.1) and *negative stability terms in the analysis* (Section 3.2.1), which may be of independent interest to the community.
> * **Regret decomposition:** our regret decomposition in Section 4 is novel, allowing the algorithm to use only one gradient query in each round. Note that such a regret decomposition is *inapplicable* to [Zhang et al., 2022] because their method cannot deal with the positive stability terms as we do.
> * **Cascaded correction terms:** although our use of cascaded correction terms is not entirely new, the application to a three-layer structure necessitated *an extensive adaptation* of the technique. Importantly, our research focus deviates significantly from the existing literature --- our work is the *first* to introduce corrections for universal online learning, whereas prior works use it for different purposes, such as non-stationary online learning [Zhao et al., 2021] and the multi-scale expert problem [Chen et al., 2021].
>
> ---
>
> **Q3**. I believe that exists other gradient-variation bounds which are similar to [Wei et al., 2016]. Can better results be obtained by means of it?
>
> **A3**. We are not sure if we get this question accurately. As far as we know, simply modifying [Wei et al., 2016] does not suffice to obtain gradient-variation universal regret bounds (as explained in Line 164-176). This is due to the lack of negative terms in their regret analysis, which makes it hard to perform cancellations as we did in our work.
>
> ---
>
> We will revise the paper to ensure the readers understand our key contributions better. If our responses have properly addressed your concerns, please consider updating your score. Thanks!

---

> > ### Author Response · Authors · 2023-08-20
> >
> > Dear Reviewer FwDP,
> >
> > We sincerely appreciate your helpful comments. As the period for author-reviewer discussions is coming to an end, we kindly ask whether our response has properly addressed your concerns and potential misunderstandings. Please let us know if you have any more questions, and we are happy to provide further clarification. Thanks!
> >
> > Best Regards,
> >
> > Authors

---

### Decision · Program_Chairs · 2023-09-21

**Decision:**

Accept (spotlight)

**Comment:**

Strengths:
The primary strength of this paper lies in its contribution to the field of online learning. The novel algorithm presented in this work has the potential to advance the state-of-the-art by significantly improving regret bounds. This innovation is a commendable achievement and undoubtedly contributes to the academic community's knowledge in this area.

Weaknesses:
Despite the remarkable contribution, it is evident that there are several areas in the paper's presentation that require attention in order to enhance its overall readability and accessibility. Both Reviewer ZhYs and Reviewer wYJC have aptly pointed out various presentation-related aspects that could be improved to make the paper more reader-friendly and comprehensible. These issues include clarity in explanations, organization of content, and the need for more detailed examples or illustrations to aid in understanding.

Recommendation:
Considering the strengths and weaknesses outlined above, I am inclined to recommend the acceptance of this paper. The condition for acceptance is that the authors should diligently address the feedback provided by Reviewer ZhYs and Reviewer wYJC. This includes improving the clarity of explanations, reorganizing the content for better flow, and incorporating additional examples or illustrations where necessary. By addressing these concerns, the paper's overall quality can be significantly enhanced.